# CONTRASTIVE POSITIVE UNLABELED LEARNING

## ABSTRACT

Positive Unlabeled(PU) learning refers to the task of learning a binary classifier given a few labeled positive samples, and a set of unlabeled samples (which could be positive or negative). Majority of the existing approaches rely on additional knowledge of the class prior, which is unavailable in practice. Furthermore, these methods tend to perform poorly in low-data regimes, especially when very few positive examples are labeled. In this paper, we propose a novel PU learning framework that overcomes these limitations. We start by learning a feature space through pretext-invariant representation learning and then apply pseudo-labeling to the unlabeled examples, leveraging the cluster-preserving property of the representation space. Overall, our proposed PU learning framework handily outperforms state-of-the-art PU learning methods across several standard PU benchmark datasets, while not requiring a-priori knowledge or estimate of class prior. Remarkably, our method remains effective even when labeled data is scant, where previous PU learning algorithms falter. We also provide simple theoretical analysis motivating our proposed algorithms.

## 1 INTRODUCTION

This paper studies classical Positive Unlabeled (PU) learning – the *weakly supervised task of learning a binary (positive vs negative) classifier in the absence of any explicitly labeled negative examples, i.e., using an incomplete set of positives and a set of unlabeled samples*.

This setting is frequently encountered in several real-world applications, especially where obtaining negative samples is either resource-intensive or impractical. For instance, consider *personalized recommendation systems*, the training data typically consists of recorded user interactions, such as the items shown to the user and the items they clicked on. While the clicked items are considered positive preferences, the items not clicked on cannot be assumed to be negatives and should be treated as unlabeled, along with items that were not shown to the user (Naumov et al., 2019; Chen et al., 2021; Kelly & Teevan, 2003). Similarly, PU learning has also found applications in diverse domains such as drug, gene, and protein identification (Yang et al., 2012; Elkan & Noto, 2008), anomaly detection (Blanchard et al., 2010), fake news detection (Ren et al., 2014), matrix completion (Hsieh et al., 2015), data imputation (Denis, 1998), named entity recognition (NER) (Peng et al., 2019) and face recognition (Kato et al., 2018) among others.

PU Learning can also be viewed as a particular instance of *learning with class dependent label noise*. However, due to the unavailability of negative examples, *statistically consistent unbiased risk estimation is generally infeasible*, without imposing strong structural assumptions on $p(\mathrm{x})$ (Blanchard et al., 2010; Lopuhaa et al., 1991; Natarajan et al., 2013). In fact, we show that no robust ERM estimator can solve the equivalent class-dependent label noise problem reliably unless certain dataset conditions are met (see Appendix A.3.2, Lemma 1). The milestone is (Elkan & Noto, 2008), which additionally assumes a-priori knowledge of **class prior** $\pi_p = p(\mathrm{y} = 1|\mathrm{x})$ and treats the unlabeled examples as a mixture of positives and negatives: $p(\mathrm{x}) = \pi_p p(\mathrm{x}|\mathrm{y} = 1) + (1 - \pi_p)p(\mathrm{x}|\mathrm{y} = 0)$. (Blanchard et al., 2010; Du Plessis et al., 2014) build on this idea, and develop *statistically consistent and unbiased risk estimators* to perform cost-sensitive learning which has become the backbone of modern large scale PU learning algorithms (Chen et al., 2020d; Garg et al., 2021).

Unfortunately, these approaches suffer from two major issues (Also see Appendix A.3.1):

- **Knowledge of class prior** ($\pi_p$)**:** Firstly, in practice, the true $\pi_p$ is often not available and needs to be estimated $\hat{\pi}_p$ from the data. Further, these estimators are highly sensitive to the $\hat{\pi}_p$ - resulting in

significantly inferior performance when the estimate is inaccurate (Kiryo et al., 2017; Chen et al., 2020a). [1] This can be attributed to the fact that the estimation error $\|\pi_p - \hat{\pi}_p\| \leq \epsilon$ introduces bias $\sim \mathcal{O}(\epsilon)$ in the risk estimation. Thus, $\pi_p$ is often estimated via a separate Mixture Proportion Estimation (MPE) (Ramaswamy et al., 2016; Ivanov, 2020; Yao et al., 2021) sub-routine, adding significant computational overhead.

- **Limited supervision:** Moreover, even when the oracle class prior is available, if the supervision is limited, i.e., only a handful of $n_P$ positive examples are labeled, existing PU learning approaches can suffer from significant drop in performance or even complete collapse (Chen et al., 2020a). This is primarily due to the increased variance in risk estimation, which scales as $\sim \mathcal{O}(1/n_P)$.

> To this end, the primary aim of this work is to *develop a parameter-free approach that facilitates Positive Unlabeled (PU) learning, even in scenarios where the availability of labeled examples is limited, without requiring any additional side information, such as class prior.*

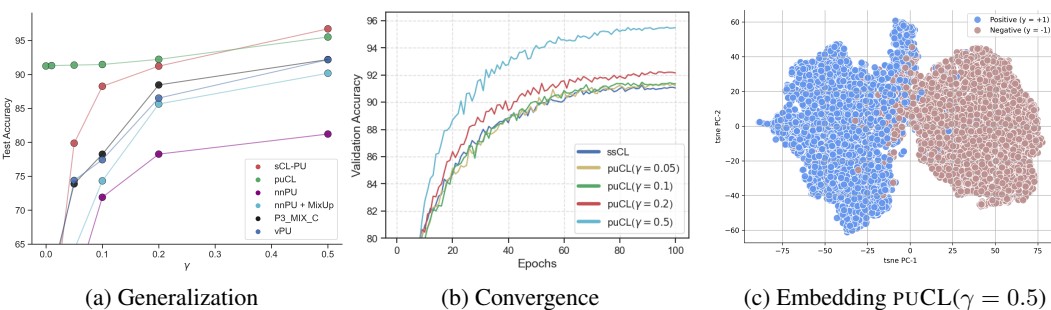

|  |  |  |
|---|---|---|
| (a) Generalization | (b) Convergence | (c) Embedding PUCL($\gamma = 0.5$) |

Figure 1: **Incorporating supervision:** In this experiment we train a ResNet-18 on **ImageNet-II: ImageWoof vs ImageNette** - two subsets of ImageNet-1k widely used in noisy label learning research https://github.com/fastai/imagenette. Amount of supervision is measured with fraction of positive to unlabeled samples $\gamma = \frac{n_P}{n_U}$. We keep the total number of samples $N = n_P + n_U$ fixed, while varying $n_P$. Observe that: **(a)Generalization:** PUCL remains robust across various levels of supervision, consistently outperforming its unsupervised counterpart, SSCL, and competing effectively with SCL-PU even in high supervision regimes. In contrast, SCL-PU experiences significant performance degradation, particularly in low-supervision regimes, where PUCL continues to demonstrate unique effectiveness. **(b) Convergence:** Incorporating additional positive also improves the convergence of PUCL over SSCL( Theorem 4 ). **(c) Embedding quality:** Finally, models that are trained with more PU supervision yields representation manifolds exhibiting better linear separability.

Orthogonal to the existing approaches, we introduce a novel framework that deftly leverages *semantic similarity* among samples, along with the available weak supervision to learn a representation space that exhibit linear separability, and develop that into an end-to-end method that remains effective even in low-data regime while obviating the need for a-priori knowledge or estimate of class prior.

> Our proposed PU Learning framework, as outlined in Algorithm 1, involves two key steps:
>
> - *Learning a cluster preserving representation manifold i.e. a feature space that preserves the underlying clusters by mapping semantically similar examples close to each other*. In particular, we adopt a simple PU specific modification of the standard self-supervised contrastive objective to take into account the available weak supervision in form of labeled positives resulting in significantly improved representations compared to the self-supervised counterpart.
>
> - *Assign pseudo-labels to the unlabeled examples by exploiting the geometry of the representation manifold learnt in the previous step*. These pseudo-labels are then used to train the downstream linear classifier using ordinary supervised objective e.g. cross-entropy (CE).

**Contributions.** Our proposed approach can be summarized into the following key contributions:

---

[1]For example, consider $\pi_p \neq \hat{\pi}_p = 1$ which leads to generate solution i.e. all the examples wrongly being predicted as positives (Chen et al., 2020a).

- In standard self-supervised contrastive learning approaches (Gutmann & Hyvärinen, 2010; Chen et al., 2020b), the loss function encourages similarity between samples and their augmentations, and dissimilarity between pairs of samples. We adopt a simple modification of this idea for the PU setting by including an additional similarity term for pairs of samples that are both labeled; we call our method PUCL and formally describe it in Algorithm 1(A). To the best of our knowledge, this paper represents the first work tailoring contrastive learning specifically to the PU setting.

- We compare PUCL to two natural baselines[2]: self-supervised contrastive learning SSCL which ignores the positive labels, and an adaptation of supervised contrastive learning SCL where all the unlabeled samples are treated as negatives. We show that the relative performances of these methods depend on the number of labeled positives, and that PUCL significantly outperforms the baselines, especially in settings where the number of labeled positives is small.

- We *theoretically ground* our empirical findings by providing a bias-variance justification, which provides more insight into the behavior of different contrastive objectives under various PU learning scenarios; see Section 2.2 for more details.

- Next, we develop a clever pseudo-labeling mechanism PUPL; that operates on the representation space learnt via PUCL. The key idea is to perform *clustering on the representation manifold, where we additionally leverage the representations of the labeled positive examples to guide the cluster assignments* as outlined in Algorithm 1(B). Theoretically, our algorithm enjoys $\mathcal{O}(1)$ multiplicative error compared to optimal clustering under mild assumption. It is worth noting that, due to judicious initialization, PUPL yields improved constant factor compared to kMeans++ ( Theorem 3 ).

- Since, even when available labeled positives are limited, representation learning is possible via proposed PUCL, PUPL is able to produce high quality pseudo-labels. Thus, our overall approach of *contrastive pretraining followed by pseudo-labeling* enables PU learning even when only a handful of labeled examples are available, a realistic setting where existing approaches often fail.

- Extensive experiments across several standard PU learning benchmark data sets reveal that our approach results in significant improvement in generalization performance compared to existing PU learning methods with $\sim 2\%$ improvement over current SOTA averaged over six benchmark data sets demonstrating the value of our approach.

## 2 CONTRASTIVE APPROACH TO PU LEARNING

### 2.1 PROBLEM SETUP

Let $x \in \mathbb{R}^d$ and $y \in Y = \{0, 1\}$ be the underlying input (i.e., feature) and output (label) random variables respectively and let $p(x, y)$ denote the true underlying joint density of $(x, y)$. Then, a PU training dataset is composed of a set $\mathcal{X}_P$ of $n_P$ positively labeled samples and a set $\mathcal{X}_U$ of $n_U$ unlabeled samples (a mixture of both positives and negatives) [3] i.e.

$$\mathcal{X}_{\text{PU}} = \mathcal{X}_P \cup \mathcal{X}_U, \ \mathcal{X}_P = \{\mathbf{x}_i^P\}_{i=1}^{n_P} \overset{\text{i.i.d.}}{\sim} p(x|y=1), \ \mathcal{X}_U = \{\mathbf{x}_i^U \overset{\text{i.i.d.}}{\sim} p(x)\}_{i=1}^{n_u} \tag{1}$$

Without the loss of generality, throughout the paper we assume that the overall classifier $f(\mathbf{x}) : \mathbb{R}^d \to \mathbb{R}^{|Y|}$ is parameterized in terms of (a) an encoder $g_{\mathbf{B}}(\cdot) : \mathbb{R}^d \to \mathbb{R}^k$ – a mapping function from the feature space to a lower dimensional manifold referred to as the *representation manifold* hereafter; and (b) a linear layer $v_{\mathbf{v}}(\cdot) : \mathbb{R}^k \to \mathbb{R}^{|Y|}$ i.e.

$$f_{\mathbf{v}, \mathbf{B}}(\mathbf{x}) = v_{\mathbf{v}} \circ g_{\mathbf{B}}(\mathbf{x}) = \mathbf{v}^T g_{\mathbf{B}}(\mathbf{x}), \ \forall \mathbf{x} \in \mathbb{R}^d. \tag{2}$$

The goal in PU learning is thus to train a binary classifier $f_{\mathbf{v}, \mathbf{B}}(\mathbf{x})$ from $\mathcal{X}_{\text{PU}}$ (1).

As discussed before, our proposed PU learning framework involves two key steps - (a) learning a mapping function $g_{\mathbf{B}}(\cdot) : \mathbb{R}^d \to \mathbb{R}^k$ to a cluster-preserving representation space via contrastive learning and (b) exploit the geometry of the feature space to pseudo-label the representations, used to train the subsequent linear layer $v_{\mathbf{v}}(\cdot) : \mathbb{R}^k \to \mathbb{R}^{|Y|}$. In the rest of this section, we discuss these two ideas in more detail, explore different design choices and systematically develop the framework.

---

[2]Additionally, in Appendix A.5.4 we discuss some recent weakly supervised contrastive approaches.

[3]This particular setup of how PU learning dataset is generated is referred to as the case-control setting (Bekker et al., 2019; Blanchard et al., 2010). However, in Appendix A.3.1 we also experiment with different contrastive objectives in the single-dataset setting (Bekker & Davis, 2020).

## 2.2 Representation Learning from Positive Unlabeled Data

Central to our approach is the construction of a representation space that fosters the proximity of semantically related instances while enforcing the separation of dissimilar ones. One way to obtain such a representation space via pretext-invariant representation learning where the representations $\mathbf{z}_i = g_{\mathbf{B}}(\mathbf{x}_i) \in \mathbb{R}^k$ are trained to be invariant to label-preserving distortions (Wu et al., 2018; Misra & Maaten, 2020)[4]. To prevent trivial solutions (Tian et al., 2021), a popular trick is to apply additional repulsive force between the embeddings of semantically dissimilar images, known as contrastive learning (Chopra et al., 2005; Schroff et al., 2015; Sohn, 2016). In particular, we study variants of InfoNCE family of losses (Oord et al., 2018) – a popular contrastive objective based on the idea of *Noise Contrastive Estimation* (NCE), a method of estimating the likelihood of a model by comparing it to a set of noise samples (Gutmann & Hyvärinen, 2010):

$$\mathcal{L}_{\text{CL}}^* = \underset{\substack{(\mathbf{x}_i, y_i) \sim p(\mathbf{x}, \mathbf{y})}}{\mathbb{E}} \underset{\substack{\mathbf{x}_j \sim p(\mathbf{x}|y_j = y_i) \\ \{\mathbf{x}_k\}_{k=1}^N \sim p(\mathbf{x}|y_k \neq y_i)}}{\mathbb{E}} \left[ \mathbf{z}_i \cdot \mathbf{z}_j - \log \left( \exp(\mathbf{z}_i \cdot \mathbf{z}_j) + \sum_{k=1}^N \exp(\mathbf{z}_i \cdot \mathbf{z}_k) \right) \right], \quad (3)$$

Where, the operator $\cdot$ is defined as: $\mathbf{z}_i \cdot \mathbf{z}_j = \frac{1}{\tau} \frac{\mathbf{z}_i^T \mathbf{z}_j}{\|\mathbf{z}_i\| \|\mathbf{z}_j\|}$ Intuitively, the loss projects the representation vectors onto hypersphere $\mathcal{S}_1^{k-1} = \{\mathbf{z} \in \mathbb{R}^k : \|\mathbf{z}\| = \frac{1}{\tau}\}$ and aims to minimize the angular distance between similar samples while maximizing the angular distance between dissimilar ones. $\tau \in \mathbb{R}^+$ is a hyper-parameter that balances the spread of the representations on the hypersphere (Wang & Isola, 2020). The objective aims to minimize the angular distance between similar samples while maximizing the angular distance between dissimilar ones.

While, several frameworks have been proposed to realize the infoNCE family of losses in the finite sample setting (Caron et al., 2020; Grill et al., 2020; He et al., 2020; Zbontar et al., 2021), in this paper we adopt the SimCLR framework (Chen et al., 2020d).

**Self Supervised Contrastive Learning (sSCL):** In the unsupervised setting, since identifying similar and dissimilar example pairs from the appropriate class conditionals is intractable; different augmentations of the same image are treated as similar, while rest are are considered as dissimilar pairs. In particular, for any random batch of samples $\mathcal{D} = \{\mathbf{x}_i\}_{i=1}^b$, corresponding *multi-viewed batch* is constructed by obtaining two augmentation (correlated views) of each sample: $\tilde{\mathcal{D}} = \{t(\mathbf{x}_i), t'(\mathbf{x}_i)\}_{i=1}^b$ where $t(\cdot), t'(\cdot) : \mathbb{R}^d \to \mathbb{R}^d$ are stochastic label preserving transformations, such as color distortion, cropping, flipping etc. To facilitate the subsequent discussion, let we introduce $\mathbb{I} \equiv \{1, \ldots, 2b\}$ corresponding to the elements of the multi-viewed batch. For augmentation indexed $i \in \mathbb{I}$, other augmentation originating from the same source sample is indexed as $a(i)$. Then, sSCL minimizes the following objective (Chen et al., 2020b) :

$$\mathcal{L}_{\text{sSCL}} = -\frac{1}{|\mathbb{I}|} \sum_{i \in \mathbb{I}} \left[ \mathbf{z}_i \cdot \mathbf{z}_{a(i)} - \log Z(\mathbf{z}_i) \right] \quad (4)$$

Where, $Z(\mathbf{z}_i) = \sum_{j \in \mathbb{I}} \mathbf{1}(j \neq i) \exp(\mathbf{z}_i \cdot \mathbf{z}_j)$ is the *finite-sample approximation of the partition function within the batch*. In practice, rather than computing the loss over the encoder outputs i.e. $\mathbf{z}_i = g_{\mathbf{B}}(\mathbf{x}_i)$; it is beneficial to feed it through a small *nonlinear projection network* $h_{\mathbf{\Gamma}}(\cdot) : \mathbb{R}^k \to \mathbb{R}^p$ to obtain a lower dimensional representation $\mathbf{z}_i = h_{\mathbf{\Gamma}} \circ g_{\mathbf{B}}(\mathbf{x}_i) \in \mathbb{R}^p$ (Chen et al., 2020b; Schroff et al., 2015). Note that *the $h_{\mathbf{\Gamma}}(\cdot)$ is only used during training and discarded during inference*.

### 2.2.1 Incorporating PU Supervision

Despite its ability to learn robust representations, sSCL is entirely agnostic to semantic annotations, hindering its ability to benefit from additional supervision, especially when such supervision is reliable. This lack of semantic guidance often leads to inferior visual representations compared to fully supervised approaches (He et al., 2020; Kolesnikov et al., 2019).

---

[4]Consider dataset $\mathbf{x} \in \mathcal{X}, y \in Y$ with underlying ground-truth labeling mechanism $y = \mathcal{Y}(\mathbf{x}) \in Y$. Parameterized representation function $f_{\mathbf{W}}(\cdot)$ is said to be invariant under transformation $t : \mathcal{X} \to \mathcal{X}$ that do not change the ground truth label i.e. $\mathcal{Y}(t(\mathbf{x})) = \mathcal{Y}(\mathbf{x})$ if $f_{\mathbf{W}}(t(\mathbf{x})) \approx f_{\mathbf{W}}(\mathbf{x})$.

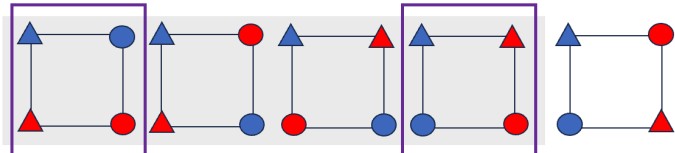

Figure 2: **Geometric intuition of incorporating contrastive (PU)-supervision:** Consider 1D feature space $\mathbf{x} \in \mathbb{R}$, e.g., $\mathbf{x}_i = 1$ if shape: triangle ($\blacktriangle$, $\blacktriangle$), $\mathbf{x}_i = 0$ if shape: circle ($\bullet$, $\bullet$). However, the labels are $y_i = 1$ if color: blue ($\blacktriangle$,$\bullet$) and $y_i = 1$ if color: red ($\blacktriangle$, $\bullet$). We show all possible configurations of arranging these points on the vertices of unit hypercube $\mathcal{H} \in \mathbb{R}^2$ when $\blacktriangle$ is fixed at $(0, 1)$. All the four shaded point configurations are favored by sSCL (4), since $\mathbf{x}_i = \mathbf{x}_j$ are placed neighboring vertices. However, the minimum loss configurations of PUCL (marked in rectangle) favors configurations, that additionally also preserve annotation consistency.

Motivated by these observations, *our goal is to design a contrastive loss that can leverage the available weak supervision in PU learning (in the form of labeled positive examples) in an efficient manner, to learn more discriminative representations compared to $\mathcal{L}_{\text{sSCL}}$.*

**Positive Unlabeled Supervised Contrastive Learning (sCL-PU):** In the fully supervised setting, Supervised Contrastive Learning (sCL) (Khosla et al., 2020) addresses this issue by utilizing the semantic annotations to guide the choice of similar and dissimilar pairs, often resulting in significant empirical gains. However, unfortunately in PU learning, *since negative examples are not available, it is non-trivial to extend* sCL *in this setting*. Consider the naive *disambiguation-free* (Li et al., 2022) adaptation, we refer as sCL-PU – wherein, the unlabeled training examples are treated as *pseudo-negative* instances[5] i.e.

$$\mathcal{L}_{\text{sCL-PU}} = -\frac{1}{|\mathbb{I}|} \sum_{i \in \mathbb{I}} \left[ \left( \mathbf{1}(i \in \mathbb{P}) \frac{1}{|\mathbb{P} \setminus i|} \sum_{j \in \mathbb{P} \setminus i} \mathbf{z}_i \cdot \mathbf{z}_j + \mathbf{1}(i \in \mathbb{U}) \frac{1}{|\mathbb{U} \setminus i|} \sum_{j \in \mathbb{U} \setminus i} \mathbf{z}_i \cdot \mathbf{z}_j \right) - \log Z(\mathbf{z}_i) \right] \tag{5}$$

Here, $\mathbb{P}$ and $\mathbb{U}$ denote the subset of indices in $\tilde{\mathcal{D}}$ that are labeled and unlabeled respectively i.e. $\mathbb{P} = \{i \in \mathbb{I} : \mathbf{x}_i = \mathcal{X}_P\}$, $\mathbb{U} = \{i \in \mathbb{I} : \mathbf{x}_i \in \mathcal{X}_U\}$. It is easy to follow that this naive adaptation suffers from statistical bias to estimate $\mathcal{L}_{\text{CL}}^*$ which becomes increasingly pronounced as the level of available supervision decreases as characterized in Theorem 1.

---

**Theorem 1.** $\mathcal{L}_{\text{sCL-PU}}$ (5) *is a biased estimator of* $\mathcal{L}_{\text{CL}}^*$ *characterized as follows:*

$$\mathbb{E}_{\mathcal{X}_{\text{PU}}} \left[ \mathcal{L}_{\text{sCL-PU}} \right] - \mathcal{L}_{\text{CL}}^* = \frac{\pi_p (1 - \pi_p)}{1 + \gamma} \left[ 2\tilde{\mu}_{\text{PN}} - (\mu_P^* + \mu_N^*) \right]$$

*Here,* $\mu_P^* = \mathbb{E}_{\mathbf{x}_i, \mathbf{x}_j \sim p(\mathbf{x}|y=1)} (\mathbf{z}_i \cdot \mathbf{z}_j)$ *and* $\mu_N^* = \mathbb{E}_{\mathbf{x}_i, \mathbf{x}_j \sim p(\mathbf{x}|y=0)} (\mathbf{z}_i \cdot \mathbf{z}_j)$ *capture the proximity between samples from same class marginals and* $\tilde{\mu}_{PN} = \mathbb{E}_{\mathbf{x}_i, \mathbf{x}_j \sim p(\mathbf{x}|y_i \neq y_j)} (\mathbf{z}_i \cdot \mathbf{z}_j)$ *captures the proximity between dissimilar samples.* $\gamma = \frac{n_P}{n_U}$ *captures the proportion of cardinality of labeled to unlabeled training subset.*

---

Consistent with the theoretical observation, our experiments (Figure 4) also reveal that, while in the low-supervision regime, $\mathcal{L}_{\text{SCL-PU}}$ might suffer from significant drop in generalization performance, it still results in significant improvements over the unsupervised $\mathcal{L}_{\text{sSCL}}$ when sufficient labeled positives are available. *indicating a bias-variance trade-off that can be further exploited to arrive at an improved loss.*

**Positive Unlabeled Contrastive Learning (PUCL):** In response, we consider a simple modification to the standard contrastive objective for the PU setting that is able to incorporate the available (weak) supervision judiciously. In particular, the modified objective dubbed PUCL leverages the available supervision as follows – each labeled positive anchor is attracted closer to all other labeled

---

[5]Note that, this is a is a reduction of PU Learning to learning with class-dependent noisy label with noise rates $E(\xi_P) = \frac{\pi_P}{\gamma + \pi_P}$ and $\xi_N = 0$. Refer to Appendix A.3.2 for more details.

positive samples in the batch, whereas an unlabeled anchor is only attracted to its own augmentation.

$$\mathcal{L}_{\text{PUCL}} = -\frac{1}{|\mathbb{I}|} \sum_{i \in \mathbb{I}} \left[ \mathbf{1}(i \in \mathbb{U}) \left( \mathbf{z}_i \cdot \mathbf{z}_{a(i)} \right) + \mathbf{1}(i \in \mathbb{P}) \frac{1}{|\mathbb{P} \setminus i|} \sum_{j \in \mathbb{P} \setminus i} \mathbf{z}_i \cdot \mathbf{z}_j - \log Z(\mathbf{z}_i) \right] \quad (6)$$

In essence, the unsupervised part in PUCL enforces consistency between representations learned via label-preserving augmentations i.e. between $\mathbf{z}_i$ and $\mathbf{z}_{a(i)} \forall i \in \mathbb{I}$, whereas the supervised component injects structural knowledge derived from labeled positives (see Figure 2, 7, Appendix A.5.2).

It is not hard to see that, $\mathcal{L}_{\text{PUCL}}$ is an unbiased estimator of $\mathcal{L}_{\text{CL}}^*$. Additionally, by using the available labeled positives it enjoys a lower variance compared to $\mathcal{L}_{\text{ssCL}}$ and the gap is a monotonically increasing function of $\gamma = \frac{n_{\text{P}}}{n_{\text{U}}}$ as summarized in Theorem 2.

---

**Theorem 2.** *Assume that $\mathbf{x}_i, \mathbf{x}_{a(i)}$ are i.i.d draws from the same class marginal (Saunshi et al., 2019; Tosh et al., 2021), then it follows that the objective functions $\mathcal{L}_{\text{ssCL}}$ (4) and $\mathcal{L}_{\text{PUCL}}$ (6) are unbiased estimators of $\mathcal{L}_{\text{CL}}^*$ (3). Additionally, it holds that:*

$$\Delta_\sigma(\gamma) \geq 0 \ \forall \gamma \geq 0 \ ; \ \Delta_\sigma(\gamma_1) \geq \Delta_\sigma(\gamma_2) \ \forall \gamma_1 \geq \gamma_2 \geq 0$$

*where, $\Delta_\sigma(\gamma) = \text{Var}(\mathcal{L}_{\text{ssCL}}) - \text{Var}(\mathcal{L}_{\text{PUCL}})$.*

---

**Empirical Evidence:** Our ablation experiments (see Figure 4, Table 2 and Appendix A.5.4) also suggest that indeed $\mathcal{L}_{\text{PUCL}}$ consistently produces representations that have improved linear separability over $\mathcal{L}_{\text{ssCL}}$ indicated by its improved downstream classification performance. These improvements are particularly pronounced when a sufficient number of labeled instances are available. Further, we observe that PUCL also significantly improves over SCL-PU objective especially in the practical settings where $\gamma$ is usually small while staying competitive for settings where a large fraction of the data is labeled as discussed in more detail in Section 3.

## 2.3 Positive Unlabeled Pseudo Labeling

While so far we have only discussed about learning a representation function, mapping the input features to a contrastive representation manifold, where semantically dissimilar samples are likely to be easily separable – performing inference on this manifold is not entirely obvious. Under the standard semi-supervised setting, the linear layer $v_\mathbf{v}(\cdot)$ can be trained using CE loss over the representations of the labeled data (Assran et al., 2020) to perform downstream inference. However, in the PU learning setting since we do not have any negative examples, a naive disambiguation-free approach would fit the bias in semantic annotations resulting in decision boundary deviation (Li et al., 2022) even for completely separable feature space (Figure 12). One natural approach would be to train the linear classifier using specialized cost-sensitive PU learning algorithm such as NNPU (Kiryo et al., 2017) - the de-facto approach to solve PU problems in practical settings and at the core of most modern PU learning algorithms. However, they suffer from the issues mentioned in Section 1.

In response, we ask the question: *Can we develop a scheme to train a downstream PU classifier over the representations learnt via contrastive pretraining, that remains effective even in extreme low-supervision regime (i.e. when only a handful of positive examples are labeled ) while not requiring the knowledge (or estimate) of dataset properties such as class prior ?*

To this end, we propose a clever pseudo-labeling mechanism that *obviates the need of class prior knowledge* and significantly simplifies the downstream inference problem which can now be solved using standard CE loss over the pseudo-labels – even when only a handful of labeled examples are available. Our algorithm relies on the fact that the contrastive representation manifold resulting from Algorithm 1(A) fosters proximity of semantically similar examples i.e. it is likely to preserve underlying clustering structure (Parulekar et al., 2023). In particular, we seek to find centers $C^* = \{\mu_{\text{P}}, \mu_{\text{N}}\}$ on the representation space, such that it approximately solves the NP-hard $k$-means problem (Mahajan et al., 2012) i.e. minimize the following potential function:

$$\phi^* = \phi(\mathcal{Z}_{\text{PU}}, C^*) = \sum_{\mathbf{x} \in \mathcal{Z}_{\text{PU}}} \min_{\mu \in C^*} \|\mathbf{x} - \mu\|^2 \ , \ \mathcal{Z}_{\text{PU}} = \{g_\mathbf{B}(\mathbf{x}_i) \in \mathbb{R}^k : \mathbf{x}_i \in \mathcal{X}_{\text{PU}}\} \quad (7)$$

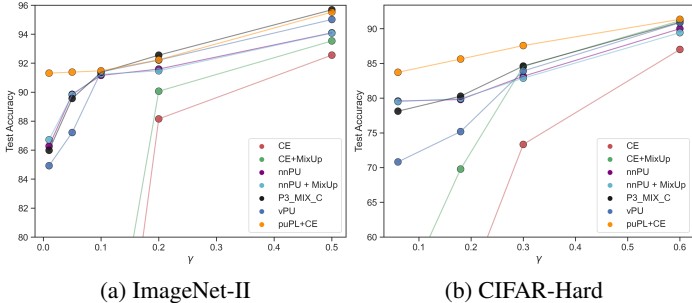

(a) ImageNet-II       (b) CIFAR-Hard

Figure 3: **Linear Probing:** Given pretrained embedding our goal is now to train a downstream linear model. In this experiment we take puCL($\gamma$) pretrained encoder (frozen) and train a linear classifier for downstream inference. In particular, we evaluate several popular SOTA PU Learning methods along with the proposed pseudo-labeling based approach. Our findings are particularly noteworthy in the context of low-data regimes. While traditional PU learning methods often struggle to maintain performance with limited data, our approach consistently demonstrates robust effectiveness.

| Contrastive PU Learning | | Datasets | | | | | | Average |
|---|---|---|---|---|---|---|---|---|
| Contrastive Loss | Linear Probing | F-MNIST-I ($\pi_p^* = 0.3$) | F-MNIST-II ($\pi_p^* = 0.7$) | CIFAR-I ($\pi_p^* = 0.4$) | CIFAR-II ($\pi_p^* = 0.6$) | STL-I ($\hat{\pi}_p = 0.51$) | STL-II ($\hat{\pi}_p = 0.49$) | |
| | | | | $n_P = 1000$ | | | | |
| SSCL | NNPU[†] | 89.5±0.9 | 85.9±0.5 | 91.7±0.3 | 90.0±0.4 | 81.1±1.2 | 81.4±0.8 | 86.6 |
| PU-SCL | NNPU[†] | 73.0±4.9 | 81.8±0.5 | 88.4±2.1 | 63.7±5.3 | 59.2±8.1 | 68.8±3.1 | 72.5 |
| PUCL | NNPU[†] | 90.0±0.1 | 86.8±0.4 | 91.8±0.2 | 90.3±0.5 | 81.5±0.7 | 82.6±0.4 | 87.2 |
| SSCL | PUPL (CE) | 91.4±1.2 | 86.2±0.6 | 91.6±0.9 | 90.7±0.4 | 81.2±1.6 | 81.3±0.7 | **87.1** |
| PU-SCL | PUPL (CE) | 77.8±0.3 | 82.5±4.1 | 90.1±1.2 | 68.9±7.5 | 58.5±8.2 | 73.9±1.2 | **75.3** |
| PUCL | PUPL (CE) | **91.8±0.8** | **89.2±0.3** | **92.3±1.9** | **91.2±0.5** | **83.8±1.4** | **84.5±0.7** | **88.8** |
| | | | $n_P = 3000$ | | | | $n_P = 2500$ | |
| SSCL | NNPU[†] | 89.6±0.1 | 85.0±0.4 | 92.3±0.3 | 92.7±0.3 | 81.6±0.9 | 84.2±1.0 | 87.6 |
| PU-SCL | NNPU[†] | 85.7±0.3 | 82.1±0.2 | 90.5±3.1 | 88.6±0.5 | 83.2±0.8 | 84.8±1.4 | 85.8 |
| PUCL | NNPU[†] | 90.3±0.1 | 87.0±0.7 | 93.2±0.1 | 92.9±0.1 | 84.9±0.7 | 85.1±0.7 | 88.9 |
| SSCL | PUPL (CE) | 90.1±0.2 | 88.8±0.6 | 92.7±1.3 | 92.9±0.8 | 82.0±1.6 | 84.3±0.2 | **88.5** |
| PU-SCL | PUPL (CE) | 85.9±1.6 | 84.8±2.4 | 92.4±0.9 | 93.4±1.2 | 83.1±2.9 | **85.5±0.6** | **87.5** |
| PUCL | PUPL (CE) | **92.0±0.7** | **89.6±1.2** | **93.5±0.8** | **93.8±0.4** | **85.0±0.9** | 85.2±2.1 | **89.9** |

Table 1: **Effectiveness of PUPL.** To demonstrate the efficacy of PUPL , we train a downstream linear classifier using PUPL(CE) and NNPU ( run with $\pi_p^*$ ). over embeddings obtained via different contrastive objectives - SSCL, SCL-PU and PUPL.

Lloyd's algorithm (Lloyd, 1982) is the de-facto approach for locally solving (7) in an unsupervised fashion. However, since we have some label positive examples, instead of initializing the centers randomly, we initialize $\mu_P$ to be the centroid of the representations of the labeled positive samples; whereas, $\mu_N$ is initialized via randomized $k$-means++ seeding strategy. The algorithm then performs usual alternating $k$-means updates. The unlabeled samples can then be pseudo-labeled based on the nearest cluster center as follows:

$$\mathbf{z}_i \in \mathcal{Z}_U : \tilde{y}_i = \begin{cases} 1 & \text{if } \mu_P = \arg\min_{\mu \in C} \|\mathbf{z}_i - \mu\|^2 \\ 0 & \text{o/w} \end{cases}$$

This immediately allow us to learn a linear decision boundary via training $v_\mathbf{v}(\cdot)$ over $\{(\mathbf{z}_i, \tilde{y}_i) : \mathbf{z}_i \in \mathcal{Z}_{PU}\}$ using any standard classification loss such as CE. Algorithm 1(B) describes PUPL in detail.

If the PU data is generated as (1), then we can prove that PUPL enjoys improved guarantees over standard $k$-means and $k$-means++ (Yoder & Priebe, 2017; Liu et al., 2010):

**Theorem 3.** *Suppose, PU data is generated as* (1)*, then running Algorithm 1(B) on $\mathcal{Z}_{PU}$ yields:* $\mathbb{E}\big[\phi(\mathcal{Z}_{PU}, C_{PUPL})\big] \leq 16\phi^*(\mathcal{Z}_{PU}, C^*)$. *In comparison, running $k$-means++ on $\mathcal{Z}_{PU}$ we get,* $\mathbb{E}\big[\phi(\mathcal{Z}_{PU}, C_{k-\mathbf{means}++})\big] \leq 21.55\phi^*(\mathcal{Z}_{PU}, C^*)$.

| Algorithms | Datasets | | | | | | Average |
|---|---|---|---|---|---|---|---|
| | **F-MNIST-I** $(\pi_p^* = 0.3)$ | **F-MNIST-II** $(\pi_p^* = 0.7)$ | **CIFAR-I** $(\pi_p^* = 0.4)$ | **CIFAR-II** $(\pi_p^* = 0.6)$ | **STL-I** $(\hat{\pi}_p = 0.51)$ | **STL-II** $(\hat{\pi}_p = 0.49)$ | |
| | $n_P = 1000$ | | | | | | |
| UPU[†] | 71.3±1.4 | 84.0±4.0 | 76.5±2.5 | 71.6±1.4 | 76.7±3.8 | 78.2±4.1 | 76.4 |
| NNPU[†] | 89.7±0.8 | 88.8±0.9 | 84.7±2.4 | 83.7±0.6 | 77.1±4.5 | 80.4±2.7 | 84.1 |
| NNPU[†] W MIXUP | 91.4±0.3 | 88.2±0.7 | 87.2±0.6 | 85.8±1.2 | 79.8±0.8 | 82.2±0.9 | 85.8 |
| SELF-PU[†] | 90.8±0.4 | 89.1±0.7 | 85.1±0.8 | 83.9±2.6 | 78.5±1.1 | 80.8±2.1 | 84.7 |
| PAN | 88.7±1.2 | 83.6±2.5 | 87.0±0.3 | 82.8±1.0 | 77.7±2.5 | 79.8±1.4 | 83.3 |
| vPU[†] | 90.6±1.2 | 86.8±0.8 | 86.8±1.2 | 82.5±1.1 | 78.4±1.1 | 82.9±0.7 | 84.7 |
| MIXPUL | 87.5±1.5 | 89.0±0.5 | 87.0±1.9 | 87.0±1.1 | 77.8±0.7 | 78.9±1.9 | 84.5 |
| PULNS | 90.7±0.5 | 87.9±0.5 | 87.2±0.6 | 83.7±2.9 | 80.2±0.8 | 83.6±0.7 | 85.6 |
| P³MIX-E | 91.9±0.3 | **89.5±0.5** | 88.2±0.4 | 84.7±0.5 | 80.2±0.9 | 83.7±0.7 | 86.4 |
| P³MIX-C | **92.0±0.4** | 89.4±0.3 | 88.7±0.4 | 87.9±0.5 | 80.7±0.7 | 84.1±0.3 | 87.1 |
| PUCL + PUPL | 91.8±0.8 | 89.2±0.3 | **92.3±1.9** | **91.2±0.5** | **83.8±1.4** | **84.5±0.7** | **88.8** |
| | $n_P = 3000$ | | | | $n_P = 2500$ | | |
| UPU[†] | 89.9±1.0 | 78.6±1.3 | 80.6±2.1 | 72.9±3.2 | 70.3±2.0 | 74.0±3.0 | 77.7 |
| NNPU[†] | 90.8±0.6 | 90.5±0.4 | 85.6±2.3 | 85.5±2.0 | 78.3±1.2 | 82.2±0.5 | 85.5 |
| RP | 92.2±0.4 | 75.9±0.6 | 86.7±2.9 | 77.8±2.5 | 67.8±4.6 | 68.5±5.7 | 78.2 |
| vPU[†] | **92.7±0.3** | **90.8±0.6** | 89.5±0.1 | 88.8±0.8 | 79.7±1.5 | 83.7±0.1 | 87.5 |
| PUCL + PUPL | 92.0±0.7 | 89.6±1.2 | **93.5±0.8** | **93.8±0.4** | **85.0±0.9** | **85.2±2.1** | **89.9** |

Table 2: **PU Learning Benchmarks.** We compare our approach against several PU Learning baselines algorithms over different datasets and different amount of labeled data. Our setup is identical as (Li et al., 2022; Chen et al., 2020a). †: These methods were run with oracle class prior knowledge.

This result indicates that, whenever the feature space exhibits clustering properties, i.e. positive and negative examples form separate clusters; and the labeled positives are drawn i.i.d from the true positive marginal, then Theorem 3 suggest that PUPL is able to recover the true underlying labels even in low-supervision regime. Further, due to the clever initialization of the cluster centroids, PUPL enjoys improved guarantees over $k$-means++.

## 3 EMPIRICAL EVIDENCE

**Experimental Setup.** Closely following the experimental setup of (Li et al., 2022; Chen et al., 2020a), we conduct our experiments on six benchmark datasets: STL-I, STL-II, CIFAR-I, CIFAR-II, FMNIST-I, and FMNIST-II, obtained via modifying STL-10 (Coates et al., 2011), CIFAR-10 (Krizhevsky et al., 2009) and Fashion MNIST (Xiao et al., 2017) respectively. Additionally, we perform ablations on a subset of dog vs non-dog images sampled from ImageNet-subset (Hua et al., 2021; Engstrom et al., 2019), CIFAR-10(cat vs dogs) and CIFAR-10(vehile vs animal). We use LeNet-5 (LeCun et al., 1998) for F-MNIST and 7-layer CNN for STL and CIFAR benchmarks (Li et al., 2022; Chen et al., 2020a). Dog vs Non Dog (ImageNet) experiments utilize ResNet-34 (He et al., 2016), Dog vs Cats (CIFAR-10) use ResNet-18. Details on experimental setup, hyper-parameters, baselines are presented in Appendix A.7.

- **Comparison with other PU baselines:** To demonstrate the efficacy of our proposed approach, we compare it with several popular PU Learning baselines. The details of the baselines can be found in Appendix A.7. As discussed before, several of the baselines rely the knowledge of class prior $\pi_p$. For CIFAR-I, CIFAR-II, FMNIST-I, and FMNIST-II, oracle $\pi_p^*$ is known exactly, with values 0.4, 0.6, 0.3, and 0.7, respectively. However, since STL dataset is naturally semi-supervised, $\pi_p^*$ is unknown and thus estimated using KM2 (Ramaswamy et al., 2016) - a popular mixture proportion estimation algorithm. The estimated class priors $\hat{\pi}_p$ for STL-I and STL-II are found to be 0.51 and 0.49 (Li et al., 2022). The empirical findings are summarized in 2 - where the baselines at $n_P = 1000$, are borrowed from (Li et al., 2022) and other reported baselines are obtained from (Chen et al., 2020a).

- **Ablations on Contrastive Representation Learning.** Theorem 1,3 indicate that the bias-variance trade-off depends on amount of positive labeled data as well as the true class prior and thus are important parameters to decide which contrastive loss to pick. Our ablation experiments Figure 4

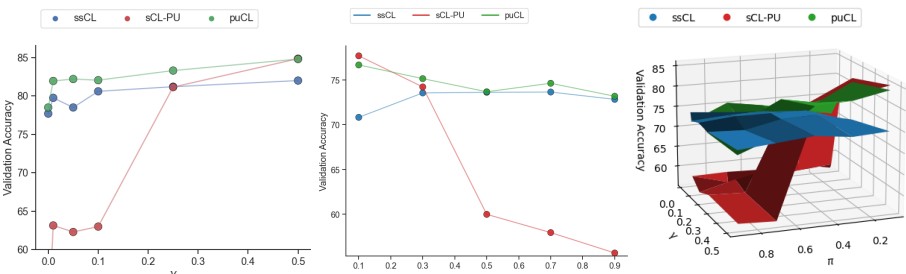

(a) Varying $\gamma$ with fixed $n_U, \pi$    (b) Varying $\pi$ with fixed $n_U, \gamma$  (c) Varying $\pi, \gamma$ with fixed $n_U$

Figure 4: **Ablation of Contrastive Representation Learning under different PU dataset settings:** To better understand the bias-variance trade-off in PU representation learning, we experiment with different PU learning settings: class prior $\pi_p$ and amount of labeled data captured by $\gamma = \frac{n_P}{n_U}$ ( Theorem 4 ). Experiments train ResNet-34 on ImageNet Dogs vs Non-Dogs. Embedding evaluation was performed using fully supervised kNN classification. They flesh out several interesting aspects of contrastive learning over PU data and supplement our theoretical findings. Please refer to Section 3 for a detailed discussion.

are aimed at understanding this bias-variance trade-off and gain insight about the behavior of contrastive objectives in the PU setting. Our observations are summarized as follows:

- **Role of $\gamma$ :** In these experiments, we fix the amount of unlabeled data $n_U$ and $\pi_p$, and vary the number of labeled positive examples $n_P$, resulting in different values of $\gamma = \frac{n_P}{n_U}$. We consistently observe that PUCL outperforms SSCL across all settings. The improvements are particularly significant when $\gamma$ is larger. While, for larger values of $\gamma$, SCL-PU shows accuracy gains, it suffers from performance degradation, especially for smaller values of $\gamma$ - aligning with Theorem 1

- **Role of $\pi_p$ :** The bias characterization of SCL-PU indicate a dependence on $\pi_p$. To understand this in isolation, we fix $\gamma$ and $n_U$ while using different $\pi_p$ to create the unlabeled set i.e. $\pi_p n_U$ positives and $(1 - \pi_p)n_U$ negatives are mixed. Indeed as $\pi_p$ approaches $1/2$ we observe that the SCL-PU loss starts to degrade and more interestingly, as $\pi_p \to 1$ SCL-PU completely collapses. This is possibly because a large value of $\pi_p$ implies that the training algorithm was presented with less amount of negative (unlabeled) examples. While the other two objectives remain fairly robust; gains of PUCL over SSCL by using available supervision is diminished.

- **Convergence**: We Find that PUCL not only has better generalization, it also enjoys uperior convergence compared to SSCL since it suffers a lower bias compared to the ideal fully supervised loss. Due to space constraint we discuss this in  Figure 9,  Theorem 4 in  Appendix A.5.

## 4   CONCLUSION, LIMITATIONS AND BROADER IMPACT

In summary, we present a novel, simple and practical PU learning solution with superior empirical performance, without needing additional knowledge like class prior. Our approach uniquely stays effective even with extremely limited labels, unlike prior PU methods. Overall, by pioneering PU learning with semantic similarity through contrastive learning and pseudo-labeling, we provide a theoretically-grounded technique that opens a valuable new research direction for PU learning.

One potential limitation of our method is that it depends on contrastive learning to find cluster-preserving embedding space and might fail when it can't do so. Further, it depends on contrasting augmentations, which might be challenging to extend in some domains e.g. time series.

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

CONTENTS

# A APPENDIX

## A.1 NOTATIONS AND ABBREVIATIONS

| | |
|---|---|
| SSCL | Self Supervised Contrastive Learning |
| sCL-PU | Naive PU adaptation of Supervised Contrastive Learning |
| PUCL | Positive Unlabeled Contrastive Learning |
| PUPL | Positive Unlabeled Pseudo Labeling |
| $a$ | A scalar (integer or real) |
| $\mathbf{a}$ | A vector |
| $\mathbf{A}$ | A matrix |
| $\mathrm{a}$ | A scalar random variable |
| $\mathbf{a}$ | A vector-valued random variable |
| $\mathbb{A}$ | A set |
| $\{0, 1\}$ | The set containing 0 and 1 |
| $\{0, 1, \ldots, n\}$ | The set of all integers between $0$ and $n$ |
| $[a, b]$ | The real interval including $a$ and $b$ |
| $\mathbb{A} \backslash \mathbb{B}$ | Set subtraction, i.e., the set containing the elements of $\mathbb{A}$ that are not in $\mathbb{B}$ |
| $\mathrm{a}_i$ | Element $i$ of the random vector $\mathbf{a}$ |
| $P(\mathrm{a})$ | A probability distribution over a discrete variable |
| $p(\mathrm{a})$ | A probability distribution over a continuous variable, or over a variable whose type has not been specified |
| $f : \mathbb{A} \to \mathbb{B}$ | The function $f$ with domain $\mathbb{A}$ and range $\mathbb{B}$ |
| $f \circ g$ | Composition of the functions $f$ and $g$ |
| $f(\mathbf{x}; \theta)$ | A function of $\mathbf{x}$ parametrized by $\theta$. (Sometimes we write $f(\mathbf{x})$ and omit the argument $\theta$ to lighten notation) |
| $\lVert \mathbf{x} \rVert_p$ | $L^p$ norm of $\mathbf{x}$ |
| $\mathbf{1}(condition)$ | is 1 if the condition is true, 0 otherwise |

### A.2 Extended Related Work

**Positive Unlabeled (PU) Learning :**    Existing PU learning algorithms primarily differ in the way they handle the semantic annotations of unlabeled examples.

One set of approaches rely on heuristic based *sample selection* where the idea is to identify potential negatives, positives or both samples in the unlabeled set; followed by performing traditional supervised learning using these pseudo-labeled instances in conjunction with available labeled positive data (Liu et al., 2002; Bekker & Davis, 2020; Luo et al., 2021; Wei et al., 2020).

A second set of approaches adopt a *re-weighting* strategy, where the unlabeled samples are treated as down-weighted negative examples (Liu et al., 2003; Lee & Liu, 2003). However, both of these approaches can be difficult to scale, as identifying reliable negatives or finding appropriate weights can be challenging or expensive to tune, especially in deep learning scenarios (Garg et al., 2021).

Recent works, further use the cost-sensitive classifiers in conjunction with other techniques.For example  (Chen et al., 2021) use self distillation to improve the initial nnPU model;  (Wei et al., 2020) uses mixup data augmentation to create augmented unlabeled samples with soft labels. Moreover, PU learning is also closely related to other robustness and weakly supervised settings, including learning under distribution shift (Garg et al., 2021), asymmetric label noise (Tanaka et al., 2021; Du & Cai, 2015) and semi-supervised learning (Chen et al., 2020c; Assran et al., 2020; Zhou, 2018).

**Contrastive Representation Learning:**    Self-supervised learning has demonstrated superior performances over supervised methods on various benchmarks. Joint-embedding methods (Chen et al., 2020b; Grill et al., 2020; Zbontar et al., 2021; Caron et al., 2021) are one the most promising approach for self-supervised representation learning where the embeddings are trained to be invariant to distortions. To prevent trivial solutions, a popular method is to apply pulsive force between embeddings from different images, known as contrastive learning. Contrastive loss is shown to be useful in various domains, including natural language processing (Gao et al., 2021), multimodal learning (Radford et al., 2021). Contrastive loss can also benefit supervised learning (Khosla et al., 2020).

**Clustering based Pseudo Labeling:**    Our approach is also closely related simultaneous clustering and pseudo-labeling approaches like DeepCluster (Caron et al., 2020) **Clustering Based Representation Learning:**  Simultaneous clustering and representation learning has gained popularity recently. **DeepCluster** (Caron et al., 2018) uses off-the-shelf clustering method e.g. kMeans to assign pseudo labels based on cluster membership and subsequently learns the representation using standard CE loss over the pseudo-labels. However, this standard simultaneous clustering and representation learning framework is often susceptible to degenerate solutions (e.g. trivially assigning all the samples to a single label) even for linear models (Xu et al., 2004; Joulin & Bach, 2012; Bach & Harchaoui, 2007). **SeLA** (Asano et al., 2019) alleviate this by adding the constraint that the label assignments must partition the data in equally-sized subsets. **Twin contrastive clustering (TCC)**  (Shen et al., 2021), **SCAN** (Van Gansbeke et al., 2020),  (Qian, 2023), **SwAV** (Caron et al., 2020; Bošnjak et al., 2023) combines ideas from contrastive learning and clustering based representation learning methods to perform simultaneous clusters the data while enforcing consistency between cluster assignments produced for different augmentations of the same image in an online fashion.

### A.3 BACKGROUND

#### A.3.1 DISCUSSION ON DIFFERENT PU LEARNING PROBLEM SETTINGS

**Case Control Setting:** Recall, the PU setting we have studied in the paper. Let $x \in \mathbb{R}^d$ and $y \in Y = \{0, 1\}$ be the underlying input (i.e., feature) and output (label) random variables respectively and let $p(x, y)$ denote the true underlying joint density of $(x, y)$. Then, a PU training dataset is composed of a set $\mathcal{X}_P$ of $n_P$ positively labeled samples and a set $\mathcal{X}_U$ of $n_U$ unlabeled samples (a mixture of both positives and negatives) i.e.

$$\mathcal{X}_{PU} = \mathcal{X}_P \cup \mathcal{X}_U, \ \mathcal{X}_P = \{\mathbf{x}_i^P\}_{i=1}^{n_P} \overset{i.i.d.}{\sim} p(x|y=1), \ \mathcal{X}_U = \{\mathbf{x}_i^U \overset{i.i.d.}{\sim} p(x)\}_{i=1}^{n_u} \tag{8}$$

This particular setup of how PU learning dataset is generated is referred to as the case-control setting (Bekker et al., 2019; Blanchard et al., 2010) and possibly widely used.

**Single Dataset Setting:** Now, we consider another setting referred to as the Single Dataset setting where there is only one dataset. The positive samples are randomly labeled from the dataset as opposed to being independent samples from the positive marginal. Thus the unlabeled set is no longer truly representative of the mixture. However our experiments **??** reveal that contrastive learning is still able learn representations following similar trends as case-control settings possibly because it is agnostic to how the data is generated unlike unbiased PU Learning methods.

#### A.3.2 CONNECTION TO LEARNING UNDER CLASS DEPENDENT LABEL NOISE

PU Learning is also closely related to the popular learning under label noise problem where the goal is to robustly train a classifier when a fraction of the training examples are mislabeled. This problem is extensively studied under both generative and discriminative settings and is an active area of research (Ghosh et al., 2015; 2017; Ghosh & Lan, 2021; Wang et al., 2019; Zhang et al., 2017).

Consider the following instance of *learning a binary classifier under class dependent label noise* i.e. the class conditioned probability of being mislabeled is $\xi_P$ and $\xi_N$ respectively for the positive and negative samples. Formally, let $\mathcal{X}_{PN}$ be the underlying clean binary dataset.

$$\mathcal{X}_{PN} = \mathcal{X}_P \cup \mathcal{X}_N, \ \mathcal{X}_P = \{\mathbf{x}_i^P\}_{i=1}^{n_P} \overset{i.i.d.}{\sim} p(x|y=+1), \ \mathcal{X}_N = \{\mathbf{x}_i^N \overset{i.i.d.}{\sim} p(x|y=-1)\}_{i=n_P+1}^{n_N} \tag{9}$$

Instead of $\mathcal{X}_{PN}$, a binary classifier needs to be trained from a noisy dataset $\tilde{\mathcal{X}}_{PN}$ with class dependent noise rates $\xi_P$ and $\xi_N$ i.e.

$$\tilde{\mathcal{X}}_{PN} = \{(\mathbf{x}_i, \tilde{y}_i)\}_{i=1}^{n_P+n_N}, \ \xi_P = p(\tilde{y}_i \neq y_i | y_i = +1), \ \xi_N = p(\tilde{y}_i \neq y_i | y_i = -1) \tag{10}$$

> **REDUCTION OF PU LEARNING 2.1 TO LEARNING WITH LABEL NOISE :** *Recall from Section 2.2 the naive disambiguation-free approach (Li et al., 2022), where the idea is to pseudo label the PU dataset as follows: Treat the unlabeled examples as negative and train an ordinary binary classifier over the pseudo labeled dataset. Clearly, since the unlabeled samples (a mixture of positives and negatives) are being pseudo labeled as negative, this is an instance of learning with class dependent label noise:*
>
> $$\tilde{\mathcal{X}}_{PN} = \mathcal{X}_P \cup \tilde{\mathcal{X}}_N, \ \mathcal{X}_P = \{\mathbf{x}_i^P\}_{i=1}^{n_P} \overset{i.i.d.}{\sim} p(x|y=1), \ \tilde{\mathcal{X}}_N = \{\mathbf{x}_i^U \overset{i.i.d.}{\sim} p(x)\}_{i=1}^{n_u} \tag{11}$$
>
> *It is easy to show that noise rates are:*
>
> $$E(\xi_P) = \frac{\pi_P}{\gamma + \pi_P} \ \text{and} \ \xi_N = 0 \tag{12}$$
>
> *Where $E(\gamma) = \frac{n_P}{n_U}$ and $\pi_P = p(y = 1|\mathbf{x})$ are training distribution dependent parameters.*

Under the standard robust Empirical Risk Minimization (ERM) framework, the goal is to robustly estimate the true risk i.e. for some loss we want the estimated risk (from the noisy data) to be close to the true risk (from the clean data) i.e. with high probability:

$$\Delta = \left\| \hat{\mathcal{R}}(\boldsymbol{\theta}) - \mathcal{R}(\boldsymbol{\theta}^*) \right\|_2 = \mathbb{E} \left\| \ell\left(f_{\boldsymbol{\theta}}(\mathbf{x}), \tilde{y}\right) - \ell\left(f_{\boldsymbol{\theta}^*}(\mathbf{x}), y\right) \right\|_2 \leq \epsilon$$

A popular way to measure the resilience of an estimator against corruption is via breakdown point analysis (Donoho & Huber, 1983; Huber, 1996; Lopuhaa et al., 1991; Acharya et al., 2022).

**Definition 1** (**Breakdown point**). *Breakdown point $\psi$ of an estimator is simply defined as the smallest fraction of corruption that must be introduced to cause an estimator to break implying $\Delta$ (risk estimation error) can become unbounded i.e. the estimator can produce arbitrarily wrong estimates.*

It is easy to show the following result:

---

**Lemma 1.** *Consider the problem of learning a binary classifier ($P$ vs $N$) in presence of class-dependent label noise with noise rates $E(\xi_P) = \frac{\pi_P}{\gamma + \pi_P}$, $\xi_N = 0$. Without additional distributional assumption, no robust estimator can guarantee bounded risk estimate $\left\| \hat{\mathcal{R}}(\boldsymbol{\theta}) - \mathcal{R}(\boldsymbol{\theta}^*) \right\|_2 \le \epsilon$ if*

$$\gamma \le 2\pi_p - 1$$

*where $\gamma = \frac{n_P}{n_U}$ and $\pi_p = p(y = 1 | \mathbf{x})$ denotes the underlying class prior.*

---

*Proof.* This result follows from using the fact that for any estimator $0 \le \psi < \frac{1}{2}$ (Lopuhaa et al., 1991; Minsker et al., 2015; Cohen et al., 2016; Acharya et al., 2022) i.e. for robust estimation to be possible, the corruption fraction $\alpha = \frac{\pi_P}{\gamma + 1} < \frac{1}{2}$. ∎

This result suggests that PU Learning cannot be solved by off-the-shelf label noise robust algorithms and specialized algorithms need to be designed.

### A.3.3 COST SENSITIVE PU LEARNING

Consider training linear classifier $v_{\mathbf{v}}(\cdot) : \mathbb{R}^k \to \mathbb{R}^{|Y|}$ where $k \in \mathbb{R}^+$ is the dimension of the features. In the (fully) supervised setting (PN) labeled examples from both class marginals are available and the linear classifier can be trained using standard supervised classification loss e.g. CE. However, in the PU learning setup since no labeled negative examples are provided it is non-trivial to train.

---

As discussed in Appendix A.3.2, without additional assumptions the equivalent class dependent label noise learning problem cannot be solved when $\gamma \le 2\pi_p - 1$. However, note that in PU Learning, we additionally know that a subset of the dataset is correctly labeled i.e.

$$p(\tilde{y}_i = y_i = 1 | \mathbf{x}_i \in \mathbb{P}) = 1$$

*Can we use this additional information to enable PU Learning even when $\gamma <= 2\pi_p - 1$ ?*

---

Remarkably, SOTA cost-sensitive PU learning algorithms tackle this by forming an unbiased estimate of the true risk from PU data (Blanchard et al., 2010) by assuming additional knowledge of the true class prior $\pi_p = p(y = 1 | \mathbf{x})$. The unbiased estimator dubbed uPU (Blanchard et al., 2010; Du Plessis et al., 2014) of the true risk $R_{PN}(v)$ from PU data is given as:

$$\hat{R}_{pu}(v) = \pi_p \hat{R}_p^+(v) + \left[ \hat{R}_u^-(v) - \pi_p \hat{R}_p^-(v) \right]$$

where we denote the empirical estimates computed over PU dataset (1) as:

$$\hat{R}_p^+(v) = \frac{1}{n_P} \sum_{i=1}^{n_P} \ell(v(\mathbf{x}_i^P), 1) , \quad \hat{R}_p^-(v) = \frac{1}{n_P} \sum_{i=1}^{n_P} \ell(v(\mathbf{x}_i^P), 0) , \quad \hat{R}_u^-(v) = \frac{1}{n_U} \sum_{i=1}^{n_U} \ell(v(\mathbf{x}_i^U), 0)$$

and $\ell(\cdot, \cdot) : Y \times Y \to \mathbb{R}$ is the classification loss e.g. CE.

In practice, clipping the estimated negative risk results in a further improvement (Kiryo et al., 2017).

$$\hat{R}_{pu}(v) = \pi_p \hat{R}_p^+(v) + \max\left\{ 0, \ \hat{R}_u^-(v) - \pi_p \hat{R}_p^-(v) \right\} \tag{13}$$

This clipped loss dubbed NNPU is the de-facto approach to solve PU problems in practical settings and we use this as a powerful baseline for training the downstream PU classifier.

As discussed before, we identify two main issues related to these cost-sensitive estimators:

- **Class Prior Estimate :** The success of these estimators hinges upon the knowledge of the oracle class prior $\pi_p^*$ for their success. It is immediate to see that an error is class prior estimate $\|\hat{\pi}_p - \pi_p^*\|_2 \leq \xi$ results in an estimation bias $\sim \mathcal{O}(\xi)$ that can result in poor generalization, slower convergence or both.

  Our experiments (Figure 15) suggest that even small approximation error in estimating the class prior can lead to notable degradation in the overall performance of the estimators.

  Unfortunately however in practical settings (e.g. large-scale recommendation) the class prior is not available and often estimating it with high accuracy using some MPE (Garg et al., 2021; Ivanov, 2020; Ramaswamy et al., 2016) algorithm can be quite costly.

- **Low Supervision Regime :** While these estimators are significantly more robust than the vanilla supervised approach, our experiments ( Figure 14) suggest that they might produce decision boundaries that are not closely aligned with the true decision boundary especially as $\gamma$ becomes smaller (Kiryo et al., 2017; Du Plessis et al., 2014). Note that, when available supervision is limited i.e. when $\gamma$ is small, the estimates $\hat{R}_p^+$ and $\hat{R}_p^-$ suffer from increased variance resulting in increase variance of the overall estimator $\sim \mathcal{O}(\frac{1}{n_p})$. For sufficiently small $\gamma$ these estimators are likely result in poor performance due to large variance.

## A.4 Full Algorithm: Parameter Free Contrastive PU Learning

---

**Algorithm 1** Contrastive Positive Unlabeled Learning

---

**initialize:** PU training data $\mathcal{X}_{\text{PU}}$; batch size b; temperature parameter $\tau > 0$; randomly initialized encoder $g_{\mathbf{B}}(\cdot) : \mathbb{R}^d \to \mathbb{R}^k$, projection network: $h_{\mathbf{\Gamma}}(\cdot) : \mathbb{R}^k \to \mathbb{R}^p$, and linear classifier $v_{\mathbf{v}}(\cdot) : \mathbb{R}^k \to \mathbb{R}^{|Y|}$; family of stochastic augmentations $\mathcal{T}$.

**A. PUCL : Positive Unlabeled Contrastive Representation Learning**
**for** *epochs e = 1, 2, ..., until convergence* **do**
    *select mini-batch*: $\mathcal{D} = \{\mathbf{x}_i\}_{i=1}^b \sim \mathcal{X}_{\text{PU}}$ and *sample augmentations*: $t(\cdot) \sim \mathcal{T}, t'(\cdot) \sim \mathcal{T}$
    *create multi-viewed batch*: $\tilde{\mathcal{D}} = \{\tilde{\mathbf{x}}_i = t(\mathbf{x}_i), \tilde{\mathbf{x}}_{a(i)} = t'(\mathbf{x}_i)\}_{i=1}^b$
    $\mathbb{I} = \{1, 2, \ldots, 2b\}$ is the index set of $\tilde{\mathcal{D}}$ and $\mathbb{P} = \{i \in \mathbb{I} : \mathbf{x}_i \in \mathcal{X}_{\text{P}}\}, \mathbb{U} = \{j \in \mathbb{I} : \mathbf{x}_j \in \mathcal{X}_{\text{U}}\}$
    *obtain representations*: $\{\mathbf{z}_j\}_{j \in \mathbb{I}} = \{\mathbf{z}_i = h_{\mathbf{\Gamma}} \circ g_{\mathbf{B}}(\tilde{\mathbf{x}}_i), \mathbf{z}_{a(i)} = h_{\mathbf{\Gamma}} \circ g_{\mathbf{B}}(\tilde{\mathbf{x}}_{a(i)})\}_{i=1}^b$
    *compute pairwise similarity*: $\mathbf{z}_i \cdot \mathbf{z}_j = \frac{1}{\tau} \frac{\mathbf{z}_i^T \mathbf{z}_j}{\|\mathbf{z}_i\|\|\mathbf{z}_j\|}$, $P_{i,j} = \frac{\exp(\mathbf{z}_i \cdot \mathbf{z}_j)}{\sum_{k \in \mathbb{I}} \mathbf{1}(k \neq i) \exp(\mathbf{z}_i \cdot \mathbf{z}_k)}, \forall i, j \in \mathbb{I}$

    *compute loss* : $\mathcal{L}_{\text{PUCL}} = -\frac{1}{|\mathbb{I}|} \sum_{i \in \mathbb{I}} \left[ \mathbf{1}(i \in \mathbb{P}) \frac{1}{|\mathbb{P} \backslash i|} \sum_{j \in \mathbb{P}} \mathbf{1}(j \neq i) \log P_{i,j} + \mathbf{1}(i \in \mathbb{U}) \log P_{i,a(i)} \right]$

    *update network parameters* $\mathbf{B}, \mathbf{\Gamma}$ to minimize $\mathcal{L}_{\text{PUCL}}$
**end**
**return:** encoder $g_{\mathbf{B}}(\cdot)$ and throw away $h_{\mathbf{\Gamma}}(\cdot)$.

**B. PUPL: Positive Unlabeled Pseudo Labeling**
*obtain representations*: $\mathcal{Z}_{\text{P}} = \{\mathbf{r}_i = g_{\mathbf{B}}(\mathbf{x}_i) : \forall \mathbf{x}_i \in \mathcal{X}_{\text{P}}\}, \mathcal{Z}_{\text{U}} = \{\mathbf{r}_j = g_{\mathbf{B}}(\mathbf{x}_j) : \forall \mathbf{x}_j \in \mathcal{X}_{\text{U}}\}$
*initialize pseudo labels* : $\tilde{y}_i = y_i = 1 : \forall \mathbf{r}_i \in \mathcal{Z}_{\text{P}}$ and $\tilde{y}_j = 0 : \forall \mathbf{r}_j \in \mathcal{Z}_{\text{U}}$

*initialize cluster centers*: $\mu_{\text{P}} = \frac{1}{|\mathcal{Z}_{\text{P}}|} \sum_{\mathbf{r}_i \in \mathcal{Z}_{\text{P}}} \mathbf{r}_i$ , $\mu_{\text{N}} \overset{D(\mathbf{x}')}{\sim} \mathcal{Z}_{\text{U}}$ where $D(\mathbf{x}') = \frac{\|\mathbf{x}' - \mu_{\text{P}}\|^2}{\sum_{\mathbf{x}} \|\mathbf{x} - \mu_{\text{P}}\|^2}$

**while** *not converged* **do**
    *pseudo-label*: $\forall \mathbf{r}_i \in \mathcal{Z}_{\text{U}} : \tilde{y}_i = 1$ if $\mu_{\text{P}} = \arg\min_{\mu \in \{\mu_{\text{P}}, \mu_{\text{N}}\}} \|\mathbf{r}_i - \mu\|^2$ else $\tilde{y}_i = 0$
    $\tilde{\mathcal{Z}}_{\text{P}} = \mathcal{Z}_{\text{P}} \cup \{\mathbf{r}_i \in \mathcal{Z}_{\text{U}} : \tilde{y}_i = 1\}$ , $\tilde{\mathcal{Z}}_{\text{N}} = \{\mathbf{z}_i \in \mathcal{Z}_{\text{U}} : \tilde{y}_i = 0\}$
    *update cluster centers*: $\mu_{\text{P}} = \frac{1}{|\tilde{\mathcal{Z}}_{\text{P}}|} \sum_{\mathbf{z}_i \in \tilde{\mathcal{Z}}_{\text{P}}} \mathbf{z}_i$ , $\mu_{\text{N}} = \frac{1}{|\tilde{\mathcal{Z}}_{\text{N}}|} \sum_{\mathbf{z}_i \in \tilde{\mathcal{Z}}_{\text{N}}} \mathbf{z}_i$
**end**
**return:** $\tilde{\mathcal{X}}_{\text{PU}} = \{(\mathbf{x}_i, \tilde{y}_i) : \forall \mathbf{x}_i \in \mathcal{X}_{\text{PU}}\}$

**C. Train Binary Classifier**
update network parameters $\mathbf{v}$ to minimize cross-entropy loss $\mathcal{L}_{\text{CE}}(\mathbf{v}^T g_{\mathbf{B}}(\mathbf{x}_i), \tilde{y}_i)$
**return:** Positive Unlabeled classifier : $f_{\mathbf{v}, \mathbf{B}} = v_{\mathbf{v}} \circ g_{\mathbf{B}}(\cdot)$

---

## A.5 puCL: Positive Unlabeled Representation Learning

**Summary:** One way to obtain a representation manifold where the embeddings (features) exhibit linear separability is via contrastive learning (Parulekar et al., 2023).

- However, standard self-supervised contrastive loss sSCL (4) is unable to leverage the available supervision in the form of labeled positives.

- (Theorem 1) On the other hand, naive adaptation of the supervised contrastive loss sCL-PU (5) suffers from statistical bias in the PU setting that can result in significantly poor representations especially in the low supervision regime i.e. when only a handful labeled positive examples are available.

- (Theorem 2) To this end, the proposed objective PUCL leverages the available supervision judiciously to form an unbiased risk estimator of the ideal objective. Further, we show that it is provably more efficient than the self-supervised counterpart.

### A.5.1 Generalization Benefits of Incorporating Additional Positives

As previously discussed, the main observation we make is that judiciously incorporating available PU supervision is crucial for the success of contrastive learning over PU Learning. The unsupervised sSCL (4) objective is completely agnostic of the labels, resulting in representation that are while robust to noisy label, has poor generalization performance on downstream PU classification. On the other hand, sCL-PU: naive adaptation of sCL while performs well in low (high) noise (supervision) settings, it suffers from major performance degradation in the high (low) noise (supervision) regime. PUCL interpolates nicely between the robustness and generalization trade-off by judiciously incorporating the labeled positive to form an unbiased version of sCL. In Figure 5, we present a few more results affirming this observation over multiple datasets, encompassing both Single Dataset and Case-Control PU learning settings.

### A.5.2 Grouping semantically different objects together :

An important underlying assumption unsupervised learning is that the features contains information about the underlying label. Indeed, if $p(x)$ has no information about $p(y|x)$, no unsupervised representation learning method e.g. sSCL can hope to learn cluster-preserving representations.

However, in **fully supervised setting**, since semantic annotations are available, it is possible to find a representation space where semantically dissimilar objects are grouped together based on labels i.e. clustered based on semantic annotations via supervised objectives e.g. CE. While, it is important to note that such models would be prone to over-fitting and might generalize poorly to unseen data. Since supervised contrastive learning objectives sCL (14) (Khosla et al., 2020) use semantic annotations to guide the contrastive training, it can also be effective in such scenarios.

In particular, in sCL, in addition to self-augmentations, each anchor is attracted to all the other augmentations in the batch that share the same class label. For a fully supervised binary setting it takes the following form:

$$\mathcal{L}_{\text{sCL}} = -\frac{1}{|\mathbb{I}|} \sum_{i \in \mathbb{I}} \left[ \mathbf{1}(i \in \mathbb{P}) \frac{1}{|\mathbb{P} \setminus i|} \sum_{j \in \mathbb{P} \setminus i} \mathbf{z}_i \cdot \mathbf{z}_j + \mathbf{1}(i \in \mathbb{N}) \frac{1}{|\mathbb{N} \setminus i|} \sum_{j \in \mathbb{N} \setminus i} \mathbf{z}_i \cdot \mathbf{z}_j - \log Z(\mathbf{z}_i) \right] \quad (14)$$

where $\mathbb{P}$ and $\mathbb{N}$ denote the subset of indices in the augmented batch $\tilde{\mathcal{D}}$ that are labeled positive and negative respectively i.e. $\mathbb{P} = \{i \in \mathbb{I} : \mathbf{y}_i = 1\}$, $\mathbb{N} = \{i \in \mathbb{I} : \mathbf{y}_i = 0\}$. Clearly, $\mathcal{L}_{\text{sCL}}$ (14) is a **consistent estimator** of the ideal objective $\mathcal{L}^*_{\text{CL}}$ (3). Since the expected similarity of positive pairs is computed over all the available samples from the same class marginal as anchor, this loss enjoys a lower variance compared to its self-supervised counterpart $\mathcal{L}_{\text{sSCL}}$ (4).

In the **PU learning setting**, PUCL (6) behaves in a similar way. It incorporates both semantic similarity (via pulling self augmentations together) and semantic annotation (via pulling together labeled positives together). Intuitively, by interpolating between supervised and unsupervised contrastive objectives, PUCL favors representations where both semantically similar (feature) examples are grouped together along with all the labeled positives (annotations) are grouped together.

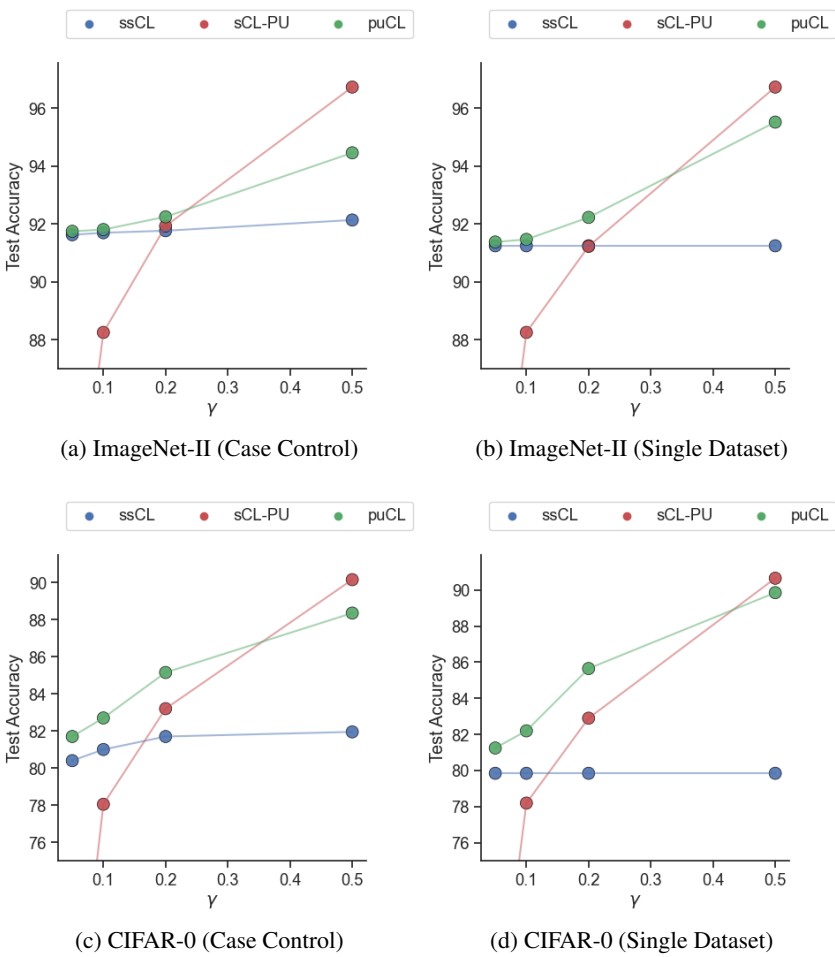

Figure 5: **Generalization with varying supervision:** In this experiment we train a ResNet-18 on CIFAR-0 (Subset of Dogs and Cats) and ImageNet-II (ImageWoof vs ImageNette) under both case-control and single-dataset PU Learning setting. For case control setting, number of unlabeled samples $n_U$ is kept fixed while we vary the number of labeled positives $n_P$. On the other hand for the Single Dataset setting we keep the total number of samples fixed $N = n_P + n_U$ while varying $n_P$. In both settings, we find PUCL to remain quite robust across different levels of supervision while consistently outperforming its unsupervised counterpart SSCL and being competitive with SCL-PU even in high supervision regimes. While., SCL-PU suffers from large degradation especially in the low-supervision regime.

ARRANGING POINTS ON UNIT HYPERCUBE:

To further understand the behavior of interpolating between semantic annotation (labels) and semantic similarity (feature) - Consider 1D feature space $\mathbf{x} \in \mathbb{R}$, e.g., $\mathbf{x}_i = 1$ if shape: triangle ($\blacktriangle$, $\blacktriangle$), $\mathbf{x}_i = 0$ if shape: circle ($\bullet$, $\bullet$). However, the labels are $y_i = 1$ if color: blue ($\blacktriangle$,$\bullet$) and $y_i = 1$ if color: red ($\blacktriangle$, $\bullet$) i.e $p(\mathbf{x})$ contains no information about $p(y|\mathbf{x})$. Figure 7 shows several representative configurations (note that, other configurations are similar) of arranging these points on the vertices of unit hypercube $\mathcal{H} \in \mathbb{R}^2$ when $\blacktriangle$ is fixed at $(0, 1)$.

- **Unsupervised objectives** e.g. SSCL (4) only rely on semantic similarity (feature) to learn embeddings, implying they attain minimum loss configuration when semantically similar objects $\mathbf{x}_i = \mathbf{x}_j$ are placed close to each other (neighboring vertices on $\mathcal{H}^2$) since this minimizes the inner product between representations of similar examples( Figure 7(a) ).

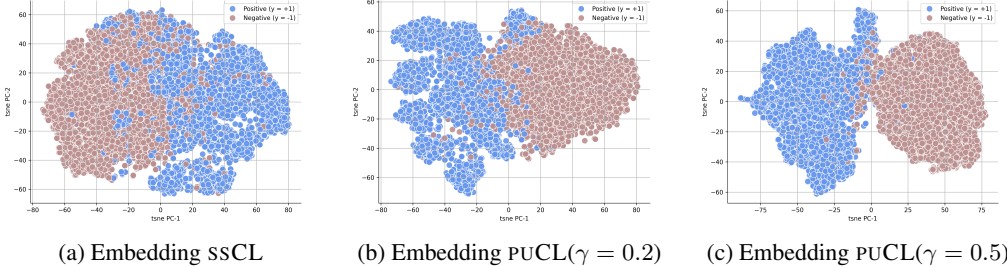

| (a) Embedding SSCL | (b) Embedding PUCL($\gamma = 0.2$) | (c) Embedding PUCL($\gamma = 0.5$) |

Figure 6: **Embedding Quality with varying supervision:** In this experiment we train a ResNet-18 on **ImageNet-II: ImageWoof vs ImageNette** - two subsets of ImageNet-1k widely used in noisy label learning research https://github.com/fastai/imagenette. Amount of supervision is measured with the ratio of labeled to unlabeled data $\gamma = \frac{n_P}{n_U}$. We keep the total number of samples $N = n_P + n_U$ fixed, while varying $n_P$. We observe that the embeddings obtained via PUCL exhibit significantly improved separability than that of the unsupervised baseline SSCL especially with increasing supervision.

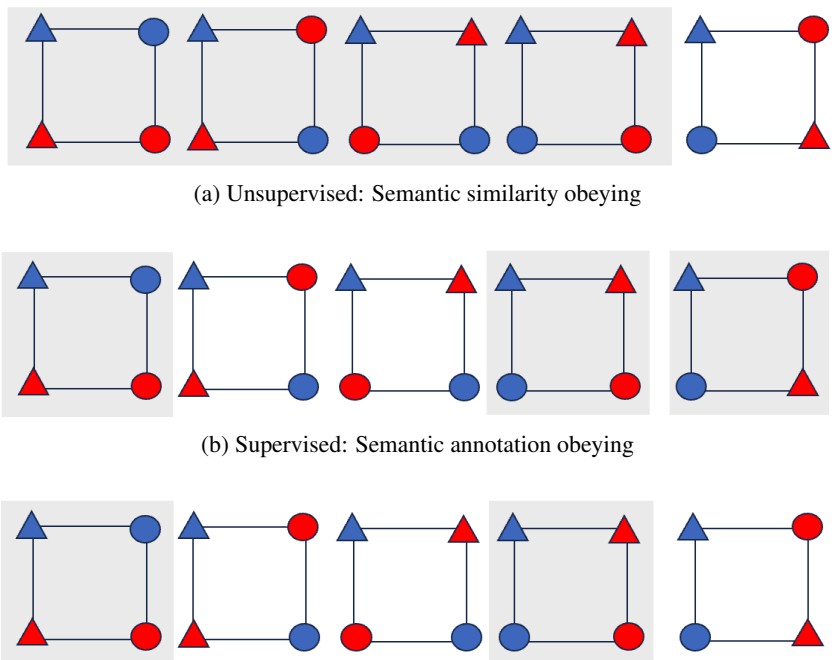

(a) Unsupervised: Semantic similarity obeying

(b) Supervised: Semantic annotation obeying

(c) Contrastive (PU)-Supervised: Semantic similarity and annotation obeying

Figure 7: **Geometric intuition of incorporating supervision:** Consider 1D feature space $\mathbf{x} \in \mathbb{R}$, e.g., $\mathbf{x}_i = 1$ if shape: triangle ($\blacktriangle$, $\blacktriangle$), $\mathbf{x}_i = 0$ if shape: circle ($\bullet$, $\bullet$). However, the labels are $y_i = 1$ if color: blue ($\blacktriangle$,$\bullet$) and $y_i = 1$ if color: red ($\blacktriangle$, $\bullet$). We show possible configurations (other configurations are similar) of arranging these points on the vertices of unit hypercube $\mathcal{H} \in \mathbb{R}^2$ when $\blacktriangle$ is fixed at $(0, 1)$. (a) Unsupervised objectives e.g. SSCL only rely on semantic similarity (feature) to learn embeddings, implying they attain minimum loss configuration when semantically similar objects are places close to each other (neighboring vertices on $\mathcal{H}^2$). (b) Supervised objectives on the other All the four shaded point configurations are favored by SSCL (4), since $\mathbf{x}_i = \mathbf{x}_j$ are placed neighboring vertices. However, the minimum loss configurations of PUCL (marked in rectangle) additionally also preserves annotation consistency.

- **Supervised objectives** e.g. CE on the other hand, updates the parameters such that the logits match the label. Thus purely supervised objectives attain minimum loss when objects sharing same annotation are placed next to each other ( Figure 7(b) ).

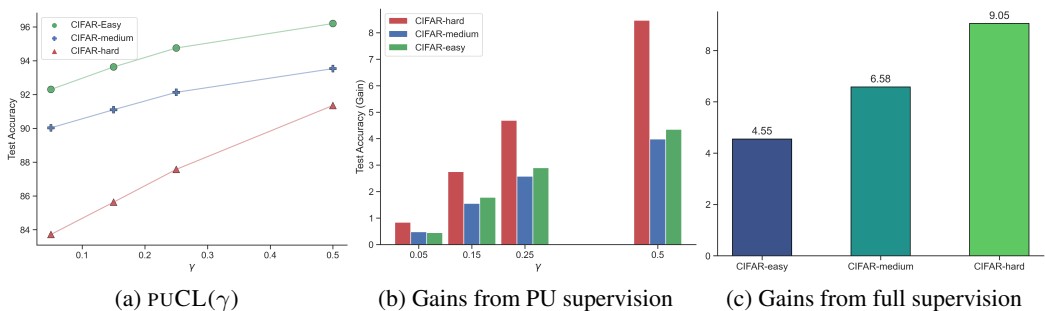

(a) PUCL($\gamma$)  (b) Gains from PU supervision  (c) Gains from full supervision

Figure 8: **Grouping dissimilar objectes together :** In this experiment we train a ResNet-18 on three CIFAR subsets carefully crafted to understand this phenomenon. In particular, we use CIFAR-hard (airplane, cat) vs (bird, dog), CIFAR-easy (airplane, bird) vs (cat, dog) and CIFAR-medium (airplane, cat, dog) vs bird. Note that, airplane and bird are semantically similar, also dog-cat are semantically closer to each other. We repeat the experiments across different supervision levels - amount of supervision is measured with $\gamma = \frac{n_P}{n_U}$. We keep the total number of samples $N = n_P + n_U$ fixed, while varying $n_P$. Observe that, (a) shows generalization of PUCL across different $\gamma$. (b), (c) denote the performance gains of PUCL and fully supervised SCL over unsupervised SSCL. Clearly, in the hard setting, SSCL i.e. PUCL($\gamma = 0$), suffers from large performance degradation. However, given enough supervision signal PUCL is still able to learn representations that preserves class label obeying linear separability.

- On the other hand, PUCL interpolates between the supervised and unsupervised objective. Simply put, by incorporating additional positives aims at learning representations that preserve annotation consistency. Thus, the minimum loss configurations are attained at the intersection of the minimum point configurations of SSCL and fully supervised SCL ( Figure 7(c) )

**Experimental Evaluation:** To understand this phenomenon experimentally, we train a ResNet-18 on three CIFAR subsets carefully crafted to simulate this phenomenon. In particular, we use CIFAR-hard (airplane, cat) vs (bird, dog), CIFAR-easy (airplane, bird) vs (cat, dog) and CIFAR-medium (airplane, cat, dog) vs bird. Note that, airplane and bird are semantically similar, also dog-cat are semantically closer to each other. Our experimental findings are reported in Figure 8. In summary, we observe that while SSCL is completely blind to supervision signals; given enough labels – PUCL is able to leverage the available positives to group the samples labeled positive together. Since we are in binary setting, being able to cluster positives together automatically solves the downstream P vs N classification problem as well.

### A.5.3 Convergence Benefits of Incorporating Additional Positives

As discussed throughout the paper, the main trick to incorporate

Our experiments also reveal that, leveraging the available positives in the loss not only improves the generalization performance, it also improves the convergence of representation learning from PU data as demonstrated in Figure 9. We argue that this is due to reduced variance resulting from incorporating multiple labeled positive examples by PUPL. We begin with deriving the gradient expressions for SSCL and PUCL

**Theorem 4.** *The gradient of $\mathcal{L}_{\text{PUCL}}$ (6) has lower bias than that of $\mathcal{L}_{\text{SSCL}}$ (4) with respect to $\mathcal{L}^*$ (3).*

*Proof.* **Gradient derivation of SSCL** Recall that, the SSCL (4) takes the following form for any random sample from the multi-viewed batch indexed by $i \in \mathbb{I}$

$$\begin{aligned}
\ell_i &= -\log \frac{\exp\left(\mathbf{z}_i \cdot \mathbf{z}_{a(i)}/\tau\right)}{Z(\mathbf{z}_i)} \; ; \; \forall i \in \mathbb{I} \\
&= -\frac{\mathbf{z}_i \cdot \mathbf{z}_{a(i)}}{\tau} + \log Z(\mathbf{z}_i)
\end{aligned} \tag{15}$$

Recall that the partition function $Z(\mathbf{z}_i)$ is defined as : $Z(\mathbf{z}_i) = \sum_{j \in \mathbb{I}} \mathbf{1}(j \neq i) \exp(\mathbf{z}_i \cdot \mathbf{z}_j/\tau)$. Note that, $\mathbf{z}_i = g_{\mathbf{w}}(\mathbf{x}_i)$ where we have consumed both encoder and projection layer into $\mathbf{w}$, and thus by

chain rule we have,

$$\frac{\partial \ell_i}{\partial \mathbf{w}} = \frac{\partial \ell_i}{\partial \mathbf{z}_i} \cdot \frac{\partial \mathbf{z}_i}{\partial \mathbf{w}} \tag{16}$$

Since, the second term depends on the encoder and fixed across the losses, the first term is sufficient to compare the gradients resulting from different losses. Thus, taking the differential of (15) w.r.t representation $\mathbf{z}_i$ we get:

$$
\begin{aligned}
\frac{\partial \ell_i}{\partial \mathbf{z}_i} &= -\frac{1}{\tau}\left[\mathbf{z}_{a(i)} - \frac{\sum_{j\in\mathbb{I}\setminus\{i\}} \mathbf{z}_j \exp(\mathbf{z}_i\cdot\mathbf{z}_j/\tau)}{Z(\mathbf{z}_i)}\right] \\
&= -\frac{1}{\tau}\left[\mathbf{z}_{a(i)} - \frac{\mathbf{z}_{a(i)}\exp(\mathbf{z}_i\cdot\mathbf{z}_{a(i)}/\tau) + \sum_{j\in\mathbb{I}\setminus\{i,a(i)\}} \mathbf{z}_j \exp(\mathbf{z}_i\cdot\mathbf{z}_j/\tau)}{Z(\mathbf{z}_i)}\right] \\
&= -\frac{1}{\tau}\left[\mathbf{z}_{a(i)}\left(1 - \frac{\exp(\mathbf{z}_i\cdot\mathbf{z}_{a(i)}/\tau)}{Z(\mathbf{z}_i)}\right) - \sum_{j\in\mathbb{I}\setminus\{i,a(i)\}}\mathbf{z}_j\frac{\exp(\mathbf{z}_i\cdot\mathbf{z}_j/\tau)}{Z(\mathbf{z}_i)}\right] \\
&= -\frac{1}{\tau}\left[\mathbf{z}_{a(i)}\left(1 - \frac{\exp(\mathbf{z}_i\cdot\mathbf{z}_{a(i)}/\tau)}{Z(\mathbf{z}_i)}\right) - \sum_{j\in\mathbb{I}\setminus\{i,a(i)\}}\mathbf{z}_j\frac{\exp(\mathbf{z}_i\cdot\mathbf{z}_j/\tau)}{Z(\mathbf{z}_i)}\right] \\
&= -\frac{1}{\tau}\left[\mathbf{z}_{a(i)}\left(1 - P_{i,a(i)}\right) - \sum_{j\in\mathbb{I}\setminus\{i,a(i)\}}\mathbf{z}_j P_{i,j}\right]
\end{aligned}
\tag{17}
$$

Where, the functions $P_{i,j}$ are defined as:

$$P_{i,j} = \frac{\exp(\mathbf{z}_i\cdot\mathbf{z}_j/\tau)}{Z(\mathbf{z}_i)} \tag{18}$$

**Gradient derivation of PUCL** Recall that, given a randomly sampled mini-batch $\mathcal{D}$, PUCL (6) takes the following form for any sample $i\in\mathbb{I}$ where $\mathbb{I}$ is the corresponding multi-viewed batch. Let, $\mathbb{P}(i) = \mathbb{P}\setminus i$ i.e. all the other positive labeled examples in the batch w/o the anchor.

$$
\begin{aligned}
\ell_i &= -\frac{1}{|\mathbb{P}(i)|}\sum_{q\in\mathbb{P}(i)}\log\frac{\exp(\mathbf{z}_i\cdot\mathbf{z}_q/\tau)}{Z(\mathbf{z}_i)} \; ; \; \forall i\in\mathbb{I} \\
&= -\frac{1}{|\mathbb{P}(i)|}\sum_{q\in\mathbb{P}(i)}\left[\frac{\mathbf{z}_i\cdot\mathbf{z}_q}{\tau} - \log Z(\mathbf{z}_i))\right]
\end{aligned}
\tag{19}
$$

where $Z(\mathbf{z}_i)$ is defined as before. Then, we can compute the gradient w.r.t representation $\mathbf{z}_i$ as:

$$\frac{\partial \ell_i}{\partial \mathbf{z}_i} = -\frac{1}{|\mathbb{P}(i)|} \sum_{q \in \mathbb{P}(i)} \left[ \frac{\mathbf{z}_q}{\tau} - \frac{\partial Z(\mathbf{z}_i)}{Z(\mathbf{z}_i)} \right]$$

$$= -\frac{1}{\tau |\mathbb{P}(i)|} \sum_{q \in \mathbb{P}(i)} \left[ \mathbf{z}_q - \frac{\sum_{j \in \mathbb{I} \setminus \{i\}} \mathbf{z}_j \exp(\mathbf{z}_i \cdot \mathbf{z}_j / \tau)}{Z(\mathbf{z}_i)} \right]$$

$$= -\frac{1}{\tau |\mathbb{P}(i)|} \sum_{q \in \mathbb{P}(i)} \left[ \mathbf{z}_q - \sum_{q' \in \mathbb{P}(i)} \mathbf{z}_{q'} P_{i,q'} - \sum_{j \in \mathbb{U}(i)} \mathbf{z}_j P_{i,j} \right]$$

$$= -\frac{1}{\tau |\mathbb{P}(i)|} \left[ \sum_{q \in \mathbb{P}(i)} \mathbf{z}_q - \sum_{q \in \mathbb{P}(i)} \sum_{q' \in \mathbb{P}(i)} \mathbf{z}_{q'} P_{i,q'} - \sum_{q \in \mathbb{P}(i)} \sum_{j \in \mathbb{U}(i)} \mathbf{z}_j P_{i,j} \right] \quad (20)$$

$$= -\frac{1}{\tau |\mathbb{P}(i)|} \left[ \sum_{q \in \mathbb{P}(i)} \mathbf{z}_q - \sum_{q' \in \mathbb{P}(i)} |\mathbb{P}(i)| \mathbf{z}_{q'} P_{i,q'} - \sum_{j \in \mathbb{U}(i)} |\mathbb{P}(i)| \mathbf{z}_j P_{i,j} \right]$$

$$= -\frac{1}{\tau} \left[ \frac{1}{|\mathbb{P}(i)|} \sum_{q \in \mathbb{P}(i)} \mathbf{z}_q - \sum_{q \in \mathbb{P}(i)} \mathbf{z}_q P_{i,q} - \sum_{j \in \mathbb{U}(i)} \mathbf{z}_j P_{i,j} \right]$$

$$= -\frac{1}{\tau} \left[ \sum_{q \in \mathbb{P}(i)} \mathbf{z}_q \left( \frac{1}{|\mathbb{P}(i)|} - P_{i,q} \right) - \sum_{j \in \mathbb{U}(i)} \mathbf{z}_j P_{i,j} \right]$$

where we have defined $\mathbb{U}(i) = \mathbb{I} \setminus \{i, \mathbb{P}(i)\}$ i.e. $\mathbb{U}(i)$ is the set of all samples in the batch that are unlabeled.

In case of fully supervised setting we would similarly get:

$$\frac{\partial \ell_i}{\partial \mathbf{z}_i} = -\frac{1}{\tau} \left[ \sum_{q \in \mathbb{P}(i)} \mathbf{z}_q \left( \frac{1}{|\mathbb{P}(i)|} - P_{i,q} \right) - \sum_{j \in \mathbb{N}(i)} \mathbf{z}_j P_{i,j} \right] \quad (21)$$

Since, in the fully supervised setting $\mathbb{I} - \mathbb{P}(i) = \mathbb{N}(i)$. Thus, by comparing the last term of the three gradient expressions, it is clear that PUCL enjoys lower bias compared to SSCL with respect to fully supervised counterpart. ∎

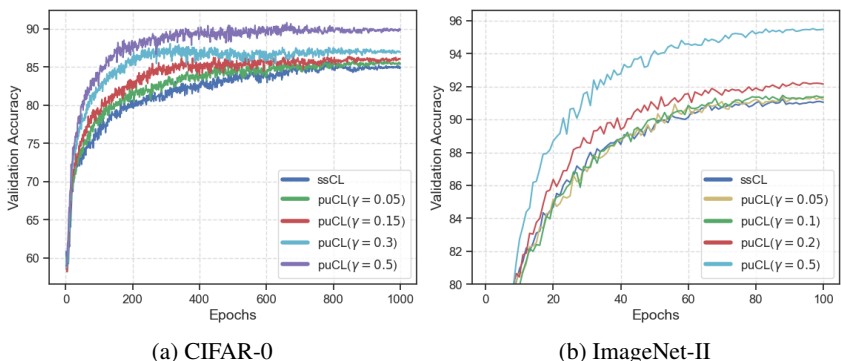

(a) CIFAR-0          (b) ImageNet-II

Figure 9: **Convergence benefits from incorporating labeled positives:** Training ResNet-18 on (a) **CIFAR-0:** Dogs vs Cats subsets from CIFAR10. (b) **ImageNet-II:** ImageWoof vs ImageNette subsets https://github.com/fastai/imagenette from ImageNet-1k. Observe that, by judiciously incorporating available labeled positives into the contrastive loss not only improves generalization it also improves convergence of contrastive representation learning from PU data as explained in Theorem 4
.

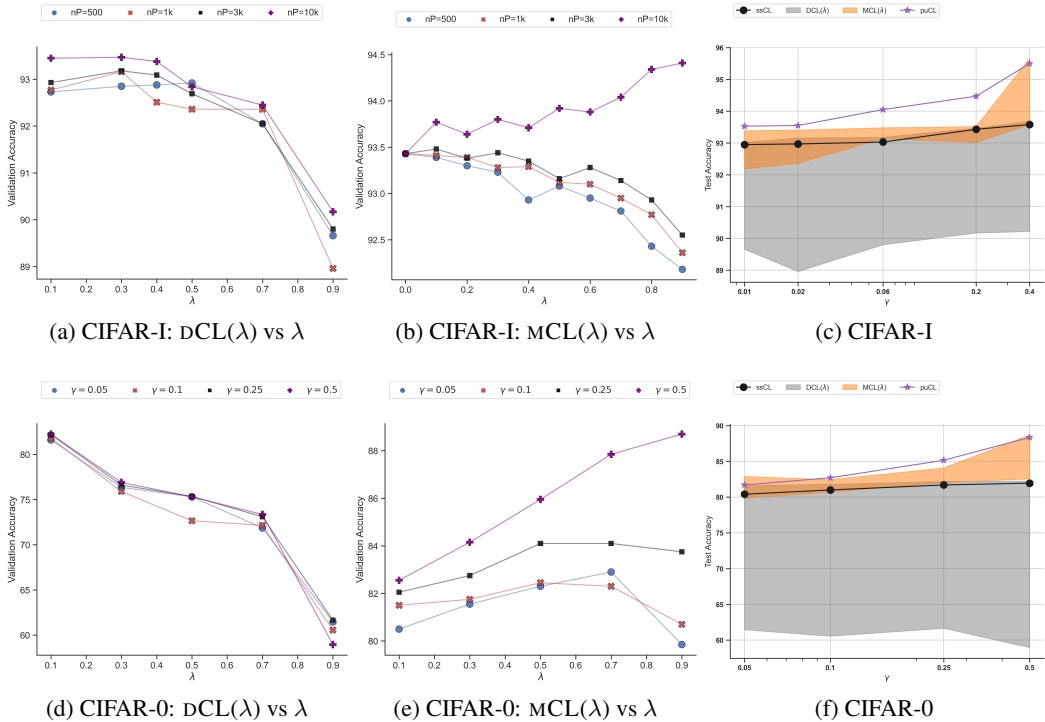

Figure 10: **Parametric Weakly/Un-supervised Contrastive Objectives:** The aprametric objective MCL and DCL are both sensitive to the choice of the hyperparameter. Further, we note that the gains from unsupervised (with bias correction) DCL over SSCL is not as significant as the gains PUCL enjoys over SSCL, thanks to incorporating available positives. MCL however, for suitable choice of $\lambda$ provide competitive performance as PUCL and in certain cases even does better. However, we note that the right choice of hyperparameter depends on the problem and data making it a less attractive choice to adopt in practice.

### A.5.4 COMPARISON WITH PARAMETRIC CONTRASTIVE LEARNING OBJECTIVES:

**Mixed Contrastive Learning (MCL):** At a high level, the main idea of PUCL is to interpolate between the supervised and unsupervised objectives judiciously. By doing so, it is able to exploit the bias-variance trade-off as discussed in Section 2.2.

Another intuitive approach to interpolate between the robustness of SSCL and generalization benefits of SCL-PU could be to simply optimize over a convex combination of the two objectives. Such a hybrid objective has been proven effective in settings with label noise (i.e. when the labels are flipped with some constant probability) (Cui et al., 2023) and thus warrants investigation in the PU learning setting. We refer to this loss as Mixed Contrastive Loss (MCL) defined as follows:

$$\mathcal{L}_{\text{MCL}}(\lambda) = \lambda \mathcal{L}_{\text{SCL-PU}} + (1 - \lambda)\mathcal{L}_{\text{SSCL}} \ , \ 0 \leq \lambda \leq 1 \quad (22)$$

Similar to PUCL; MCL combines two key components: the unsupervised part in MCL enforces consistency between representations learned via label-preserving augmentations (i.e. between $z_i$ and $z_{a(i)} \forall i \in \mathbb{I}$), whereas the supervised component injects structural knowledge derived from available semantic annotation (labeled positives).

It is worth noting that, PUCL can be viewed as a special case of MCL where loss on unlabeled samples is equivalent to $\mathcal{L}_{\text{MCL}}(\lambda = 0)$ and on the labeled samples $\mathcal{L}_{\text{MCL}}(\lambda = 1)$ i.e.

$$\mathcal{L}_{\text{PUCL}} = \frac{1}{n} \sum_{i=1}^{n} \mathbf{1}(\mathbf{x}_i \in \mathbb{P})\ell_{\text{MCL}}^i(1) + \mathbf{1}(\mathbf{x}_i \notin \mathbb{P})\ell_{\text{MCL}}^i(0)$$

In the PU setting, since the structural knowledge of classes perceived by the disambiguation-free objective $\mathcal{L}_{\text{SCL-PU}}$ is noisy, the generalization performance of MCL is sensitive to the choice of

hyper-parameter $\lambda$. This can be attributed to a similar bias-variance trade-off argument as discussed before. We validate this intuition by extensive ablation experiments across various choices of $\lambda$ (Figure 10) under different PU learning scenarios i.e. under varying levels of supervision (varying $\gamma$).

Our experiments suggest that, when available supervision is limited i.e. for small values of $\gamma$ a smaller value of $\lambda$ (i.e. less reliance on supervised part of the loss) is preferred. Conversely, for larger values of $\gamma$ larger contribution from the supervised counterpart is necessary.

Since, the success of MCL is sensitive to the appropriate choice of $\lambda$, tuning which can be quite challenging and depends on the dataset and amount of available supervision, making MCL often less practical in the real world PU learning scenario. Thus, overall, PUCL is a more practical method as it alleviates the need for hyper-parameter tuning and works across various PU Learning scenarios while not suffering from performance degradation.

**Debiased Contrastive Loss (DCL):**   Another popular approach to incorporate latent weak supervision in the unsupervised setting is via appropriately compensating for the sampling bias referred to as debiased contrastive learning (DCL) (Chuang et al., 2020).

Recall the infoNCE family of losses (3):

$$\mathcal{L}^*_{\text{CL}} = \mathop{\mathbb{E}}_{(\mathbf{x}_i, y_i) \sim p(\mathbf{x}, \mathbf{y})} \mathop{\mathbb{E}}_{\substack{\mathbf{x}_j \sim p(\mathbf{x}|y_j = y_i) \\ \{\mathbf{x}_k\}_{k=1}^N \sim p(\mathbf{x}|y_k \neq y_i)}} \left[ \mathbf{z}_i \cdot \mathbf{z}_j - \log \left( \exp(\mathbf{z}_i \cdot \mathbf{z}_j) + \sum_{k=1}^N \exp(\mathbf{z}_i \cdot \mathbf{z}_k) \right) \right],$$

Further, recall that in the fully unsupervised setting, since no supervision is available the negatives are chosen as all the samples in the batch (Chen et al., 2020b) (4).

$$\mathcal{L}_{\text{ssCL}} = -\frac{1}{|\mathbb{I}|} \sum_{i \in \mathbb{I}} \left[ \mathbf{z}_i \cdot \mathbf{z}_{a(i)} - \log \sum_{j \in \mathbb{I}} \mathbf{1}(j \neq i) \exp(\mathbf{z}_i \cdot \mathbf{z}_j) \right]$$

$$= -\frac{1}{|\mathbb{I}|} \sum_{i \in \mathbb{I}} \left[ \mathbf{z}_i \cdot \mathbf{z}_{a(i)} - \log \left( \mathbf{z}_i \cdot \mathbf{z}_{a(i)} + \underbrace{\sum_{i \in \mathbb{I} \setminus \{i, a(i)\}} \exp(\mathbf{z}_i \cdot \mathbf{z}_j)}_{R_{\text{N}}:\text{Negative pairs sum}} \right) \right]$$

Compared to $\mathcal{L}^*_{\text{CL}}$ this finite sample objective is biased since some of the samples treated as negative might belong to the same latent class as the anchor. (Chuang et al., 2020) refers to this phenomenon as sampling bias and propose a modified objective Debiased Contrastive Learning (DCL) to alleviate this issue. In particular, they follow (13) to form an estimate of the negative sum as:

$$R_n^- = \frac{1}{1 - \lambda} \left[ \hat{R}_u^-(v) - \lambda \hat{R}_p^-(v) \right]$$

$\lambda$ is a hyper-parameter that needs to be tuned. Our experiments Figure 10 suggest that we note that the gains from unsupervised (with bias correction) DCL over SSCL is not as significant as the gains PUCL enjoys over SSCL, thanks to incorporating available positives. Moreover, we see that DCL is quite sensitive to the choice of hyperparameter making it hard to adopt in real world.

A.5.5   BIAS VARIANCE TRADEOFF

PROOF OF THEOREM 1.

We restate Theorem 1 for convenience -

**Theorem 1.** $\mathcal{L}_{\text{sCL-PU}}$ (5) is a biased estimator of $\mathcal{L}^*_{\text{CL}}$ characterized as follows:

$$\mathop{\mathbb{E}}_{\mathcal{X}_{\text{PU}}} \left[ \mathcal{L}_{\text{sCL-PU}} \right] - \mathcal{L}^*_{\text{CL}} = \frac{\pi_p(1 - \pi_p)}{1 + \gamma} \left[ 2\tilde{\mu}_{\text{PN}} - (\mu_{\text{P}}^* + \mu_{\text{N}}^*) \right]$$

*Here, $\mu_{\text{P}}^* = \mathbb{E}_{\mathbf{x}_i, \mathbf{x}_j \sim p(\mathbf{x}|y=1)} \left( \mathbf{z}_i \cdot \mathbf{z}_j \right)$ and $\mu_{\text{N}}^* = \mathbb{E}_{\mathbf{x}_i, \mathbf{x}_j \sim p(\mathbf{x}|y=0)} \left( \mathbf{z}_i \cdot \mathbf{z}_j \right)$ capture the proximity between samples from same class marginals and $\tilde{\mu}_{\text{PN}} = \mathbb{E}_{\mathbf{x}_i, \mathbf{x}_j \sim p(\mathbf{x}|y_i \neq y_j)} \left( \mathbf{z}_i \cdot \mathbf{z}_j \right)$ captures the*

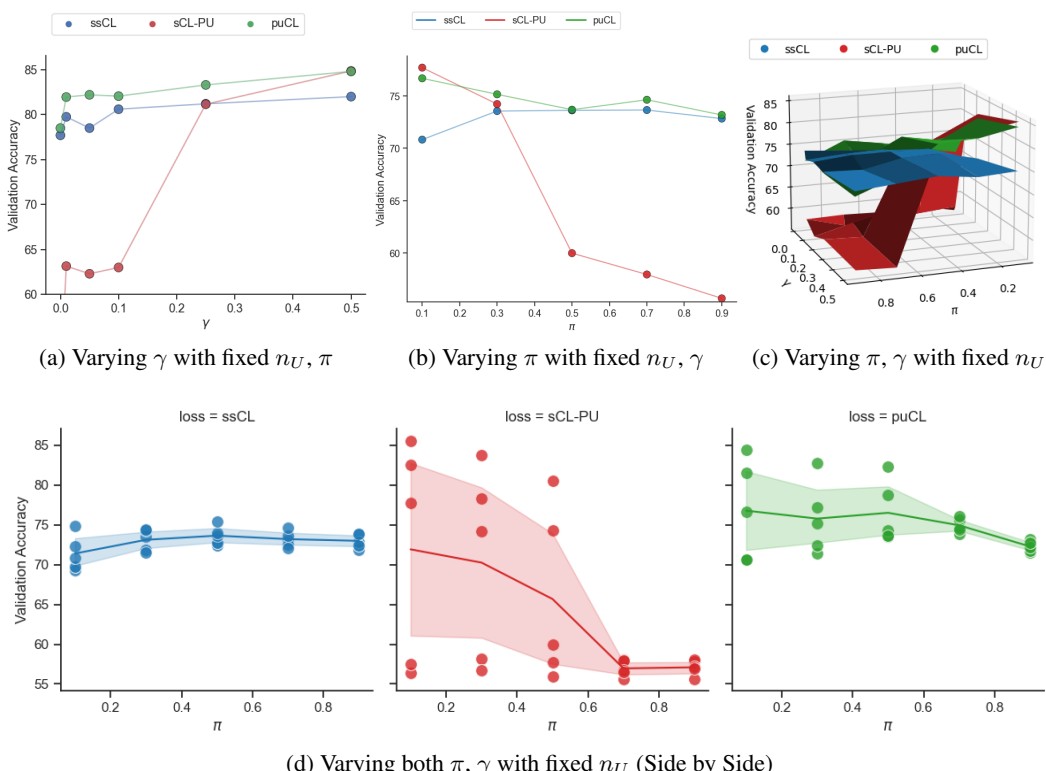

(d) Varying both $\pi, \gamma$ with fixed $n_U$ (Side by Side)

Figure 11: **Ablation of Non-Parametric Contrastive Losses under different PU settings:** To better understand the bias-variance trade-off we experiment with different PU learning settings: class prior $\pi_p$ and amount of labeled data captured by $\gamma = \frac{n_P}{n_U}$ Experiments train ResNet-34 on ImageNet Dogs vs Non-Dogs. Embedding evaluation was performed using fully supervised kNN classification. Overall these experiments indicate that unlike SCL, PUCL and SSCL remain robust across various PU learning settings wherein, PUCL enjoys superior generalization performance on downstream classification. Further, they flesh out several interesting aspects of contrastive learning over PU data and supplement our theoretical findings. Please refer to Section 3 for a detailed discussion.

proximity between dissimilar samples. $\gamma = \frac{n_P}{n_U}$ captures the proportion of cardinality of labeled to unlabeled training subset.

*Proof.* Suppose, $\mathcal{X}_{\text{PU}}$ is generated from the underlying supervised dataset $\mathcal{X}_{\text{PN}} = \mathcal{X}_{\text{P}} \cup \mathcal{X}_{\text{N}}$ i.e. labeled positives $\mathcal{X}_{\text{P}_{\text{L}}}$ is a subset of $n_{\text{P}_{\text{L}}}$ elements chosen uniformly at random from all subsets of $\mathcal{X}_{\text{P}}$ of size $n_{\text{L}} : \mathcal{X}_{\text{P}_{\text{L}}} \subset \mathcal{X}_{\text{P}} = \{\mathbf{x}_i \in \mathbb{R}^d \sim p(\mathbf{x}|y=1)\}_{i=1}^{n_{\text{P}}}$. Further, denote the set positive and negative examples that are unlabeled as $\mathcal{X}_{\text{P}_{\text{U}}}$ and $\mathcal{X}_{\text{N}_{\text{U}}}$.

$$\mathcal{X}_{\text{PU}} = \mathcal{X}_{\text{P}_{\text{L}}} \cup \mathcal{X}_{\text{P}_{\text{U}}} \cup \mathcal{X}_{\text{P}_{\text{N}}} \ , \ \mathcal{X}_{\text{P}} = \mathcal{X}_{\text{P}_{\text{L}}} \cup \mathcal{X}_{\text{P}_{\text{U}}} \text{ and } \mathcal{X}_{\text{U}} = \mathcal{X}_{\text{P}_{\text{U}}} \cup \mathcal{X}_{\text{N}_{\text{U}}} \tag{23}$$

Now, we can establish the result by carefully analyzing the bias of $\mathcal{L}_{\text{SCL-PU}}$ (5) in estimating the ideal contrastive loss (3) over each of these subsets.

For the labeled positive subset $\mathcal{X}_{\text{P}_{\text{L}}}$ the bias can be computed as:

$$\mathcal{B}_{\mathcal{L}_{\text{SCL-PU}}}(\mathbf{x}_i \in \mathcal{X}_{\text{P}_{\text{L}}}) = \mathbb{E}_{\mathbf{x}_i \in \mathcal{X}_{\text{P}_{\text{L}}}} \left[ \frac{1}{n_{\text{P}_{\text{L}}}} \sum_{\mathbf{x}_j \in \mathcal{X}_{\text{P}_{\text{L}}}} \mathbf{z}_i \cdot \mathbf{z}_j \right] - \mathbb{E}_{\mathbf{x}_i, \mathbf{x}_j \sim p(\mathbf{x}|y=1)} \left[ \mathbf{z}_i \cdot \mathbf{z}_j \right] \tag{24}$$

Clearly under the PU setting (1), since the labeled positives are i.i.d samples from the positive marginal the first expectation the two expectations align implying the zero bias. However obviously the variance scales inversely with the amount of labeled positives $\mathcal{O}(\frac{1}{|\mathcal{X}_{\text{P}_{\text{L}}}|})$.

For the unlabeled positive subset $\mathcal{X}_{P_U}$ the bias can be computed as:

$$
\begin{aligned}
\mathcal{B}_{\mathcal{L}_{\text{sCL-PU}}}(\mathbf{x}_i \in \mathcal{X}_{P_U}) &= \mathbb{E}_{\mathbf{x}_i \in \mathcal{X}_{P_U}}\left[\frac{1}{n_U}\sum_{\mathbf{x}_j \in \mathcal{X}_U} \mathbf{z}_i \cdot \mathbf{z}_j\right] - \mathbb{E}_{\mathbf{x}_i, \mathbf{x}_j \sim p(\mathbf{x}|y=1)}\left[\mathbf{z}_i \cdot \mathbf{z}_j\right] \\
&= \mathbb{E}_{\mathbf{x}_i \in \mathcal{X}_{P_U}}\left[\pi_p \mathbb{E}_{\mathbf{x}_j \in \mathcal{X}_{P_U}}\left(\mathbf{z}_i \cdot \mathbf{z}_j\right) + (1-\pi_p)\mathbb{E}_{\mathbf{x}_j \in \mathcal{X}_{N_U}}\left(\mathbf{z}_i \cdot \mathbf{z}_j\right)\right] - \mu_P^* \\
&= \pi_p \mu_P^* + (1-\pi_p)\mathbb{E}_{\mathbf{x}_i \in \mathcal{X}_{P_U}}\left[\mathbb{E}_{\mathbf{x}_j \in \mathcal{X}_{N_U}}\left(\mathbf{z}_i \cdot \mathbf{z}_j\right)\right] - \mu_P^* \\
&= (1-\pi_p)\mathbb{E}_{\mathbf{x}_i \sim p(\mathbf{x}|y=1)}\left[\mathbb{E}_{\mathbf{x}_j \sim p(\mathbf{x}|y=0)}\left(\mathbf{z}_i \cdot \mathbf{z}_j\right)\right] - (1-\pi_p)\mu_P^* \\
&= (1-\pi_p)\mathbb{E}_{\mathbf{x}_i, \mathbf{x}_j \sim p(\mathbf{x}|y_i \neq y_j)}\left(\mathbf{z}_i \cdot \mathbf{z}_j\right) - (1-\pi_p)\mu_P^* \\
&= (1-\pi_p)\tilde{\mu}_{PN} - (1-\pi_p)\mu_P^*
\end{aligned}
$$

Finally, for the negative unlabeled set:

$$
\begin{aligned}
\mathcal{B}_{\mathcal{L}_{\text{sCL-PU}}}(\mathbf{x}_i \in \mathcal{X}_{N_U}) &= \mathbb{E}_{\mathbf{x}_i \in \mathcal{X}_{N_U}}\left[\frac{1}{n_U}\sum_{\mathbf{x}_j \in \mathcal{X}_U} \mathbf{z}_i \cdot \mathbf{z}_j\right] - \mathbb{E}_{\mathbf{x}_i, \mathbf{x}_j \sim p(\mathbf{x}|y=0)}\left[\mathbf{z}_i \cdot \mathbf{z}_j\right] \\
&= \mathbb{E}_{\mathbf{x}_i \in \mathcal{X}_{N_U}}\left[\pi_p \mathbb{E}_{\mathbf{x}_j \in \mathcal{X}_{P_U}}\left(\mathbf{z}_i \cdot \mathbf{z}_j\right) + (1-\pi_p)\mathbb{E}_{\mathbf{x}_j \in \mathcal{X}_{N_U}}\left(\mathbf{z}_i \cdot \mathbf{z}_j\right)\right] - \mu_N^* \\
&= \pi_p \mathbb{E}_{\mathbf{x}_i, \mathbf{x}_j \sim p(\mathbf{x}|y_i \neq y_j)}\left(\mathbf{z}_i \cdot \mathbf{z}_j\right) + (1-\pi_p)\mathbb{E}_{\mathbf{x}_i, \mathbf{x}_j \sim p(\mathbf{x}|y=0)}\left(\mathbf{z}_i \cdot \mathbf{z}_j\right) - \mu_N^* \\
&= \pi_p \tilde{\mu}_{PN} - \pi_p \mu_N^*
\end{aligned}
$$

Now, using the fact that the unlabeled examples are sampled uniformly at random from the mixture distribution with positive mixture weight $\pi_p$ we can compute the total bias as follows:

$$
\mathcal{B}_{\mathcal{L}_{\text{sCL-PU}}}(\mathbf{x}_i \in \mathcal{X}_{PU}) = \frac{\pi_p}{1+\gamma}\mathcal{B}_{\mathcal{L}_{\text{sCL-PU}}}(\mathbf{x}_i \in \mathcal{X}_{P_U}) + \frac{1-\pi_p}{1+\gamma}\mathcal{B}_{\mathcal{L}_{\text{sCL-PU}}}(\mathbf{x}_i \in \mathcal{X}_{N_U}) \text{ where } \gamma = \frac{|\mathcal{X}_{P_L}|}{|\mathcal{X}_U|}
$$

We get the desired result by plugging in the bias of the subsets and simplifying. ∎

PROOF OF THEOREM 2.

We restate Theorem 1, 2 for convenience -

**Theorem 2.** *Assume that $\mathbf{x}_i, \mathbf{x}_{a(i)}$ are i.i.d draws from the same class marginal (Saunshi et al., 2019; Tosh et al., 2021), then it follows that the objective functions $\mathcal{L}_{\text{SSCL}}$ (4) and $\mathcal{L}_{\text{PUCL}}$ (6) are unbiased estimators of $\mathcal{L}_{\text{CL}}^*$ (3). Additionally, it holds that:*

$$
\Delta_\sigma(\gamma) \geq 0 \ \forall \gamma \geq 0 \ ; \ \Delta_\sigma(\gamma_1) \geq \Delta_\sigma(\gamma_2) \ \forall \gamma_1 \geq \gamma_2 \geq 0
$$

*where, $\Delta_\sigma(\gamma) = \text{Var}(\mathcal{L}_{\text{SSCL}}) - \text{Var}(\mathcal{L}_{\text{PUCL}})$.*

*Proof.* We first prove that both $\mathcal{L}_{\text{SSCL}}$ (4) and $\mathcal{L}_{\text{PUCL}}$ (6) are unbiased estimators of $\mathcal{L}_{\text{CL}}^*$ (3) using a similar analysis as the previous proof.

For the labeled positive subset $\mathcal{X}_{P_L}$ the bias can be computed as:

$$
\mathcal{B}_{\mathcal{L}_{\text{PUCL}}}(\mathbf{x}_i \in \mathcal{X}_{P_L}) = \mathbb{E}_{\mathbf{x}_i \in \mathcal{X}_{P_L}}\left[\frac{1}{n_{P_L}}\sum_{\mathbf{x}_j \in \mathcal{X}_{P_L}} \mathbf{z}_i \cdot \mathbf{z}_j\right] - \mathbb{E}_{\mathbf{x}_i, \mathbf{x}_j \sim p(\mathbf{x}|y=1)}\left[\mathbf{z}_i \cdot \mathbf{z}_j\right] = 0
$$

Here we have used the fact that labeled positives are drawn i.i.d from the positive marginal. For the unlabeled samples

$$\mathcal{B}_{\mathcal{L}_{\text{PUCL}}}(\mathbf{x}_i \in \mathcal{X}_{\text{U}}) = \mathbb{E}_{\mathbf{x}_i \in \mathcal{X}_{\text{U}}}\left[\mathbf{z}_i \cdot \mathbf{z}_{a(i)}\right] - \mathbb{E}_{\mathbf{x}_i, \mathbf{x}_j \sim p(\mathbf{x}|\mathbf{y}_i = \mathbf{y}_j)}\left[\mathbf{z}_i \cdot \mathbf{z}_j\right]$$

$$= \mathbb{E}_{\mathbf{x}_i, \mathbf{x}_j \sim p(\mathbf{x}|\mathbf{y}_i = \mathbf{y}_j)}\left[\mathbf{z}_i \cdot \mathbf{z}_j\right] - \mathbb{E}_{\mathbf{x}_i, \mathbf{x}_j \sim p(\mathbf{x}|\mathbf{y}_i = \mathbf{y}_j)}\left[\mathbf{z}_i \cdot \mathbf{z}_j\right] = 0$$

Thus $\mathcal{L}_{\text{PUCL}}$ is an unbiased estimator of $\mathcal{L}_{\text{CL}}^*$. Clearly, the i.i.d assumption similarly implies $\mathcal{L}_{\text{SSCL}}$ is also an unbiased estimator.

Next we can do a similar decomposition of the variances for both the objectives. Then the difference of variance under the PU dataset -

$$\Delta_\sigma(\mathcal{X}_{\text{PU}}) = \text{Var}_{\mathcal{L}_{\text{SSCL}}}(\mathcal{X}_{\text{PU}}) - \text{Var}_{\mathcal{L}_{\text{PUCL}}}(\mathcal{X}_{\text{PU}})$$
$$= \Delta_\sigma(\mathcal{X}_{\text{P}_{\text{L}}}) + \Delta_\sigma(\mathcal{X}_{\text{U}})$$
$$= \Delta_\sigma(\mathcal{X}_{\text{P}_{\text{L}}})$$
$$= \left(1 - \frac{1}{n_{\text{P}_{\text{L}}}}\right)\text{Var}\left(\mathbf{z}_i \cdot \mathbf{z}_j : \mathbf{x}_i, \mathbf{x}_j \in \mathcal{X}_{\text{P}_{\text{L}}}\right)$$
$$= \left(1 - \frac{1}{\gamma|\mathcal{X}_{\text{U}}|}\right)\text{Var}\left(\mathbf{z}_i \cdot \mathbf{z}_j : \mathbf{x}_i, \mathbf{x}_j \in \mathcal{X}_{\text{P}_{\text{L}}}\right)$$

Clearly, since variance is non-negative we have $\forall \gamma > 0 : \Delta_\sigma(\mathcal{X}_{\text{PU}}) \geq 0$

Now consider two settings where we have different amounts of labeled positives defined by ratios $\gamma_1$ and $\gamma_2$ and denote the two resulting datasets $\mathcal{X}_{\text{PU}}^{\gamma_1}$ and $\mathcal{X}_{\text{PU}}^{\gamma_2}$ then

$$\Delta_\sigma(\mathcal{X}_{\text{PU}}^{\gamma_1}) - \Delta_\sigma(\mathcal{X}_{\text{PU}}^{\gamma_2}) = \Delta_\sigma(\mathcal{X}_{\text{P}_{\text{L}}}^{\gamma_1}) - \Delta_\sigma(\mathcal{X}_{\text{P}_{\text{L}}}^{\gamma_2})$$
$$= \frac{1}{|\mathcal{X}_{\text{U}}|}\left(\frac{1}{\gamma_2} - \frac{1}{\gamma_1}\right)\text{Var}\left(\mathbf{z}_i \cdot \mathbf{z}_j : \mathbf{x}_i, \mathbf{x}_j \in \mathcal{X}_{\text{P}_{\text{L}}}\right)$$
$$\geq 0$$

The last inequality holds since $\gamma_1 \geq \gamma_2$. This concludes the proof. ∎

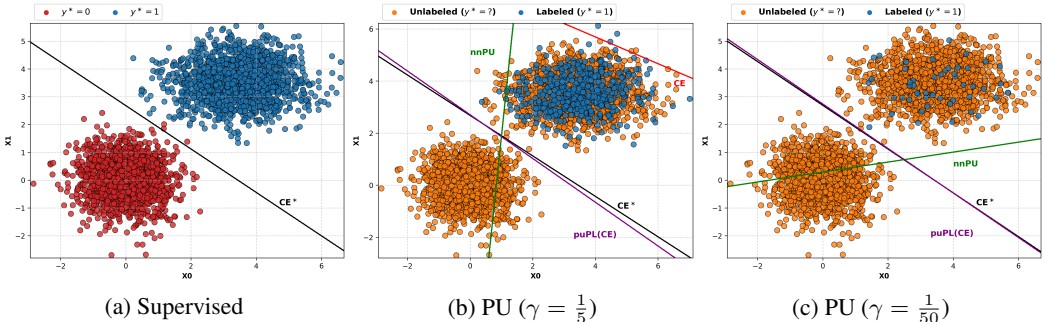

| (a) Supervised | (b) PU ($\gamma = \frac{1}{5}$) | (c) PU ($\gamma = \frac{1}{50}$) |

Figure 12: **Geometric Intuition of PUPL (separable):** We train logistic regression over (almost) separable 2D Gaussian Mixture. **CE**$^*$ denote the supervised classifier for comparison with the decision boundaries obtained by **CE**, **NNPU** (trained with $\pi_p^*$) and PUPL(CE). It is clear that, as $\gamma = \frac{n_P}{n_U}$ is decreased **CE** soon diverges; **NNPU** suffers from significant decision boundary deviation. On the other hand, PUPL(CE) almost surely remains close to the true decision boundary as long as the feature space displays inherent clustering structure.

## A.6  PUPL: Positive Unlabeled Pseudo Labeling

**Summary:** Consider performing PU learning over a (almost) linearly separable feature space (i.e., where the true positives and true negatives form separate clusters).

- Standard supervised classification loss, e.g., CE, suffers from decision boundary deviation when the number of labeled examples is limited.

- Cost-sensitive PU learning approaches address this issue by forming an unbiased risk estimator by leveraging the unlabeled and labeled positives (see Appendix A.3.3). However, we observe that when only a handful of positives are labeled, even these approaches are unable to recover the ideal decision boundary as the unbiased estimate suffers from large variance.

- Our proposed approach PUPL (Algorithm 1(B)) on the other hand, is able to identify the correct pseudo-labels (i.e. cluster assignments) almost surely (within constant multiplicative approximation error).

- Consequently training using the pseudo-labels with standard classification loss e.g. CE loss often achieves decision boundaries that closely align with the true boundaries.

Consider the naive disambiguation-free setup where the unlabeled samples are treated as pseudo-negatives and the classifier is trained using standard CE loss. As demonstrated in Figure 14, the bias induced via the pseudo-labeling results in the decision boundary to deviate further from the true (fully supervised) decision boundary when only a limited amount of labeled examples are available.

### A.6.1  Necessary Definitions and Intermediate Lemmas

Before proceeding with the proofs we would need to make some definitions more formal and state some intermediate results.

**Definition 2 (Supervised Dataset).** *Let $\mathcal{X}_{\mathrm{PN}}$ denote the true underlying fully supervised (PN) dataset*

$$\mathcal{X}_{\mathrm{PN}} = \{\mathbf{x}_i \sim p(\mathbf{x})\}_{i=1}^n = \mathcal{X}_{\mathrm{P}} \cup \mathcal{X}_{\mathrm{N}}, \ \mathcal{X}_{\mathrm{P}} = \{\mathbf{x}_i^{\mathrm{P}}\}_{i=1}^{n_{\mathrm{P}}} \overset{i.i.d.}{\sim} p_p(\mathbf{x}), \ \mathcal{X}_{\mathrm{N}} = \{\mathbf{x}_i^{\mathrm{N}}\}_{i=1}^{n_{\mathrm{N}}} \overset{i.i.d.}{\sim} p_n(\mathbf{x})$$

*where, $p(\mathbf{x})$ denotes the underlying true mixture distribution; $p_p(\mathbf{x}) = p(\mathbf{x}|y = 1)$ and $p_n(\mathbf{x}) = p(\mathbf{x}|y = 0)$ denote the underlying positive and negative class marginal respectively.*

**Definition 3 (Class Prior).** *The mixture component weights of $p(\mathbf{x})$ are $\pi_p$ and $\pi_n = 1 - \pi_p$.*

$$p(\mathbf{x}) = \pi_p p_p(\mathbf{x}) + (1 - \pi_p)p(\mathbf{x}|y = 0) \ where, \ \pi_p = p(y = 1|\mathbf{x})$$

**Definition 4 (Positive Unlabeled Dataset).** *Let $p(\mathbf{x})$ denotes the underlying true mixture distribution of positive and negative class with class prior $\pi_p = p(y = 1|\mathbf{x})$. Further let, $p_p(\mathbf{x}) = p(\mathbf{x}|y = 1)$ and $p_n(\mathbf{x}) = p(\mathbf{x}|y = 0)$ denote the true underlying positive and negative class marginal respectively.*

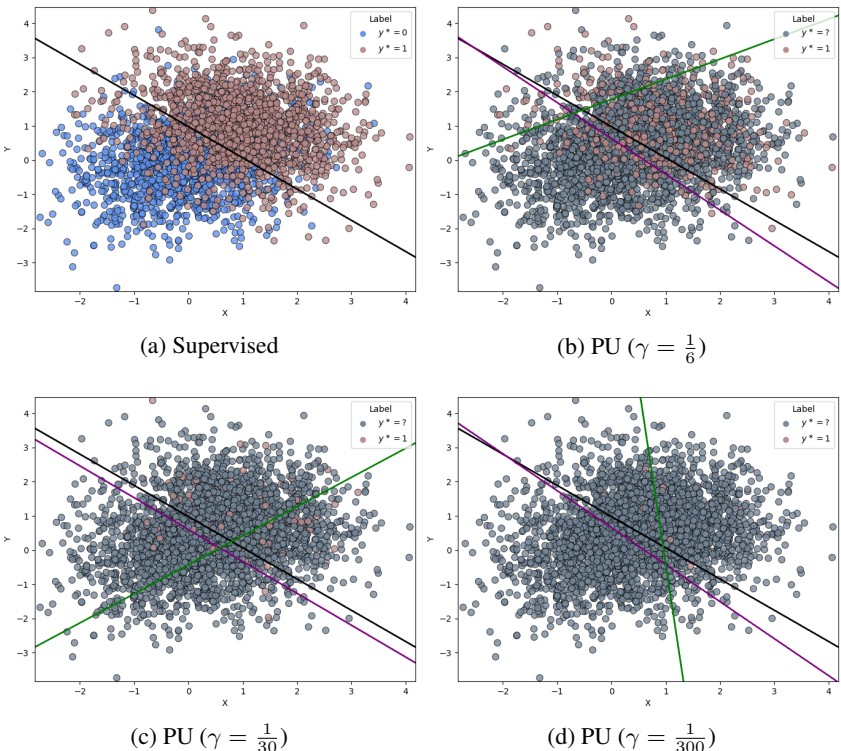

Figure 13: **Geometric Intuition of PUPL (overlapping):** Here we aim to learn a linear classifier over overlapping gaussians. We note that, puPL matches the bounday of supervised CE even in this setting.

*Then the PU dataset is generated as:*

$$\mathcal{X}_{\text{PU}} = \mathcal{X}_{\text{P}_{\text{L}}} \cup \mathcal{X}_{\text{U}}, \ \mathcal{X}_{\text{P}_{\text{L}}} = \{\mathbf{x}_i^{\text{P}}\}_{i=1}^{n_{\text{P}_{\text{L}}}} \overset{i.i.d.}{\sim} p_p(\mathbf{x}), \ \mathcal{X}_{\text{U}} = \{\mathbf{x}_i^{\text{U}} \overset{i.i.d.}{\sim} p(\mathbf{x})\}_{i=1}^{n_u}$$

**Definition 5** (**Clustering**). *A clustering refers to a set of centroids* $C = \{\mu_{\text{P}}, \mu_{\text{N}}\}$ *that defines the following pseudo-labels to the unlabeled instances:*

$$\forall \mathbf{z}_i \in \mathcal{Z}_U : \tilde{y}_i = \begin{cases} 1, & \text{if } \mu_P = \arg\min_{\mu \in C} \|\mathbf{z}_i - \mu\|^2 \\ 0, & \text{otherwise} \end{cases}$$

**Definition 6** (**Potential Function**). *Given a clustering* $C$ *the potential function over the dataset is:*

$$\phi(\mathcal{Z}_{\text{PU}}, C) = \sum_{\mathbf{z}_i \in \mathcal{Z}_{\text{PU}}} \min_{\mu \in C} \|\mathbf{z}_i - \mu\|^2, \ \mathcal{Z}_{\text{PU}} = \{\mathbf{z}_i = g_{\mathbf{B}}(\mathbf{x}_i) \in \mathbb{R}^k : \mathbf{x}_i \in \mathcal{X}_{\text{PU}}\}$$

**Definition 7** (**Optimal Clustering**). *Refers to the optimal clustering* $C^* = \{\mu_{\text{P}}^*, \mu_{\text{N}}^*\}$ *that solves the k-means problem i.e. attains the minimum potential function:*

$$\phi^*(\mathcal{Z}_{\text{PU}}, C^*) = \sum_{\mathbf{z}_i \in \mathcal{Z}_{\text{PU}}} \min_{\mu \in C^*} \|\mathbf{z}_i - \mu\|^2, \ \mathcal{Z}_{\text{PU}} = \{\mathbf{z}_i = g_{\mathbf{B}}(\mathbf{x}_i) \in \mathbb{R}^k : \mathbf{x}_i \in \mathcal{X}_{\text{PU}}\}$$

**Definition 8** ($D^2$). *Given clustering* $C$ *the* $D^2(\cdot) : \mathbb{R}^d \to \mathbb{R}^+$ *score is:*

$$D^2(\mathbf{x}) = \phi(\{\mathbf{x}\}, C)$$

Central to the analysis is the following two lemmas:

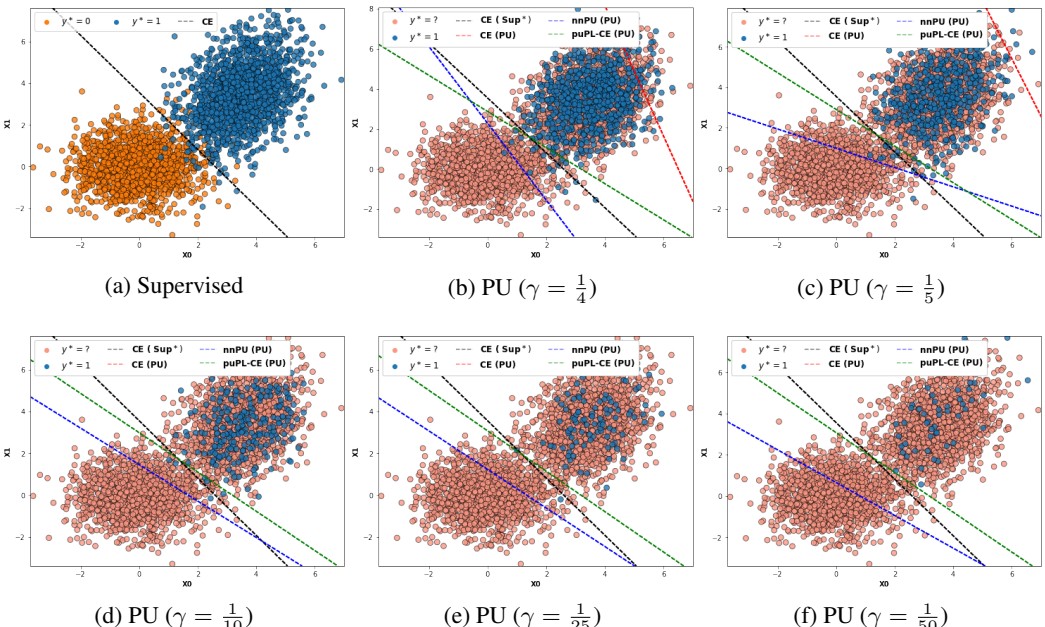

(a) Supervised     (b) PU ($\gamma = \frac{1}{4}$)     (c) PU ($\gamma = \frac{1}{5}$)

(d) PU ($\gamma = \frac{1}{10}$)     (e) PU ($\gamma = \frac{1}{25}$)     (f) PU ($\gamma = \frac{1}{50}$)

Figure 14: **Decision Boundary Deviation :** Training a linear classifier over 2D Gaussian Mixture.
(0) CE* - Denotes the ideal supervised classification boundary.
(1) CE - We consider the standard disambiguation-free CE loss - wherein the unlabeled samples are simply treated as pseudo-negatives and a binary classifier is trained to separate the labeled positives from these pseudo negatives (unlabeled) examples. CE loss is unable to recover the true decision boundary (i.e. the decision boundary learnt in the fully supervised setting) in this setting. In fact as amount of labeled positives decrease (i.e. for smaller $\gamma$) clearly the decision boundary deviates dramatically from the true boundary due to the biased supervision and eventually diverges.
(2) nnPU - Models trained with nnPU objective (Kiryo et al., 2017). We note that nnPU is significantly more robust than the naive disambiguation-free approach. Especially when sufficient labeled positives are provided the decision boundary learnt by nnPU is closely aligned with the true decision boundary. However, when only a handful of positives are labeled we observe that nnPU might also result in significant generation gap possibly because the variance of the estimator is high in this case. Note that, *All nnPU experiments here are run with oracle knowledge of class prior information $\pi_p^* = \frac{1}{2}$*.
(3) puPL + CE - On the other hand our clustering based pseudo-labeling approach almost surely recovers the true underlying labels even when only a few positive examples are available resulting in consistent improvement over existing SOTA cost-sensitive approaches. Further our approach obviates the need of class prior knowledge unlike nnPU.

**Lemma 2** (**Positive Centroid Estimation**). *Suppose, $\mathcal{Z}_{P_L}$ is a subset of $n_L$ elements chosen uniformly at random from all subsets of $\mathcal{Z}_P$ of size $n_L$ : $\mathcal{Z}_{P_L} \subset \mathcal{Z}_P = \{\mathbf{z}_i = g_\mathbf{B}(\mathbf{x}_i) \in \mathbb{R}^k : \mathbf{x}_i \in \mathbb{R}^d \sim p(\mathbf{x}|\mathbf{y} = 1)\}_{i=1}^{n_P}$ implying that the labeled positives are generated according to (1). Let, $\mu$ denote the centroid of $\mathcal{Z}_{P_L}$ i.e. $\mu = \frac{1}{n_{P_L}} \sum_{\mathbf{z}_i \in \mathcal{Z}_{P_L}} \mathbf{z}_i$ and $\mu^*$ denote the optimal centroid of $\mathcal{Z}_P$ i.e. $\phi^*(\mathcal{Z}_P, \mu^*) = \sum_{\mathbf{z}_i \in \mathcal{Z}_{PU}} \|\mathbf{z}_i - \mu^*\|^2$ then we can establish the following result:*

$$\mathbb{E}\left[\phi(\mathcal{Z}_P, \mu)\right] = \left(1 + \frac{n_P - n_{P_L}}{n_{P_L}(n_P - 1)}\right)\phi^*(\mathcal{Z}_P, \mu^*)$$

**Lemma 3** ($k$-**means++ Seeding**). *Given initial cluster center $\mu_P = \frac{1}{n_{P_L}} \sum_{\mathbf{z}_i \in \mathcal{Z}_{P_L}} \mathbf{z}_i$, if the second centroid $\mu_N$ is chosen according to the distribution $D(\mathbf{z}) = \frac{\phi(\{\mathbf{z}\}, \{\mu_P\})}{\sum_{\mathbf{z} \in \mathcal{Z}_U} \phi(\{\mathbf{z}\}, \{\mu_P\})} \; \forall \mathbf{z} \in \mathcal{Z}_U$, then:*

$$\mathbb{E}\left[\phi(\mathcal{Z}_{PU}, \{\mu_P, \mu_N\})\right] \leq 2\phi(\mathcal{Z}_{P_L}, \{\mu_P\}) + 16\phi^*(\mathcal{Z}_U, C^*)$$

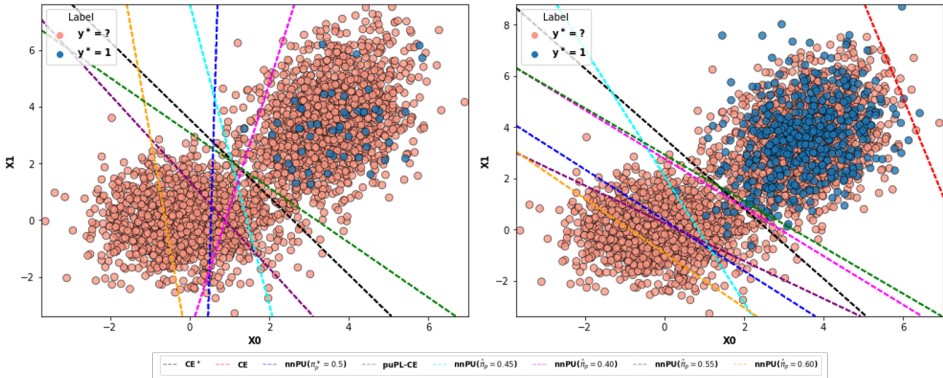

Figure 15: **Sensitivity to Class Prior Estimate:** Training a linear classifier over 2D Gaussian Mixture. We vary the estimated class prior as $\hat{\pi}_p = (1 \pm 0.2)\pi_p^*$ to test the robustness. As we see that the cost-sensitive baseline nnPU suffers significant variance due to approximation error in class prior estimation. The proposed PUCL followed by CE training on the other hand obviates the need for class prior estimation and consistently produces better PU classification than nnPU.

### A.6.2 PROOF OF THEOREM 3.

We restate Theorem 3 for convenience -

**Theorem 3.** *Suppose, PU data is generated as* (1)*, then running Algorithm 1(B) on $\mathcal{Z}_{\mathrm{PU}}$ yields:* $\mathbb{E}\big[\phi(\mathcal{Z}_{\mathrm{PU}}, C_{\mathrm{PUPL}})\big] \leq 16\phi^*(\mathcal{Z}_{\mathrm{PU}}, C^*)$. *In comparison, running k-means++ on $\mathcal{Z}_{\mathrm{PU}}$ we get,* $\mathbb{E}\big[\phi(\mathcal{Z}_{\mathrm{PU}}, C_{\mathbf{k-means++}})\big] \leq 21.55\phi^*(\mathcal{Z}_{\mathrm{PU}}, C^*)$.

We will closely follows the proof techniques from (Arthur & Vassilvitskii, 2007) *mutatis mutandis* to prove this theorem.

*Proof.* Recall that we choose our first center from supervision i.e. $\mu_{\mathrm{P}} = \frac{1}{n_{\mathrm{P_L}}} \sum_{\mathbf{z}_i \in \mathcal{Z}_{\mathrm{P_L}}} \mathbf{z}_i$ and then choose the next center from the unlabeled samples according to probability $D(\mathbf{z}) = \frac{\phi(\{\mathbf{z}\}, \{\mu_{\mathrm{P}}\})}{\sum_{\mathbf{z} \in \mathcal{Z}_{\mathrm{U}}} \phi(\{\mathbf{z}\}, \{\mu_{\mathrm{P}}\})} \ \forall \mathbf{z} \in \mathcal{Z}_{\mathrm{U}}$. Then, from Lemma 3:

$$\mathbb{E}\left[\phi(\mathcal{Z}_{\mathrm{PU}}, \{\mu_{\mathrm{P}}, \mu_{\mathrm{N}}\})\right] \leq 2\phi(\mathcal{Z}_{\mathrm{P_L}}, \{\mu_{\mathrm{P}}\}) + 16\phi^*(\mathcal{Z}_{\mathrm{U}}, C^*)$$

$$= 2\phi(\mathcal{Z}_{\mathrm{P_L}}, \{\mu_{\mathrm{P}}\}) + 16\left(\phi^*(\mathcal{Z}_{\mathrm{PU}}, C^*) - \phi^*(\mathcal{Z}_{\mathrm{P_L}}, C^*)\right)$$

$$= 2\left(\phi(\mathcal{Z}_{\mathrm{P_L}}, \{\mu_{\mathrm{P}}\}) - 8\phi^*(\mathcal{Z}_{\mathrm{P_L}}, \{\mu_{\mathrm{P}}^*\})\right) + 16\phi^*(\mathcal{Z}_{\mathrm{PU}}, C^*)$$

Now we use Lemma 2 to bound the first term -

$$\mathbb{E}\left[\phi(\mathcal{Z}_{\mathrm{PU}}, \{\mu_{\mathrm{P}}, \mu_{\mathrm{N}}\})\right] \leq 2\left[\left(1 + \frac{n_{\mathrm{P}} - n_{\mathrm{P_L}}}{n_{\mathrm{P_L}}(n_{\mathrm{P}} - 1)}\right) - 8\right]\phi^*(\mathcal{Z}_{\mathrm{P_L}}, \{\mu_{\mathrm{P}}^*\}) + 16\phi^*(\mathcal{Z}_{\mathrm{PU}}, C^*)$$

$$\leq 2\left[\frac{n_{\mathrm{P}} - n_{\mathrm{P_L}}}{n_{\mathrm{P_L}}(n_{\mathrm{P}} - 1)} - 7\right]\phi^*(\mathcal{Z}_{\mathrm{P_L}}, \{\mu_{\mathrm{P}}^*\}) + 16\phi^*(\mathcal{Z}_{\mathrm{PU}}, C^*)$$

$$\leq 16\phi^*(\mathcal{Z}_{\mathrm{PU}}, C^*)$$

Note that this bound is much tighter in practice when a large amount of labeled examples are available i.e. for larger values of $n_{\mathrm{P_L}}$. Additionally our guarantee holds only after the initial cluster assignments are found. Subsequent standard $k$-means iterations can only further decrease the potential.

On the other hand for $k$-means++ strategy (Arthur & Vassilvitskii, 2007) the guarantee is:

$$\mathbb{E}\left[\phi(\mathcal{Z}_{\text{PU}}, C_{k-\text{means++}})\right] \leq \left(2 + \ln 2\right) 8\phi^*(\mathcal{Z}_{\text{PU}}, C^*) \approx 21.55\phi^*(\mathcal{Z}_{\text{PU}}, C^*)$$

This concludes the proof. ∎

### A.6.3 PROOF OF LEMMA 2

*Proof.*

$$\mathbb{E}\left[\phi(\mathcal{Z}_{\text{P}}, \mu)\right] = \mathbb{E}\left[\sum_{\mathbf{z}_i \in \mathcal{Z}_{\text{P}}} \|\mathbf{z}_i - \mu\|^2\right]$$

$$= \mathbb{E}\left[\sum_{\mathbf{z}_i \in \mathcal{Z}_{\text{P}}} \|\mathbf{z}_i - \mu^*\|^2 + n_{\text{P}}\|\mu - \mu^*\|^2\right]$$

$$= \phi^*(\mathcal{Z}_{\text{P}}, \mu^*) + n_{\text{P}}\mathbb{E}\left[\|\mu - \mu^*\|^2\right]$$

Now we can compute the expectation as:

$$\mathbb{E}\left[\|\mu - \mu^*\|^2\right] = \mathbb{E}\left[\mu^T\mu\right] + \mu^{*T}\mu^* - 2\mu^{*T}\mathbb{E}\left[\frac{1}{n_{\text{P}_{\text{L}}}} \sum_{\mathbf{z}_i \in \mathcal{Z}_{\text{P}_{\text{L}}}} \mathbf{z}_i\right]$$

$$= \mathbb{E}\left[\mu^T\mu\right] + \mu^{*T}\mu^* - 2\mu^{*T}\frac{1}{n_{\text{P}_{\text{L}}}}\mathbb{E}\left[\sum_{\mathbf{z}_i \in \mathcal{Z}_{\text{P}_{\text{L}}}} \mathbf{z}_i\right]$$

$$= \mathbb{E}\left[\mu^T\mu\right] + \mu^{*T}\mu^* - 2\mu^{*T}\frac{1}{n_{\text{P}_{\text{L}}}}n_{\text{P}_{\text{L}}}\mathbb{E}_{\mathbf{z}_i \in \mathcal{Z}_{\text{P}}}\left[\mathbf{z}_i\right]$$

$$= \mathbb{E}\left[\mu^T\mu\right] - \mu^{*T}\mu^*$$

We can compute the first expectation as:

$$\mathbb{E}\left[\mu^T\mu\right] = \frac{1}{n_{\text{P}_{\text{L}}}^2}\mathbb{E}\left[\left(\sum_{\mathbf{z}_i \in \mathcal{Z}_{\text{P}_{\text{L}}}} \mathbf{z}_i\right)^T\left(\sum_{\mathbf{z}_i \in \mathcal{Z}_{\text{P}_{\text{L}}}} \mathbf{z}_i\right)\right]$$

$$= \frac{1}{n_{\text{P}_{\text{L}}}^2}\left[p(i \neq j)\sum_{\mathbf{z}_i, \mathbf{z}_j \in \mathcal{Z}_{\text{P}}, i \neq j} \mathbf{z}_i^T\mathbf{z}_j + p(i = j)\sum_{\mathbf{z}_i \in \mathcal{Z}_{\text{P}}} \mathbf{z}_i^T\mathbf{z}_i\right]$$

$$= \frac{1}{n_{\text{P}_{\text{L}}}^2}\left[\frac{\binom{n_{\text{P}}-2}{n_{\text{P}_{\text{L}}}-2}}{\binom{n_{\text{P}}}{n_{\text{P}_{\text{L}}}}}\sum_{\mathbf{z}_i, \mathbf{z}_j \in \mathcal{Z}_{\text{P}}, i \neq j} \mathbf{z}_i^T\mathbf{z}_j + \frac{\binom{n_{\text{P}}-1}{n_{\text{P}_{\text{L}}}-1}}{\binom{n_{\text{P}}}{n_{\text{P}_{\text{L}}}}}\sum_{\mathbf{z}_i \in \mathcal{Z}_{\text{P}}} \mathbf{z}_i^T\mathbf{z}_i\right]$$

$$= \frac{1}{n_{\text{P}_{\text{L}}}^2}\left[\frac{n_{\text{P}_{\text{L}}}(n_{\text{P}_{\text{L}}} - 1)}{n_{\text{P}}(n_{\text{P}} - 1)}\sum_{\mathbf{z}_i, \mathbf{z}_j \in \mathcal{Z}_{\text{P}}, i \neq j} \mathbf{z}_i^T\mathbf{z}_j + \frac{n_{\text{P}_{\text{L}}}}{n_{\text{P}}}\sum_{\mathbf{z}_i \in \mathcal{Z}_{\text{P}}} \mathbf{z}_i^T\mathbf{z}_i\right]$$

Plugging this back we get:

$$
\begin{aligned}
\mathbb{E}\left[\|\mu - \mu^*\|^2\right] &= \frac{1}{n_{P_L}^2}\left[\frac{n_{P_L}(n_{P_L}-1)}{n_P(n_P-1)}\sum_{\mathbf{z}_i,\mathbf{z}_j\in\mathcal{Z}_P, i\neq j}\mathbf{z}_i^T\mathbf{z}_j + \frac{n_{P_L}}{n_P}\sum_{\mathbf{z}_i\in\mathcal{Z}_P}\mathbf{z}_i^T\mathbf{z}_i\right] - \mu^{*T}\mu^* \\
&= \frac{n_P - n_{P_L}}{n_{P_L}(n_P-1)}\left[\frac{1}{n_P}\sum_{\mathbf{z}_i\in\mathcal{Z}_P}\mathbf{z}_i^T\mathbf{z}_i - \mu^{*T}\mu^*\right] \\
&= \left(1 + \frac{n_P - n_{P_L}}{n_{P_L}(n_P-1)}\right)\phi^*(\mathcal{Z}_P, \mu^*)
\end{aligned}
$$

This concludes the proof. ∎

### A.6.4  PROOF OF LEMMA 3

*Proof.* This result is a direct consequence of Lemma 3.3 from (Arthur & Vassilvitskii, 2007) and specializing to our case where we only have 1 uncovered cluster i.e. $t = u = 1$ and consequently the harmonic sum $H_t = 1$. ∎

### A.7 Additional Reproducibility Details

In this section we present more details on our experimental setup.

For all the experiments in Table 2, contrastive training is done using LARS optimizer (You et al., 2019), cosine annealing schedule with linear warm-up, batch size 1024, initial learning rate 1.2. We use a 128 dimensional projection layer $h_{\Gamma}(\cdot)$ composed of two linear layers with relu activation and batch normalization. We leverage Faiss (Johnson et al., 2019) for efficient implementation of PUPL. To ensure reproducibility, all experiments are run with deterministic cuDNN back-end and repeated 5 times with different random seeds and the confidence intervals are noted.

#### A.7.1 Positive Unlabeled Benchmark Datasets

Consistent with recent literature on PU Learning (Li et al., 2022; Chen et al., 2020a) we conduct our experiments on six benchmark datasets: STL-I, STL-II, CIFAR-I, CIFAR-II, FMNIST-I, and FMNIST-II, obtained via modifying STL-10 (Coates et al., 2011), CIFAR-10 (Krizhevsky et al., 2009), and Fashion MNIST (Xiao et al., 2017), respectively. The specific definitions of labels ("positive" vs "negative") are as follows:

- **FMNIST-I**: The labels "positive" correspond to the classes "1, 4, 7", while the labels "negative" correspond to the classes "0, 2, 3, 5, 6, 8, 9".
- **FMNIST-II**: The labels "positive" correspond to the classes "0, 2, 3, 5, 6, 8, 9", while the labels "negative" correspond to the classes "1, 4, 7".
- **CIFAR-I**: The labels "positive" correspond to the classes "0, 1, 8, 9", while the labels "negative" correspond to the classes "2, 3, 4, 5, 6, 7".
- **CIFAR-II**: The labels "positive" correspond to the classes "2, 3, 4, 5, 6, 7", while the labels "negative" correspond to the classes "0, 1, 8, 9".
- **STL-I**: The labels "positive" correspond to the classes "0, 2, 3, 8, 9", while the labels "negative" correspond to the classes "1, 4, 5, 6, 7".
- **STL-II**: The labels "positive" correspond to the classes "1, 4, 5, 6, 7", while the labels "negative" correspond to the classes "0, 2, 3, 8, 9".

#### A.7.2 Positive Unlabeled Baselines

Next, we describe the PU baselines used in Table 2:

- **Unbiased PU learning (UPU)** (Du Plessis et al., 2014): This method is based on unbiased risk estimation and incorporates cost-sensitivity.
- **Non-negative PU learning (NNPU)** (Kiryo et al., 2017): This approach utilizes non-negative risk estimation and incorporates cost-sensitivity. Suggested settings: $\beta = 0$ and $\gamma = 1.0$.
- **nnPU w Mixup** Zhang et al. (2017) : This cost-sensitive method combines the nnPU approach with the mixup technique. It performs separate mixing of positive instances and unlabeled ones.
- **SELF-PU** Chen et al. (2020d): This cost-sensitive method incorporates a self-supervision scheme. Suggested settings: $\alpha = 10.0$, $\beta = 0.3$, $\gamma = \frac{1}{16}$, Pace1 = 0.2, and Pace2 = 0.3.
- **Predictive Adversarial Networks (PAN)** (Hu et al., 2021): This method is based on GANs and specifically designed for PU learning. Suggested settings: $\lambda = 1e - 4$.
- **Variational PU learning (VPU)** (Chen et al., 2020a): This approach is based on the variational principle and is tailored for PU learning. The public code from net.9 was used for implementation. Suggested settings: $\alpha = 0.3$, $\beta \in \{1e - 4, 3e - 4, 1e - 3, \ldots, 1, 3\}$.
- **MIXPUL** (Wei et al., 2020): This method combines consistency regularization with the mixup technique for PU learning. The implementation utilizes the public code from net.10. Suggested settings: $\alpha = 1.0$, $\beta = 1.0$, $\eta = 1.0$.
- **Positive-Unlabeled Learning with effective Negative sample Selector PULNS** (Luo et al., 2021): This approach incorporates reinforcement learning for sample selection. We implemented a custom Python code with a 3-layer MLP selector, as suggested by the paper. Suggested settings: $\alpha = 1.0$ and $\beta \in \{0.4, 0.6, 0.8, 1.0\}$.
- **P³MIX-C/E** (Li et al., 2022): Denotes the heuristic mixup based approach.

### A.7.3 IMAGE AUGMENTATIONS FOR CONTRASTIVE TRAINING

We provide the details of transformations used to obtain the contrastive learning benchmarks in this paper for each datasets.

```
cifar_transform = transforms.Compose([
    transforms.RandomResizedCrop(input_shape),
    transforms.RandomHorizontalFlip(p=0.5),
    transforms.RandomApply([GaussianBlur([0.1, 2.0])], p=0.5),
    transforms.RandomApply(
    [transforms.ColorJitter(0.4, 0.4, 0.4, 0.1)], p=0.8),
    transforms.RandomGrayscale(p=0.2),
    transforms.ToTensor(),
    transforms.Normalize(mean=self.mean, std=self.std)])
```

```
fmnist_transform = transforms.Compose([
    transforms.RandomResizedCrop(input_shape),
    transforms.RandomApply(
    [transforms.ColorJitter(0.4, 0.4, 0.2, 0.1)], p=0.8),
    transforms.ToTensor(),
    transforms.Normalize(mean=self.mean, std=self.std)])
```

```
stl_transform = transforms.Compose([
    transforms.RandomHorizontalFlip(),
    transforms.RandomResizedCrop(size=96),
    transforms.RandomApply(
    [transforms.ColorJitter(0.5, 0.5, 0.5, 0.1)], p=0.8),
    transforms.RandomGrayscale(p=0.2),
    transforms.GaussianBlur(kernel_size=9),
    transforms.ToTensor(),
    transforms.Normalize((0.5,), (0.5,))])
```

```
imagenet_transform = transforms.Compose([
    transforms.RandomResizedCrop(224, interpolation=Image.BICUBIC),
    transforms.RandomHorizontalFlip(p=0.5),
    transforms.RandomApply(
    [transforms.ColorJitter(0.4, 0.4, 0.2, 0.1)], p=0.8),
    transforms.RandomGrayscale(p=0.2),
    transforms.RandomApply([GaussianBlur([0.1, 2.0])], p=0.5),
    Solarization(p=0.2),
    transforms.ToTensor(),
    transforms.Normalize(mean=self.mean, std=self.std)])
```

```
imagenet_transform = transforms.Compose([
    transforms.RandomResizedCrop(224, interpolation=Image.BICUBIC),
    transforms.RandomHorizontalFlip(p=0.5),
    transforms.RandomApply(
    [transforms.ColorJitter(0.4, 0.4, 0.2, 0.1)], p=0.8),
    transforms.RandomGrayscale(p=0.2),
    transforms.RandomApply([GaussianBlur([0.1, 2.0])], p=0.5),
    Solarization(p=0.2),
    transforms.ToTensor(),
    transforms.Normalize(mean=self.mean, std=self.std)])
```

### A.7.4 PYTORCH STYLE PSEUDO CODES

```
class SelfSupConLoss(nn.Module):
    """
    Self Supervised Contrastive Loss
    """
    def __init__(self, temperature: float = 0.5, reduction="mean"):
        super(SelfSupConLoss, self).__init__()
```

```python
 7        self.temperature = temperature
 8        self.cross_entropy = nn.CrossEntropyLoss(reduction=reduction)
 9
10    def forward(self, z: torch.Tensor, z_aug: torch.Tensor, *kwargs) ->
      torch.Tensor:
11        """
12        :param z: features
13        :param z_aug: augmentations
14        :return: loss value, scalar
15        """
16
17        batch_size, _ = z.shape
18        # project onto hypersphere
19        z = nn.functional.normalize(z, dim=1)
20        z_aug = nn.functional.normalize(z_aug, dim=1)
21
22        # calculate similarities block-wise
23        inner_pdt_00 = torch.einsum('nc,mc->nm', z, z) / self.temperature
24        inner_pdt_01 = torch.einsum('nc,mc->nm', z, z_aug) / self.temperature
25        inner_pdt_10 = torch.einsum("nc,mc->nm", z_aug, z) / self.temperature
26        inner_pdt_11 = torch.einsum('nc,mc->nm', z_aug, z_aug) / self.
      temperature
27
28        # remove similarities between same views of the same image
29        diag_mask = torch.eye(batch_size, device=z.device, dtype=torch.bool)
30        inner_pdt_00 = inner_pdt_00[~diag_mask].view(batch_size, -1)
31        inner_pdt_11 = inner_pdt_11[~diag_mask].view(batch_size, -1)
32
33        # concatenate blocks
34        inner_pdt_0100 = torch.cat([inner_pdt_01, inner_pdt_00], dim=1)
35        inner_pdt_1011 = torch.cat([inner_pdt_10, inner_pdt_11], dim=1)
36        logits = torch.cat([inner_pdt_0100, inner_pdt_1011], dim=0)
37
38        labels = torch.arange(batch_size, device=z.device, dtype=torch.long)
39        labels = labels.repeat(2)
40        loss = self.cross_entropy(logits, labels)
41
42        return loss
```

```python
 1 class SupConLoss(nn.Module):
 2     """
 3     Supervised Contrastive Loss
 4     """
 5     def __init__(self, temperature: float = 0.5, reduction="mean"):
 6        super(SupConLoss, self).__init__()
 7        self.temperature = temperature
 8        self.reduction = reduction
 9
10     def forward(self, z: torch.Tensor, z_aug: torch.Tensor, labels: torch.
       Tensor, *kwargs) -> torch.Tensor:
11        """
12
13        :param z: features => bs * shape
14        :param z_aug: augmentations => bs * shape
15        :param labels: ground truth labels of size => bs
16        :return: loss value => scalar
17        """
18        batch_size, _ = z.shape
19
20        # project onto hypersphere
21        z = nn.functional.normalize(z, dim=1)
22        z_aug = nn.functional.normalize(z_aug, dim=1)
23
24        # calculate similarities block-wise
```

```
25    inner_pdt_00 = torch.einsum('nc,mc->nm', z, z) / self.temperature
26    inner_pdt_01 = torch.einsum('nc,mc->nm', z, z_aug) / self.temperature
27    inner_pdt_10 = torch.einsum("nc,mc->nm", z_aug, z) / self.temperature
28    inner_pdt_11 = torch.einsum('nc,mc->nm', z_aug, z_aug) / self.
      temperature
29
30    # concatenate blocks
31    inner_pdt_0001 = torch.cat([inner_pdt_00, inner_pdt_01], dim=1)
32    inner_pdt_1011 = torch.cat([inner_pdt_10, inner_pdt_11], dim=1)
33    inner_pdt_mtx = torch.cat([inner_pdt_0001, inner_pdt_1011], dim=0)
34
35    max_inner_pdt, _ = torch.max(inner_pdt_mtx, dim=1, keepdim=True)
36    inner_pdt_mtx = inner_pdt_mtx - max_inner_pdt.detach()  # for
      numerical stability
37
38    # compute negative log-likelihoods
39    nll_mtx = torch.exp(inner_pdt_mtx)
40    # mask out self contrast
41    diag_mask = torch.ones_like(inner_pdt_mtx, device=z.device, dtype=
      torch.bool).fill_diagonal_(0)
42    nll_mtx = nll_mtx * diag_mask
43    nll_mtx /= torch.sum(nll_mtx, dim=1, keepdim=True)
44    nll_mtx[nll_mtx != 0] = - torch.log(nll_mtx[nll_mtx != 0])
45
46    # mask out contributions from samples not from same class as i
47    mask_label = torch.unsqueeze(labels, dim=-1)
48    eq_mask = torch.eq(mask_label, torch.t(mask_label))
49    eq_mask = torch.tile(eq_mask, (2, 2))
50    similarity_scores = nll_mtx * eq_mask
51
52    # compute the loss -by averaging over multiple positives
53    loss = similarity_scores.sum(dim=1) / (eq_mask.sum(dim=1) - 1)
54    if self.reduction == 'mean':
55      loss = torch.mean(loss)
56    return loss
```

```
1  class PUConLoss(nn.Module):
2    """
3      Positive Unlabeled Contrastive Loss
4    """
5
6    def __init__(self, temperature: float = 0.5):
7      super(PUConLoss, self).__init__()
8      # per sample unsup and sup loss : since reduction is None
9      self.sscl = SelfSupConLoss(temperature=temperature, reduction='none')
10     self.scl = SupConLoss(temperature=temperature, reduction='none')
11
12   def forward(self, z: torch.Tensor, z_aug: torch.Tensor, labels: torch.
     Tensor, *kwargs) -> torch.Tensor:
13     """
14         @param z: Anchor
15         @param z_aug: Mirror
16         @param labels: annotations
17     """
18     # get per sample sup and unsup loss
19     sup_loss = self.scl(z=z, z_aug=z_aug, labels=labels)
20     unsup_loss = self.sscl(z=z, z_aug=z_aug)
21
22     # label for M-viewed batch with M=2
23     labels = labels.repeat(2).to(z.device)
24
25     # get the indices of P and  U samples in the multi-viewed batch
26     p_ix = torch.where(labels == 1)[0]
27     u_ix = torch.where(labels == 0)[0]
```

```python
28
29      # if no positive labeled it is simply SelfSupConLoss
30      num_labeled = len(p_ix)
31      if num_labeled == 0:
32        return torch.mean(unsup_loss)
33
34      # compute expected similarity
35      # ------------------------
36      risk_p = sup_loss[p_ix]
37      risk_u = unsup_loss[u_ix]
38
39      loss = torch.cat([risk_p, risk_u], dim=0)
40      return torch.mean(loss)
```

```python
1  def puPL(x_PU, y_PU, num_clusters=2):
2  """
3  puPL: Positive Unlabeled Pseudo Labeling
4  """
5      p_ix = y_PU==1
6      u_ix = y_PU==0
7      x_P = x_PU[p_ix]
8      x_U = x_PU[u_ix]
9
10     ## Initialize Cluster Centers ##
11     # Compute the mean of x_P as the first centroid
12     centroid_P = np.mean(x_P, axis=0)
13     # Next, use K-means++ to choose the second center from x_U
14     kmeans_pp = KMeans(n_clusters=1, init=np.array([centroid_2]))
15     kmeans_pp.fit(x_U)
16     centroid_N = kmeans_pp.cluster_centers_[0]
17     # Initialize the centroids with the computed values
18     centroids = np.array([centroid_N, centroid_P])
19
20     ## Perform K-means iterations ##
21     kmeans = KMeans(n_clusters=num_clusters, init=centroids)
22     kmeans.fit(np.concatenate((x_U, x_P), axis=0))
23
24     labels = kmeans.labels_
25     data = np.concatenate((x_U, x_P), axis=0)
26
27     return labels, data
```

