# OpenReview forum: "Contrastive Positive Unlabeled Learning"
_ICLR.cc/2024/Conference — Submitted to ICLR 2024_

### Official Review · Reviewer_FfAt · 2023-10-29

**Soundness:** 3 good
**Presentation:** 3 good
**Contribution:** 3 good
**Rating:** 6
**Confidence:** 3

**Summary:**

The paper proposes a new approach for positive unlabeled learning, which involves two steps: contrastive representation learning and psudo labeling. The proposed method is practical, theoretical analysis is provided, and good experiment results are shown. However, there are some concerns in motivation and experiments.

**Strengths:**

1. The proposed method combing contrastive learning and psudo-labeling makes sense and seems to be practical.
2. The paper is well written and one can easily follow.
3. Theoretical analysis is provided for different components of the method.
4. Good experiment results are shown.

**Weaknesses:**

1. The paper was motivated by applications in recommender systems, drug, and others, but is only evaluated on image classification dataset. In fact, the proposed method relies on self-supervised constrastive learning, which mostly works well for images (or texts), as you can easily augment the data. However, for recommender systems and drug applications, there is no easy way to perform augmentation. It would be better to properly motivate the method to avoid overclaim.
2. The novelty of the method seems to be limited. Using contrastive learning for positive unlabeled learning is not new, e.g. it is considered in[1]. Psudo-labeling with kmeans using positive examples for positive unlabeled learning is also considered literature[2]. It seems to be the novelty of the method is combing the two existing methods.
3. Experiment evaluation can be improved to make the work more solid.

[1] Chuang, Ching-Yao, et al. "Debiased contrastive learning." Advances in neural information processing systems 33 (2020): 8765-8775.

[2] Liu, Qinchao, et al. "A novel k-means clustering algorithm based on positive examples and careful seeding." 2010 International Conference on Computational and Information Sciences. IEEE, 2010.

**Questions:**

1. How does the each component of the method compare to existing methods in literature? Specifically, how does contrastive learning component compare to the one in[1] and the kmeans component compare to the one in [2]? These are only two examples and I believe there should be more alternatives in literature.

2. For image classification task, the most popular benchmark dataset is imagenet. Can you also evaluate the method on imagenet, e.g. by sampling 100 class following[1]?

3. How many positive examples are used in each dataset? How does the method compare to the baselines when the number of positive examples vary? Can you show a plot about this?

[1] Chuang, Ching-Yao, et al. "Debiased contrastive learning." Advances in neural information processing systems 33 (2020): 8765-8775.

[2] Liu, Qinchao, et al. "A novel k-means clustering algorithm based on positive examples and careful seeding." 2010 International Conference on Computational and Information Sciences. IEEE, 2010.

---

> ### Author Response · Authors · 2023-11-23
>
> We thank you for your valuable and constructive feedback. Below we respond to each concern in detail.
>
> ( W1 ) **recommendation experiments**
>
> You've rightly pointed out the significance of designing label-preserving augmentations in contrastive learning (CL), a task that can be challenging in domains where such augmentations are not readily obtainable. This limitation, however, has not deterred the exploration of CL across various domains, buoyed by its success in fields like vision and text, and notably in multi-modal settings such as CLIP. The active pursuit of domain-specific augmentations is a testament to the adaptability and potential of CL. In diverse areas—from graph data, where augmentations are made by randomly dropping edges, to ranking problems that integrate random noise or mask entries in tabular data—innovative techniques are being developed. These advancements extend to recommendation systems, audio processing, and beyond, highlighting the versatility of CL in handling various data modalities to name a few e.g.[1-5].
>
> While extending contrastive learning (CL) across various data modalities, is a fascinating and burgeoning field of research—emphasizing the need for domain-specific augmentations and thoughtful design choices—it's important to note the core objective of this paper.
>
> In this paper, as a methodological exploration of contrastive learning (CL), we primarily concentrate on vision experiments. This focus is deliberate and sufficient for our objectives (and consistent with the large body of work in both contrastive learning and PU Learning ). As a methods paper, our primary goal is to elucidate and validate the proposed approaches and techniques within the realm of CL. By using vision data, which offers a rich and well-understood context, we can clearly demonstrate the efficacy and versatility of our methods. Additionally, vision experiments provide a tangible and accessible means for the broader research community to evaluate and replicate our findings. While we acknowledge the potential of extending our methods to other data modalities, this paper intentionally narrows its scope to provide a deep and thorough exploration of contrastive representation learning from PU supervision and develop that into simple and practical PU learning solution with superior empirical performance, without needing additional knowledge like class prior. Through extensive experiments and intuitive theoretical justifications we show that our approach uniquely stays effective even with extremely limited labels, unlike prior PU methods. We conduct experiments on multiple subsets of CIFAR, ImageNet, FMNIST, STL-10, including comparing our method with a large number of baselines consistent with the experimental setups of recent PU Learning literature.
>
> This exploration represents an exciting avenue for future research in PU learning, inviting the development of innovative approaches and domain-specific augmentations that can unlock the benefits of CL in less conventional or more complex data environments. Our work lays a foundational framework, opening the door for future studies to adapt and expand upon our methodologies, thereby broadening the applicability and impact of CL across diverse and challenging domains.
>
> [1] Majmundar et.al. Met: Masked encoding for tabular data.
> [2] You et.al. Graph Contrastive Learning with Augmentations.
> [3] Hou et.al. PRETRAINED DEEP MODELS OUTPERFORM GBDTS IN LEARNING-TO-RANK UNDER LABEL SCARCITY.
> [4] Guo et.al. SimCSE: Simple Contrastive Learning of Sentence Embeddings.
> [5] Liu et.al. Contrastive Learning for Recommender System.

---

> ### Author Response · Authors · 2023-11-23
>
> ( W2 ) **Contribution**
>
> As we have discussed in great depth (Introduction, ), existing PU Learning  methods suffer from two major issues:
>
> **Class Prior Estimate :** The success of these estimators hinges upon the knowledge of the oracle class prior $\pi_p^\ast$ for their success. It is immediate to see that an error is class prior estimate $\|\hat{\pi}_p - \pi_p^\ast\|_2 \leq \xi$ results in an estimation bias $\sim O(\xi)$ that can result in poor generalization, slower convergence or both. Unfortunately however in practical settings the class prior is not available and often estimating it with high accuracy using some MPE algorithm can be quite costly.
>
> **Low Supervision Regime :** When available supervision is limited i.e. when $\gamma$ is small, the estimates  from increased variance resulting in increased variance leading to poor performance.
>
> To this end, the primary aim of this work is to {\em develop a parameter-free approach that facilitates Positive Unlabeled (PU) learning, even in scenarios where the availability of labeled examples is limited, without requiring any additional side information, such as class prior.
>
> Our proposed PU Learning framework, involves two key steps:
>
> (a)    Learning a cluster preserving representation manifold i.e. a feature space that preserves the underlying clusters by mapping semantically similar examples close to each other}.
>
> (b) Assign pseudo-labels to the unlabeled examples by exploiting the geometry of the representation manifold learnt in the previous step}. These pseudo-labels are then used to train the downstream linear classifier using ordinary supervised objective e.g. cross-entropy (CE). This enables PU learning even in settings where only a few labeled examples are available, while obviating the need for any additional side information such as class prior.
>
> One way to obtain a representation manifold where the embeddings (features) exhibit linear separability is via contrastive learning (https://arxiv.org/pdf/2302.07920.pdf ). To the best of our knowledge, we make the first attempt to investigate the value of contrastive approach in the PU Learning setting. However, this is not a trivial plug-and-play – we show that standard self-supervised contrastive loss dubbed ssCL is unable to leverage PU supervision. While, naive adaptation of the supervised contrastive loss to the PU setting dubbed sCL-PU suffers from statistical bias. To this end, we adopt a simple PU specific modification of the standard self-supervised contrastive objective to take into account the available weak supervision in form of labeled positives resulting in significantly improved representations compared to the self-supervised counterpart.
>
> It's important to note that the downstream classifier must also be trained using Positive-Unlabeled (PU) data unlike standard semi-supervised contrastive learning setting where fully supervised data (even if small) is available.
> In spite of the separability properties of the contrastive representation manifold, supervised loss e.g. CE are ineffective in PU setting, making downstream classification non-trivial. One idea would be to use a PU specific loss function to train the classifier. However, even over such separable feature space, existing cost-sensitive losses suffer from the issues discussed earlier, especially in the low-supervision regime (handful labeled positives). Instead, we propose to use some static clustering algorithm to obtain pseudo labels from the contrastive representations.
>
> In summary, this is not a paper on a new contrastive loss, nor is this a paper on clustering. This paper tackles PU Learning --- a noisy label problem where existing approaches are inadequate especially in low-data-regime. To this end, the aim of this paper is to propose a modern framework that can alleviate these issues.  To the best of our knowledge, this paper represents the first work tailoring contrastive learning specifically to the PU setting and introducing deep insights about design choices in PU setting e.g. incorporating available positives in an unbiased way (compared to MCL where a cvx combination of sCL and ssCL is considered), subsequently training downstream classifier using pseudo-labels.

---

> ### Author Response · Authors · 2023-11-23
>
> ( W2, Q1) **Comparison with DCL**
> It is important to note that DCL is an unsupervised CL objective and does not incorporate any form of supervision.
>
> In unsupervised setting i.e. ssCL pulls together self augmentations while pushing the anchor farther from all other samples in the batch (captured in the denominator). In fact we can rewrite the denominator $Z = <x_i, x_{a(i)}> + \sum_{j \neq i, a(i)} <x_i, x_j>$ i.e. pos pair sum + neg pairs sum. However, DCL argues that since not all other examples in the batch are truly negative to the anchor, this introduces sampling bias. The authors take inspiration from PU Learning to rescale (debias) $Z$ , see Sec A.5.4 for more detail.
>
> While, reducing the sampling bias improves over vanilla ssCL - DCL is unable to leverage the additional supervision. Empirically, we observe that gains from incorporating supervision (puCL) is far more significant compared to correcting the sampling bias (DCL). See Fig 11.
>
> Additionally, we find that DCL is very sensitive to the choice of the hyperaparameter and there is no theoretical suggestion available to choose the right parameter making it difficult to adopt in practical settings. Intuitively, the optimal hyperparam should be close to the class prior -- however, since we do not have class prior knowledge, the adaptation of DCL to PU setting does not seem like a good idea.
>
> ( W2, Q2) **Comparison with clustering methods**
> Indeed, we do not claim any novelty about discovering a new clustering algorithm. Our goal is simply to leverage the cluster preserving property of the representation manifold to learn a downstream classifier. In this regard, any off-the-shelf clustering algorithm is sufficient. In this paper as we have mentioned multiple times, we take a straight-forward adaptation of semi-supervised kMeans++ [1]. However, we are also happy to cite the paper you pointed.
>
> ( W3 ) **More Experiments**
> During the rebuttal period, we have significantly enhanced the enhanced the paper. In summary here are the new figs / results added based on reviews:
>
> (a) Fig 1: ImageNet subset - i. Compare with existing PU methods in terms of generalization.  ii. Convergence comparison of CL methods.  iii\. embedding visualization.
>
> (b) Fig 2, 8: Aligning points on hypercube: To provide more intuition about the underpinnings, we discuss a energy configuration argument to provide intuitive explanations about optimal point configurations of various CL discussed. Also see Section A.5.2.
>
> (c) Fig 3: Linear Probing Choices: Given pretrained embedding our goal is now to train a downstream linear model. In this experiment we take puCL($\gamma$) pretrained encoder (frozen) and train a linear classifier for downstream inference. In particular, we evaluate several popular SOTA PU Learning methods along with the proposed pseudo-labeling based approach.
> Our findings are particularly noteworthy in the context of low-data regimes. While traditional PU learning methods often struggle to maintain performance with limited data, our approach consistently demonstrates robust effectiveness.
>
> (d) Fig 6: Comparison of CL objective across $\gamma$ on imagenet and cifar subset.
>
> (e) Fig 9: To study the scenario when p(x) doesn not contain information about p(y|x) we construct three PU subsets from CIFAR:ahrd, medium, easy. see Section A.5.2 for details.
>
> (f) Comparison with Parametric CL: We compare with DCL and MCL two CL objectives that try to modify the infoNCE loss from inferred weak supervision. See Section A.5.4 for details
>
> (g) We also provide evidence (both theoretical Theorem 4 and empirical : Fig 1, 10 that by incorporating PU supervision judiciously puCL not only enjoys better generalization - it also converges faster than ssCL.
>
> ( Q2 ) **ImageNet Subset**
> We include a new exp: ImageNet-II: {ImageWoof vs ImageNette} - two subsets of ImageNet-1k widely used in noisy label learning research \href{https://github.com/fastai/imagenette}{https://github.com/fastai/imagenette}. Our experimental findings from smaller datasets (CIFAR small images) translate to ImageNet as well (Fig 1, 3, 5, 6, 9). In fact, we note that in ImageNet the gains over previous SOTA e.g. P3Mix-E, vPU, nnPU+MixUp is more prominent even in high supervision regime. On the other hand, even for ImageNet, for limited supervision PU methods fail, whereas our proposed approach remains uniquely effective.
>
> ( Q3 ) **Positive Example**
> Throughout the paper, we have studied the effect of available supervision across different components of the framework. We use $\gamma = \frac{n_P}{n_U}$ as a measure of available supervision. Most of our theoretical justifications, and experiments (Theorem 1, 2, Lemma 1, Fig 1-15 Table1, 2) consider the training behavior under different supervision levels.
>
> [1] Jordan Yoder and Carey E Priebe. Semi-supervised k-means++. Journal of Statistical Computation
> and Simulation, 87(13):2597–2608, 2017

---

### Official Review · Reviewer_sTW4 · 2023-10-30

**Soundness:** 4 excellent
**Presentation:** 3 good
**Contribution:** 3 good
**Rating:** 8
**Confidence:** 4

**Summary:**

This paper focuses on the positive unlabeled classification problem. Unlike in the usual binary classification setup, our training data does not have any negatives. Instead, we have positives and unlabeled samples. There are two issues in the previous PU learning approaches. One is that most of them assume access to the underlying class prior, or if we do not have this knowledge, we need to estimate the class prior. The second issue is that previous approaches tend to perform poorly with few positive training data. This paper proposes a new method based on contrastive learning which improves over the self-supervised baseline empirically and theoretically. After the representation learning step, the paper further propose a method called PUPL, which is a pseudo-labeling clustering method with theoretical guarantees on when it can recover the true underlying labels. As a framework, the paper finally propose to train a classifier based on the clustering results with pseudo labels.

**Strengths:**

- The statistical properties of the proposed method are studied: it is unbiased but also has smaller variance compared to the self supervised contrastive learning (ssCL) counterpart. These results are intuitive, since we are utilizing the additional positive label information that is not used in the ssCL algorithm.
- Furthermore, a clustering algorithm is proposed to prepare pseudo labels for classifier training.
- The experimental results show comparison with many baselines, and also show the proposed method is often the best performing method.
- Code and jupyter notebook are provided in the supplementary link.

**Weaknesses:**

- I am wondering how strong the separable assumption discussed towards the end of Section 3 is. For example, it would be interesting to empirically check if the method's performance will degrade and class-prior based PU methods (with oracle class prior) become better when the two Gaussians approach each other in Figure 10 in the Appendix (corresponding to the case that the classes do not form a cluster).
- The experiments that use previous PU methods after representation learning with puCL (instead of the proposed puPL) are interesting and important as an ablation study (shown in Table 1). However, the main baseline is nnPU, which is a method that is motivated to learn a classifier directly from input space and is not meant to be used for a linear classifier. It makes me wonder if there are other suitable baselines here, e.g., the other ones used in the other experiments. (However, if the experiments here are showing negative training loss even with a linear model without the non-negative component, then I feel it may be fine to use this as a comparison here.)
- Some discussions about the relationship with other weakly supervised contrastive learning papers, such as "PiCO: Contrastive label disambiguation for partial label learning" (ICLR 2022) or "ComCo: Complementary supervised contrastive learning for complementary label learning" (Neural Networks, 2024) would be helpful. I understand that the type of weak label is different since this paper focuses on PU learning, not partial labels/complementary labels. However, it would make the contributions more clear if we can see that similar ideas have not been proposed before in papers that worked on weak supervision + contrastive learning.

**Questions:**

Other than the points I raised in the Weaknesses section, I would like to ask some minor questions and list some minor suggestions.

- The class prior in page 1 is defined as $\pi_p = p( y = 1 \mid x)$ and this is also used in the appendix p22 (Definition 2). I am wondering if this should be $\pi_p = p(y=1)$?
- The legend of figures is quite small. For example, I cannot read the legend on Figure 11. Some of the colors seem similar to my eye and it is hard to distinguish between them.
- In Algorithm 1: In my understanding, the training is based on (pseudo-)PN labels. Would it be more accurate to denote it as "C. Train PN Classifier" in the 3rd step? While the term "PU Classifier" is not incorrect, this adjustment might offer a more precise representation. Nevertheless, this is a minor comment, and I respect the authors' choice on this matter.

---

> ### Author Response · Authors · 2023-11-23
>
> We thank you for your valuable and constructive feedback. Below we respond to each concern in detail.
>
> **Separability**
> Note that, without separability learning a decision boundary is generally infeasible - even for fully supervised loss since there is no separating hyperplane. However, initially X was not linearly separable -- the goal of CL is to find a mapping such that they encoder(x) becomes linearly separable. If the representation manifold is not separable - there is no hope for a perfect linear classification.
>
> Nonetheless, we perform this experiment based on your suggestion - see Fig 13, where the gaussian components are chosen to be overlapping. We note that even in this setting the puPL decision boundary is aligned with supervised CE.
>
> **Additional PU LP Baselines**
> This is a valid point. Based on your suggestion we have performed LP experiments on 2 datasets as included in Fig 3.
>
> **Connection to PLL**
> Indeed, PLL is related to weakly supervised learning, learning with noisy label, robust optimization, PU Learning etc. It is common to see similar foundational ideas applied in different problem settings. While related in essence, NLL setting is a sufficiently different problem than PU Learning and thus need to be studied as different problems.
>
> Closest to our work is mCL where a cvx combination is used to combine ssCL and sCL to adapt for label noise setting please see Sec A.5.4, Fig 10 where we have discussed this in greater depth.
>
>
> **\pi_p**  $\pi_p = p(y|x)$ should be a dataset dependent parameter strictly speaking. However, it is common to use the notations interchangeably when the underlying distribution is assumed.  However, we will make our notation consistent following your suggestion.
>
> **Legend** We will take care of these minor issues in the final version.
>
> **Algo 1** This is a great suggestion -- as PU learning is misleading once we have a pseudo-labeled dataset. We modified the manuscript to reflect this change.

---

### Official Review · Reviewer_Yt2Y · 2023-10-31

**Soundness:** 2 fair
**Presentation:** 2 fair
**Contribution:** 2 fair
**Rating:** 3
**Confidence:** 4

**Summary:**

This paper introduces a framework for Positive Unlabeled (PU) learning, targeting the limitations of existing PU methods that require additional class prior knowledge and struggle in low-data scenarios. It utilizes pretext-invariant representation learning to create a feature space where unlabeled examples are pseudo-labeled based on the cluster assumption. The authors show that the proposed framework is particularly effective in scenarios with a lot of labeled data. Empirical results demonstrate the effectiveness of the proposed method.

**Strengths:**

+ Some theoretical analyses are provided. In a PU setting, the authors have shown that the loss based on treating the unlabeled training examples as pseudo-negative instances is biased. The authors have also analyzed the consistency of their method.
+ Clarity of Presentation. The paper is well-written and easy to follow.

**Weaknesses:**

+ The theoretical advantage is not clear. Firstly, the authors propose that the proposed method is not sensitive to the estimated class prior. It seems to be a more general method. However, to make contrastive learning reliable, it relies on another assumption related to the data generative process. Specifically, it requires the distribution of data $P(X)$ containing information about $P(Y|X)$. A clustering assumption is assumed in the paper which is an instance of the assumption. This actually cannot always hold in different datasets. For example, it can be hard to generally apply the proposed method to causal datasets as the cluster assumption cannot be satisfied. Moreover, it is unknown how to use contrastive learning on non-image datasets with theoretical guarantees such as UCI datasets. I believe existing PU methods with theoretical guarantees do not suffer from these issues and can be generally applied to other non-image datasets.
+ Minor Empirical Improvement. The empirical improvements shown in the paper, while present, are relatively minor. This raises questions about the practical significance of the proposed method. It would be beneficial if the authors could demonstrate more substantial improvements or discuss scenarios where their method is expected to have a more pronounced impact.

**Questions:**

- How does the proposed method handle scenarios where the distribution of data $P(X)$ does not contain adequate information about  $P(Y|X)$, particularly in causal datasets where the clustering assumption may not hold?
- Can you provide insights or theoretical justification for the applicability of your contrastive learning approach to non-image datasets and causal datasets (could refer https://pl.is.tue.mpg.de/p/causal-anticausal/) where clustering assumptions may not be valid?
- Given the relatively minor empirical improvements reported, could you elaborate on specific scenarios or types of datasets where the proposed method is expected to yield more significant advantages?
- Could you discuss how the proposed method compares in terms of practical utility and significance against existing Positive and Unlabeled (PU) learning methods, which have theoretical guarantees and broader applicability?
- Are there plans to test the proposed method on large-scale, real-world datasets, particularly high-dimensional ones like image datasets, to evaluate its performance and generalizability in more complex scenarios?
- How do you anticipate the proposed method would perform on such datasets, and what are the potential challenges you foresee in these environments?

---

> ### Author Response · Authors · 2023-11-23
>
> We thank you for your valuable and constructive feedback. Below we respond to each concern in detail.
>
> * ( W1, Q1 ) **p(X) containing information about P(y|x) :**
>
> This is an excellent question. Indeed, an important underlying assumption in unsupervised learning is that the features  contains information about the underlying label. Indeed, if $p(x)$ has no information about $p(y|x)$, no unsupervised representation learning method e.g. ssCL can hope to learn cluster-preserving representations.
> However, in *fully supervised settings*, since semantic annotations are available, it is possible to find a representation space ( given model has capacity ) where semantically dissimilar objects are grouped together based on labels i.e. clustered based on semantic annotations via supervised objectives e.g. CE. While, it is important to note that such models would be prone to over-fitting and might generalize poorly to unseen data. Since fully supervised contrastive learning objectives sCL~\citep{khosla2020supervised} use semantic annotations to guide the contrastive training, it can also be effective in such scenarios as it is trained on both semantic similarity and annotation.
>
> In the **PU learning setting**, puCL behaves in a similar way. It incorporates both semantic similarity (via pulling self augmentations together) and semantic annotation (via pulling together labeled positives together). Intuitively, by interpolating between supervised and unsupervised contrastive objectives, puCL favors representations where both semantically similar (feature) examples are grouped together along with all the labeled positives (annotations) are grouped together.
>
> **Arranging points on unit hypercube:**  To further understand the behavior of interpolating between semantic annotation (labels) and semantic similarity (feature) -
> Consider 1D feature space $x \in R$, e.g., $x_i = 1$ if shape: triangle, $x_i = 0$ if shape: circle . However, the labels are $y_i=1$ if color: blue and $y_i=1$ if color: red i.e $p(x)$ contains no information about $p(y | x)$. Fig 7 shows different configurations of arranging these points on the vertices of unit hypercube $H \in R^2$. It is easy to see that:
>
> **Unsupervised objectives** e.g. ssCL only rely on semantic similarity (feature) to learn embeddings, implying they attain minimum loss configuration when semantically similar objects $x_i = x_j$ are placed close to each other (neighboring vertices on $H^2$) since this minimizes the inner product between representations of similar examples.
>
> **Supervised objectives** e.g. CE on the other hand, updates the parameters such that the logits match the label. Thus purely supervised objectives attain minimum loss when objects sharing same annotation are placed next to each other ( Fig7(b) ).
>
> On the other hand, **puCL** interpolates between the supervised and unsupervised objective. Simply put, by incorporating additional positives it aims at learning representations that preserves both semantic similarity and annotation consistency.
> Thus, the minimum loss configurations are attained at the intersection of the minimum point configurations of ssCL and fully supervised sCL ( Fig 7 (c) ).
>
> **Experiment** To empirically verify this phenomenon - we train a ResNet-18 on three CIFAR subsets carefully crafted to simulate this phenomenon. In particular, we use CIFAR-hard (airplane, cat) vs (bird, dog), CIFAR-easy (airplane, bird) vs (cat,  dog) and CIFAR-medium (airplane, cat, dog) vs bird. Note that,  airplane and bird are semantically similar, also dog-cat are semantically closer to each other. Our experimental findings are reported in Fig 8. In summary, we observe that while ssCL is completely blind to supervision signals; given enough labels -- puCL}is able to leverage the available positives to group the samples labeled positive together. Since we are in binary setting, being able to cluster positives together automatically solves the downstream P vs N classification problem as well.
>
> We provide detailed discussion in Section A.5.2 (also see Fig 2, 7, 8).

---

> > ### Author Response · Authors · 2023-11-23
> >
> > * ( W1, Q2 ) **Extension to different data modalities**
> >
> > You've rightly pointed out the significance of designing label-preserving augmentations in contrastive learning (CL), a task that can be challenging in domains where such augmentations are not readily obtainable. This limitation, however, has not deterred the exploration of CL across various domains, buoyed by its success in fields like vision and text, and notably in multi-modal settings such as CLIP. The active pursuit of domain-specific augmentations is a testament to the adaptability and potential of CL. In diverse areas—from graph data, where augmentations are made by randomly dropping edges, to ranking problems that integrate random noise or mask entries in tabular data—innovative techniques are being developed. These advancements extend to recommendation systems, audio processing, and beyond, highlighting the versatility of CL in handling various data modalities to name a few e.g.[1-5].
> >
> > While extending contrastive learning (CL) across various data modalities, including causal datasets, is a fascinating and burgeoning field of research—emphasizing the need for domain-specific augmentations and thoughtful design choices—it's important to note the core objective of this paper.
> >
> > In this paper, as a methodological exploration of contrastive learning (CL), we primarily concentrate on vision experiments. This focus is deliberate and sufficient for our objectives (and consistent with the large body of work in both contrastive learning and PU Learning ). As a methods paper, our primary goal is to elucidate and validate the proposed approaches and techniques within the realm of CL. By using vision data, which offers a rich and well-understood context, we can clearly demonstrate the efficacy and versatility of our methods. Additionally, vision experiments provide a tangible and accessible means for the broader research community to evaluate and replicate our findings. While we acknowledge the potential of extending our methods to other data modalities, this paper intentionally narrows its scope to provide a deep and thorough exploration of contrastive representation learning from PU supervision and develop that into **simple and practical PU learning solution with superior empirical performance, without needing additional knowledge like class prior**. Through extensive experiments and intuitive theoretical justifications we show that our approach uniquely stays effective even with extremely limited labels, unlike prior PU methods. We conduct experiments on multiple subsets of CIFAR, ImageNet, FMNIST, STL-10, including comparing our method with a large number of baselines consistent with the experimental setups of recent PU Learning literature.
> >
> > This exploration represents an exciting avenue for future research in PU learning, inviting the development of innovative approaches and domain-specific augmentations that can unlock the benefits of CL in less conventional or more complex data environments. Our work lays a foundational framework, opening the door for future studies to adapt and expand upon our methodologies, thereby broadening the applicability and impact of CL across diverse and challenging domains.
> >
> > [1] Majmundar et.al. Met: Masked encoding for tabular data.
> > [2] You et.al. Graph Contrastive Learning with Augmentations.
> > [3] Hou et.al. PRETRAINED DEEP MODELS OUTPERFORM GBDTS IN LEARNING-TO-RANK UNDER LABEL SCARCITY.
> > [4] Guo et.al. SimCSE: Simple Contrastive Learning of Sentence Embeddings.
> > [5] Liu et.al. Contrastive Learning for Recommender System.

---

> ### Author Response · Authors · 2023-11-23
>
> * ( W3, Q3, Q4 ) **Empirical Benefit and Settings where it is most beneficial over existing PU methods**
> As discussed in the paper, due to the unavailability of negative examples, statistically consistent unbiased risk estimation is generally infeasible, without imposing strong structural assumptions on p(x). In fact, we show that no robust ERM estimator can solve the equivalent class-dependent label noise problem reliably unless certain dataset conditions $\gamma > 2\pi_p -1$ are met (see Section A.3.2, Lemma 1).
>
> Remarkably, SOTA cost-sensitive PU learning algorithms tackle this by forming an unbiased estimate of the true risk from PU data by assuming additional knowledge of the true class prior $\pi_p = p(y=1 |x)$.
> The unbiased estimator dubbed uPU of the true risk from PU data is given as:
>
> $$
> \hat{R}_{pu}(v) = \pi_p \hat{R}_p^+(v) + \textcolor{blue}{\Bigg[\hat{R}_u^-(v) - \pi_p \hat{R}_p^-(v)\Bigg]}
> $$
> As discussed before, we identify two main issues related to these cost-sensitive estimators:
>
> **Class Prior Estimate :** The success of these estimators hinges upon the knowledge of the oracle class prior $\pi_p^\ast$ for their success. It is immediate to see that an error is class prior estimate $\|\hat{\pi}_p - \pi_p^\ast\|_2 \leq \xi$ results in an estimation bias $\sim O(\xi)$ that can result in poor generalization, slower convergence or both. Our experiments (Fig 14) suggest that even small approximation error in estimating the class prior can lead to notable degradation in the overall performance of the estimators. Unfortunately however in practical settings the class prior is not available and often estimating it with high accuracy using some MPE algorithm can be quite costly.
>
> **Low Supervision Regime :**  While these estimators are significantly more robust than the vanilla supervised approach, our experiments (Fig 3, 12, 13 ) suggest that they might produce decision boundaries that are not closely aligned with the true decision boundary especially as $\gamma$ becomes smaller. Note that, when available supervision is limited i.e. when $\gamma$ is small, the estimates $\hat{R}_p^+$ and $\hat{R}_p^-$ suffer from increased variance resulting in increase variance of the overall estimator $\sim O(\frac{1}{n_P})$. For sufficiently small $\gamma$ these estimators are likely result in poor performance due to large variance.
>
> This work alleviates both these issues by incorporating both semantic similarity (features) as well as clean supervision (labeled positives) to learn a representation space that fosters linear separability. Further, by leveraging the cluster-preserving properties of the embedding space we propose to pseudo-label the representations via some static clustering algorithm e.g. kMeans. In particular, we adopt a semi-supervised  variant of kMeans++ that uses labeled positives for better initialization.
>
> Further, by incorporating available label information, puCL is also able to operate in settings where p(x) does not contain enough information about p(y|x), unlike purely unsupervised objective e.g. ssCL.
>
> On the other hand, when supervision is limited but p(x) contains information about p(y|x), by exploiting the semantic similarity (via feature similarity) puCL is able to enable learning even in label scant situations.
>
> Our experiments suggest, our overall representation learning followed by learning a classifier via pseudo labeling is an effective approach. Our method enjoys superior generalization performance compared to previous SOTA PU Learning algorithms overall. In vision benchmarks (consistent with the experimental setup of prior works as mentioned in Experiments Section) our approach outperforms previous SOTA $\sim 2\%$ averaged over six benchmark datasets. Further, we observe similar performance improvement on other datasets we used e,g, ImageNet subsets, CIFAR subsets (Fig 1, Table 2, Figure 15).
>
> More importantly, we see that when available supervision is extremely limited (i.e. small $\gamma$) existing PU learning methods can completely collapse due to increased variance in risk estimation(Fig 1, Figure 15).  However, our approach remains uniquely effective even in such scenarios, since when label information is unavailable, it can still learn representations from semantic similarity.
>
> Finally, most PU methods either rely on knowledge of class prior or some other parameter that needs to be estimated. Similarly, existing weakly supervised approaches that try to interpolate between sCL and ssCL also suffer from parameter estimation issue (e.g. in Section A.5.4 we discuss two parametric CL objectives that suffer from similar issue).
>
> Overall, our method shows largest gains over existing PU methods in the low-supervision regime where existing methods can't work well.

---

> > ### Author Response · Authors · 2023-11-23
> >
> > * **High Dimensional Image Datasets**
> > While, we have shown extensive evidence over existing PU vision benchmark data (identical exp setup as prior works e.g. P3MIX-E, vPU) of our approach being competitive or superior over existing PU methods across the board (over all the 10 datasets and several dataset settings).
> > We also add new experiments on an ImageNet subset. In particular we discriminate between ImageWoof vs ImageNette - two subsets of ImageNet-1k widely used in noisy label learning research : https://github.com/fastai/imagenette. Our experimental findings from smaller datasets (CIFAR small images) translate to ImageNet as well (Fig 1, 3, 5, 6, 9). In fact, we note that in ImageNet the gains over previous SOTA e.g. P3Mix-E, vPU, nnPU+MixUp is more prominent even in high supervision regime.
> > On the other hand, even for ImageNet, for limited supervision PU methods fail, whereas our proposed approach remains uniquely effective.

---

### Official Review · Reviewer_xzEP · 2023-11-01

**Soundness:** 3 good
**Presentation:** 3 good
**Contribution:** 3 good
**Rating:** 6
**Confidence:** 3

**Summary:**

This manuscript adopts a simple PU-specific modification of the standard self-supervised contrastive objective to take into account the available weak supervision in the form of labeled positives.

**Strengths:**

This manuscript represents the work tailoring contrastive learning specifically to the PU setting. Based on the self-supervised learning, the PU setting introduced more positive supervised information, which could result in a better performance.

**Weaknesses:**

The experiments are not conducted on a recommendation data set though the PU learning setting applies in recommendation.

**Questions:**

In step (b), assigning pseudo-labels to the unlabeled examples may introduce label noise, Can the methods deal with label noise can improve the performance? I'm interested in seeing these outcomes.

---

> ### Author Response · Authors · 2023-11-23
>
> We thank you for your valuable and constructive feedback. Below we respond to each concern in detail.
>
> * ( Q1 ) **assigning pseudo-labels to the unlabeled examples may introduce label noise, Can the methods deal with label noise can improve the performance ?**
>
> **Connection to Learning under Class Dependent Label Noise :**  PU Learning is closely related to the popular learning under label noise problem, where the goal is to robustly train a classifier when a fraction of the training examples are mislabeled. This problem is extensively studied under both generative and discriminative settings and is an active area of research.
>
> Consider the following instance of  *learning a binary classifier under class dependent label noise* i.e. the class conditioned probability of being mislabeled is $\xi_P = p(\tilde{y}_i \neq y_i | y_i=+1)$ and  $\xi_N = p(\tilde{y}_i \neq y_i | y_i=-1)$ respectively for the positive and negative samples. Recall from Section 3.2 the naive disambiguation-free approach (Li et al., 2022), where the idea is to pseudo label the PU dataset as follows: Treat the unlabeled examples as negative and train an ordinary binary classifier over the pseudo labeled dataset.
>
> It is easy to see that this is an instance of learning with class dependent label noise where $\xi_P = \frac{\pi_P}{\lambda+\pi_p}$ and $\xi_N = 0$. Further, from breakdown point analysis we show  that no ERM estimator can be reliable when $\gamma = \frac{n_P}{n_U} \leq 2 \pi_p -1$ (see Section A.3.2 and Lemma 1). Here, breakdown point $\psi$ of an estimator is simply defined as the smallest fraction of corruption that must be introduced to cause an estimator to break implying $\Delta$ (estimation error) can become unbounded i.e. the estimator can produce arbitrarily wrong estimates.
>
> This result suggests that PU Learning cannot be solved by off-the-shelf label noise robust algorithms and specialized algorithms need to be designed that should be robust in principle even for $\gamma \leq 2 \pi_p -1$
>
> However, note that in PU Learning, we additionally know that a subset of the dataset is correctly labeled i.e.  $p(\tilde{y}_i = y_i = 1 | x_i \in P) = 1$. which leaves us with hope. The question is: can we use this additional information to enable PU Learning even when $\gamma <= 2 \pi_p - 1$ ?
>
> We have included Appendix: Section A.3.2 and Lemma 1 that discusses these backgrounds and insights in more depth.
>
> **Cost Sensitive PU Learning :**
> As discussed in the paper, SOTA cost-sensitive PU learning algorithms tackle this by forming an unbiased estimate of the true risk from PU data (blanchard2010semi) by assuming additional knowledge of the true class prior $\pi_p = p(y=1 |x)$.
> The unbiased estimator dubbed uPU (blanchard2010semi, du2014analysis) of the true risk $R_{PN}(v)$ from PU data is given as:
> $$
> \hat{R}_{pu}(v) = \pi_p \hat{R}_p^+(v) + \textcolor{blue}{\Bigg[\hat{R}_u^-(v) - \pi_p \hat{R}_p^-(v)\Bigg]}
> $$
> Here, $\hat{R}_p^+$ denotes estimated positive risk i.e. the empirical risk estimate over the labeled positives, $\hat{R}_p^-$ is the risk of treating  positives as negative. Since both these estimates are obtained over labeled positives, these estimates suffer from large variance resulting in significant performance degradation when only a few positives are labeled.
>
> Further, these approaches are dependent of class prior which is unavailable / expensive to estimate. Unfortunately, these methods are quite sensitive to class prior estimation error e.g. $\pi_p^*\neq\hat{\pi_p} = 1$ leads to degenerate solution (Also se Fig 12, 13).
>
> We discuss these approaches in more detail in Sec A.3.3.
>
> Recall that, **puCL** simultaneously optimizes over both semantic similarity (feature) and semantic annotation (labeled positives). Meaning, even when amount of labeled example is limited it is able to learn from the semantic similarity by exploiting the geometry of the embedding space. In other words, puCL interpolates between supervised and unsupervised learning - enabling representation learning even in low supervision regime.
>
> Note that, for the downstream classification still needs to be trained from PU data. Thus, instead of relying on existing PU algorithms -- which again suffer from similar issues (low supervision regime and class prior estimation), we propose to perform static clustering on the representation manifold. Intuitively, if contrastive learning is able to learn separable representations, the pseudo-labels are likely to be close to true labels implying has low label noise.
>
> **New Experiment:** We train (only) linear layer over frozen embeddings obtained from puCL, across different SOTA PU algorithms (Fig 15 in addition to Table 1). We consistently observe that  even for linear probing - existing PU algorithms breakdown when label data is scant while puPL remains uniquely effective, while staying competitive at high supervision regime.

---

> > ### Author Response · Authors · 2023-11-23
> >
> > **The experiments are not conducted on a recommendation data set though the PU learning setting applies in recommendation.**
> >
> > Designing label-preserving augmentations in contrastive learning (CL), a task that can be challenging in domains where such augmentations are not readily obtainable. This limitation, however, has not deterred the exploration of CL across various domains, buoyed by its success in fields like vision and text, and notably in multi-modal settings such as CLIP. The active pursuit of domain-specific augmentations is a testament to the adaptability and potential of CL. In diverse areas—from graph data, where augmentations are made by randomly dropping edges, to ranking problems that integrate random noise or mask entries in tabular data—innovative techniques are being developed. These advancements extend to recommendation systems, audio processing, and beyond, highlighting the versatility of CL in handling various data modalities to name a few e.g.[1-5].
> >
> > While extending contrastive learning (CL) across various data modalities, is a fascinating and burgeoning field of research—emphasizing the need for domain-specific augmentations and thoughtful design choices—it's important to note the core objective of this paper.
> >
> > In this paper, as a methodological exploration of contrastive learning (CL), we primarily concentrate on vision experiments. This focus is deliberate and sufficient for our objectives (and consistent with the large body of work in both contrastive learning and PU Learning ). As a methods paper, our primary goal is to elucidate and validate the proposed approaches and techniques within the realm of CL. By using vision data, which offers a rich and well-understood context, we can clearly demonstrate the efficacy and versatility of our methods. Additionally, vision experiments provide a tangible and accessible means for the broader research community to evaluate and replicate our findings. While we acknowledge the potential of extending our methods to other data modalities, this paper intentionally narrows its scope to provide a deep and thorough exploration of contrastive representation learning from PU supervision and develop that into simple and practical PU learning solution with superior empirical performance, without needing additional knowledge like class prior. Through extensive experiments and intuitive theoretical justifications we show that our approach uniquely stays effective even with extremely limited labels, unlike prior PU methods. We conduct experiments on multiple subsets of CIFAR, ImageNet, FMNIST, STL-10, including comparing our method with a large number of baselines consistent with the experimental setups of recent PU Learning literature.
> >
> > This exploration represents an exciting avenue for future research in PU learning, inviting the development of innovative approaches and domain-specific augmentations that can unlock the benefits of CL in less conventional or more complex data environments. Our work lays a foundational framework, opening the door for future studies to adapt and expand upon our methodologies, thereby broadening the applicability and impact of CL across diverse and challenging domains.
> >
> > [1] Majmundar et.al. Met: Masked encoding for tabular data.
> >
> > [2] You et.al. Graph Contrastive Learning with Augmentations.
> >
> > [3] Hou et.al. PRETRAINED DEEP MODELS OUTPERFORM GBDTS IN LEARNING-TO-RANK UNDER LABEL SCARCITY.
> >
> > [4] Guo et.al. SimCSE: Simple Contrastive Learning of Sentence Embeddings.
> >
> > [5] Liu et.al. Contrastive Learning for Recommender System.

---

### Author Response · Authors · 2023-11-23

During the rebuttal period, we have significantly enhanced the enhanced the paper. In summary here are the new figs / results added based on reviews:

(a) Fig 1: ImageNet subset - i. Compare with existing PU methods in terms of generalization.  ii. Convergence comparison of CL methods.  iii\. embedding visualization.

(b) Fig 2, 8: Aligning points on hypercube: To provide more intuition about the underpinnings, we discuss a energy configuration argument to provide intuitive explanations about optimal point configurations of various CL discussed. Also see Section A.5.2.

(c) Fig 3: Linear Probing Choices: Given pretrained embedding our goal is now to train a downstream linear model. In this experiment we take puCL($\gamma$) pretrained encoder (frozen) and train a linear classifier for downstream inference. In particular, we evaluate several popular SOTA PU Learning methods along with the proposed pseudo-labeling based approach.
Our findings are particularly noteworthy in the context of low-data regimes. While traditional PU learning methods often struggle to maintain performance with limited data, our approach consistently demonstrates robust effectiveness.

(d) Fig 6: Comparison of CL objective across $\gamma$ on imagenet and cifar subset.

(e) Fig 9: To study the scenario when p(x) doesn not contain information about p(y|x) we construct three PU subsets from CIFAR:ahrd, medium, easy. see Section A.5.2 for details.

(f) Comparison with Parametric CL: We compare with DCL and MCL two CL objectives that try to modify the infoNCE loss from inferred weak supervision. See Section A.5.4 for details

(g) We also provide evidence (both theoretical Theorem 4 and empirical : Fig 1, 10 that by incorporating PU supervision judiciously puCL not only enjoys better generalization - it also converges faster than ssCL.

---

### Meta-Review · Area_Chair_Tpc6 · 2023-12-05

**Metareview:**

This is a borderline paper worked on positive-unlabeled learning and proposed a novel PU contrastive learning objective as its major contribution. It has received mixed ratings from 3 to 8 where the lowest and highest reviewers are both top experts of closely related topics. They argued that
> **Yt2Y** Although the authors have not addressed my concern. I will keep my score.\
> *Limited insights.* Contrastive learning can be used to improve classification performance is not something interesting. Even if we discord all labels, some contrastive-learning-based methods can achieve 90%+ accuracies on benchmark image datasets (such as CIFAR10, and F-MNIST). Existing PU methods also can be easily tweaked to include contrastive learning to boost their performance.\
> *Unclear advantage to existing methods.* Although the authors propose that their method is general and does not heavily rely on estimating class prior, their method is only limited to image datasets. I fail to see a clear theoretical and practical advantage.

> **sTW4** For the concern that self-supervised contrastive learning without using labels already performs really well, I think the linear probe experiments shown in Table 1 show that we can improve over the ssCL with the proposed PUCL by +0.6, +1.7, +1.3, +1.4 points (depending on the setup). It is interesting to see that the proposal is helpful for learning even better representations, although I agree that it may be marginal and practitioners may end up simply choosing ssCL even when they have PU data.\
> For the concerns about estimating the class prior and the effect on the generality of the proposed method, I don't have much experience in this aspect and do not have a strong opinion.\
> Overall, I still feel positive about this paper, but since I have no direct experience in PU, class-prior estimation, and self-supervised learning, I may be too optimistic. I see merits in accepting the paper but I would also be comfortable if this paper is not presented at the conference this time.

Since they cannot agree with each other, I have taken a closer look at the paper and will talk about my own thoughts. BTW, the proper name "Contrastive Positive Unlabeled Learning" has been used [1] and you may want to have a different name for your own paper/method (and you need a hyphen between positive and unlabeled, i.e., positive-unlabeled learning). I didn't talk about novelty as that is a different method with the same name. I am going to talk about fair conceptual and/or experimental comparison.

This paper used some key components not involved in existing distribution-assumption-free discriminative PU learning, namely, the cluster assumption (CA) and the pseudo-labeling (PL) technique. Then, many claims are actually unfair or even incorrect. For example, as the motivation, the authors wrote in the abstract that
> Furthermore, these methods tend to perform poorly in low-data regimes, especially when very few positive examples are labeled.

which is not true:
- when it is distribution-assumption-based, PU learning can be quite simple yet quite powerful (considering two highly overlapping Gaussian distributions), while note that CA doesn't hold here (two classes but only one big cluster);
- when we have and enjoy CA, generative PU learning that is also distribution-assumption-free can be extremely good given very few P data [2];
- sometimes, with the help of CA, even discriminative PU learning can also be very good [3].

Thus, the claimed motivation does not generally hold, and comparing a method relying on CA against a method without relying on CA is conceptually unfair.

What is more, when CL and PL are both involved, the most relevant baseline is no longer self-supervised CL (SSCL) but supervised CL with PL (SCLwPL) for the sake of representation learning, similar to [4]. Though [4] was designed for partial-label learning, the problem is tightly connected to noisy-label learning and PU learning, and [4] was an outstanding paper honorable mention --- it is indeed very representative about replacing SSCL with (only partially correct) SCL whenever we have access to some weak supervision. Comparing PUCL with this SCLwPL is strictly fair both conceptually and experimentally, and this should be the real motivation of PUCL.

As a consequence, we cannot accept the current version of this submission for publication at ICLR 2024. This promising submission is not ready in its current version and needs a major revision.

[1] Fraud Detection via Contrastive Positive Unlabeled Learning, IEEE International Conference on Big Data (Big Data) 2022.

[2] Generative Adversarial Positive-Unlabeled Learning, IJCAI 2018.

[3] Loss Decomposition and Centroid Estimation for Positive and Unlabeled Learning, TPAMI 2021.

[4] PiCO: Contrastive label disambiguation for partial label learning, ICLR 2022.

**Justification For Why Not Higher Score:**

Many issues. See above.

Moreover, the authors were self-appealing too much. Sometimes, negative reviews are more helpful for improving yourself and/or your paper itself.

**Justification For Why Not Lower Score:**

N/A

---

### Decision · Program_Chairs · 2024-01-16

Reject